# Bacteria employ lysine acetylation of transcriptional regulators to adapt gene expression to cellular metabolism

Magdalena Kremer[1,2,6], Sabrina Schulze [2,6], Nadja Eisenbruch[2], Felix Nagel[3], Robert Vogt[2], Leona Berndt[2], Babett Dörre[2], Gottfried J. Palm [2], Jens Hoppen[2], Britta Girbardt[2], Dirk Albrecht[4], Susanne Sievers [4], Mihaela Delcea[3], Ulrich Baumann [1], Karin Schnetz[5] & Michael Lammers [2] ✉

The *Escherichia coli* TetR-related transcriptional regulator RutR is involved in the coordination of pyrimidine and purine metabolism. Here we report that lysine acetylation modulates RutR function. Applying the genetic code expansion concept, we produced site-specifically lysine-acetylated RutR proteins. The crystal structure of lysine-acetylated RutR reveals how acetylation switches off RutR-DNA-binding. We apply the genetic code expansion concept in *E. coli* in vivo revealing the consequences of RutR acetylation on the transcriptional level. We propose a model in which RutR acetylation follows different kinetic profiles either reacting non-enzymatically with acetyl-phosphate or enzymatically catalysed by the lysine acetyltransferases PatZ/YfiQ and YiaC. The NAD⁺-dependent sirtuin deacetylase CobB reverses enzymatic and non-enzymatic acetylation of RutR playing a dual regulatory and detoxifying role. By detecting cellular acetyl-CoA, NAD⁺ and acetyl-phosphate, bacteria apply lysine acetylation of transcriptional regulators to sense the cellular metabolic state directly adjusting gene expression to changing environmental conditions.

All organisms of life are dependent on the continuous sensing of their environment. These sensing machineries, including one-component regulatory systems, enable the cell to constantly perceive the extra- and intracellular status and to translate changes directly into altered gene expression allowing a quick response to rapidly changing conditions[1–5]. For bacteria, these systems are highly important as these organisms often live in environments characterized by fast and dynamic changes, i.e. in the availability and composition of nutrients or in the presence of compounds that affect cellular function and growth such as antibiotics secreted by other organisms like fungi[5–7]. About 20 classes of one-component regulatory systems were

identified in bacteria, many of which are characterized functionally and structurally up to atomic resolution[7–9]. One-component regulators of the TetR (Tetracycline repressor protein class D) family are homodimers and contain a C-terminal ligand-binding domain (LBD) and an N-terminal DNA-binding domain (DBD) encompassing a helix-turn-helix (HTH) motif important for DNA-binding[10–12]. TetR proteins were shown to be involved in a variety of cellular processes ranging from antibiotic production and resistance as well as production of small-molecule exporters to quorum sensing and metabolism[7,13–19]. The founding member of the TetR-family, TetR, was shown to bind tetracycline with its C-terminal LBD resulting in conformational changes

[1]Institute of Biochemistry, University of Cologne, Zülpicher Straße 47, 50674 Cologne, Germany. [2]Institute of Biochemistry, Department of Synthetic and Structural Biochemistry, University of Greifswald, Felix-Hausdorff-Str. 4, 17489 Greifswald, Germany. [3]Institute of Biochemistry, Department of Biophysical Chemistry, University of Greifswald, Felix-Hausdorff-Str. 4, 17489 Greifswald, Germany. [4]Institute of Microbiology, Department of Microbial Physiology and Molecular Biology, University of Greifswald, Felix-Hausdorff-Str. 8, 17489 Greifswald, Germany. [5]Institute for Genetics, University of Cologne Zülpicher Straße 47a, 50674 Cologne, Germany. [6]These authors contributed equally: Magdalena Kremer, Sabrina Schulze. ✉e-mail: michael.lammers@uni-greifswald.de

lowering the affinity of the N-terminal HTH DNA-binding motif for the tetracycline resistance gene (*tetA*) promoter resulting in expression of the tetracycline efflux pump TetA via de-repression[10,12].

The TetR-family member RutR (pyrimidine utilization repressor) was identified as a transcriptional repressor for genes of the *rutAG*-operon and the *rutR* gene as well as an indirect activator of the *carAB*-operon (Fig. 1a)[19]. Mapping of the RutR DNA-target sites revealed that RutR binds to an approximately 16 bp palindromic sequence, the RutR consensus sequence (Fig. 1b)[19,20]. These RutR-boxes are located upstream of the target genes but also in intergenic sequences with unknown consequences[21]. Gene expression analyzes assessing the mRNA level by microarrays revealed that the *rutAG* operon is expressed under nitrogen limitation in an NtrC-dependent manner suggesting that RutR is inactivated under those conditions[22].

The *rutAG* operon contains seven genes (*rutA*-*rutG*) encoding for seven proteins (six enzymes and one uracil transporter) necessary for degrading pyrimidines to form 3-hydroxypropionic acid[23,24]. RutR binds to the promoter region between the divergently transcribed *rutAG* operon and *rutR* gene (Fig. 1a). Enzymes encoded by the *rutAG*-operon were characterized structurally and/or functionally[17,24–26]. The gene cluster encodes an uracil transporter (RutG) suggesting that it is important for utilizating exogenous pyrimidines. In fact, it was shown that the Rut pathway enables the use of pyrimidines as sole nitrogen source at room temperature but not at 37 °C[23]. This led to the hypothesis that the pathway is important for metabolic adjustment if *E. coli* undergoes a drastic environmental change such as leaving the mammalian gut[18]. Moreover, later studies showed the importance of RutR not only for pyrimidine degradation but also for its synthesis by regulating the pyrimidine de novo synthesis via activation of *carAB* operon expression (Fig. 1a, b)[19]. In fact, the *carAB* operon was shown to be the highest affinity target for RutR[16]. The de novo synthesis of pyrimidines begins with carbamoyl phosphate. Carbamoyl phosphate

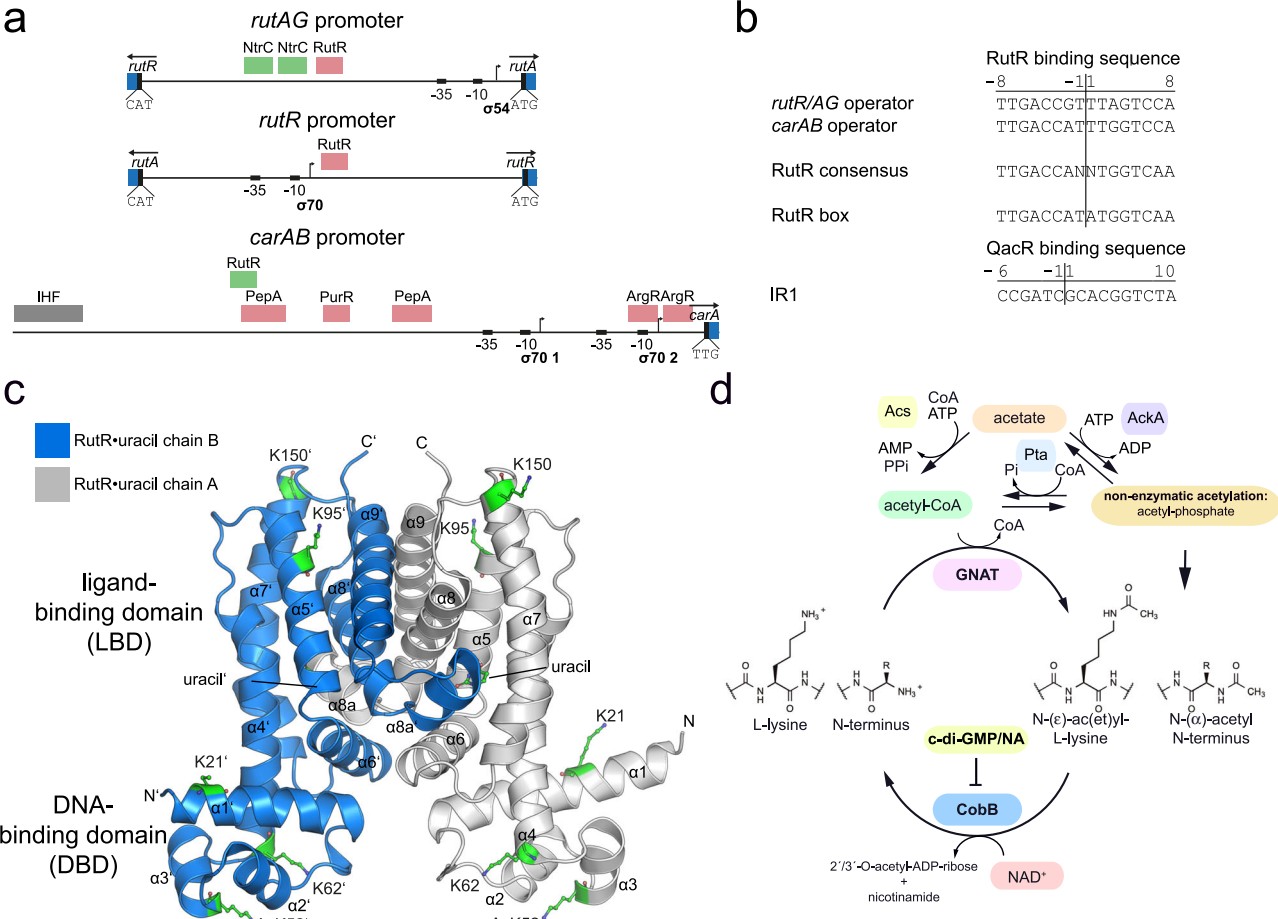

**Fig. 1 | The transcriptional regulator RutR is lysine acetylated. a** Promoter organization of RutR target genes (*rutAG* (upper panel), *rutR* (middle panel), and *carAB* (lower panel)). Open reading frames are depicted as blue boxes with labels/arrows indicating the gene/coding direction. Translational start codons are highlighted as black boxes. Small black boxes represent the promoter −35 and −10 elements. An arrow shows the transcription start site initiated by the indicated sigma factor. The red box shows the binding site of the repressor RutR in *rutAB* and *rutR*. Green boxes mark binding sites of NtrC. For *carAB* RutR acts as activator and is shown by a green box overlapping with PepA (aminopeptidase A) repressor binding site (red box). Further repressor binding sites (purine regulator (PurR), arginine regulator (ArgR)) are also shown by red boxes. The grey box represents the binding site of the integration host factor (IHF). **b** Alignment of the RutR-binding sequences in the promoter regions of *rutR*/*rutAG* and *carAB* in comparison to the defined RutR consensus sequence. The binding sequence of the RutR-related transcriptional regulator QacR from *Staphylococcus aureus* in the IR1 operator sequence downstream of the *qacA* and *qacB* promoters is shown. Numbering shows the position relative to the inversion site of the palindromic binding motifs. **c** Position of the lysines reported to be lysine-acetylated and studied here are highlighted in the structure of RutR•uracil (PDB: 4JYK). K21 is located in the N-terminal α1 helix, K52 is within α3 of the helix-turn-helix (HTH) motif of the DNA-binding domain (DBD), K62 is in the N-terminus of helix α4, K95 in the N-terminal side of α5 and K150 lies at the top of the ligand-binding domain (LBD) at the C-terminal end of α7. **d** Regulatory cycle of lysine acetylation in *Escherichia coli*. N-(ε)-lysine acetylation and/or N-(α)-acetylation is regulated enzymatically by the lysine acetyltransferases (KATs) PatZ /YfiQ, PhnO, YiaC, RimI, and YjaB using acetyl-CoA as acetyl group donor molecule or non-enzymatically by acetyl-phosphate. Acetyl-phosphate is generated by acetate-kinase (AckA) or by phosphotransacetylase (Pta). *E. coli* encodes one single NAD⁺-dependent sirtuin deacetylase, i.e. CobB, inhibited by nicotinamide (NA). The long isoform of CobB is inhibited by the second messenger cyclic-di-GMP (c-di-GMP)[38].

synthetase encoded by the *carAB* operon catalyzes the formation of carbamoyl phosphate from bicarbonate and glutamine in an ATP-dependent way[27]. Carbamoyl phosphate is a precursor not only for pyrimidine synthesis but also for synthesis of purines and of arginine[28,29]. The regulation of the expression of the *carAB* operon is tightly controlled at the transcriptional level by multiple transcriptional regulators[27,30–34]. In absence of uracil, RutR was shown to bind to the *carAB* promoter P1 located upstream of the *carAB* operon[20]. RutR-binding results in de-repression of the *carAB*-operon by interfering with the negative transcriptional regulator PepA, thereby, indirectly resulting in the activation of *carAB* expression (Fig. 1a). By modulating the pyrimidine degradation and synthesis pathways via transcriptional regulation of the *rutAG* operon and the *carAB* operon, RutR is an important regulator for pyrimidine utilization[17,19,21].

Structural analyzes of TetR transcription factors in complexes with DNA showed that they bind DNA as dimers[16,35]. Sequence specificity of RutR-DNA-binding arises by the formation of contacts to residues of the helix α3 from the HTH motif, namely K52 and K62, with bases of the major groove. Additional DNA sequence-independent contacts are described for several TetR proteins of residues N-terminal of the HTH-motif (α2-α3) to the DNA sugar-phosphate backbone[16,20,35]. For RutR, a complex structure with DNA is not known. However, the structure of RutR in complex with uracil has been solved by X-ray crystallography[17]. In analogy to structures of other TetR-family members, the uracil ligand is bound in the C-terminal LBD (Fig. 1c)[17]. The fact that uracil was co-purified with RutR from overexpression in *E. coli* together with a high occupancy in the crystal structure suggests that uracil binds with high affinity[17]. This indicates a regulatory mechanism in which uracil binding stabilizes a conformation of RutR that is incompatible with DNA-binding. For RutR bound to DNA, the addition of rather low uracil concentrations in the (sub)micromolar range shifts the equilibrium towards a conformation that precludes DNA-binding[17,19,20].

Recent data reported that RutR is targeted by post-translational lysine acetylation (Fig. 1c)[36–38]. In eukaryotes it is known since the 1960s that acetylation of lysine side chains in the unstructured, flexible N-terminal histone tails affects RNA synthesis[39]. Today it is established that histone acetylation is a major regulator of gene expression[39,40]. Regulation of transcription factors by lysine acetylation might exert analogously important functions in bacteria, i.e. to translate the metabolic state via lysine acetylation to alterations of gene expression patterns by modulating transcription factor activity[41]. Similar mechanisms were shown for the transcriptional regulators such as CRP, RcsB, and others[40,42–54]. Lysine acetylation is a post-translational modification that is tightly connected to the cellular metabolic state[43]. Lysine acetyltransferases (KATs) use acetyl-CoA as acetyl group donor for acetylation (Fig. 1d)[55–59]. In *E. coli*, the best-studied KAT is PatZ/YfiQ[44,57,60–62]. Recently, four additional Gcn5-like acetyltransferases were identified acting as epsilon-lysine acetyltransferases: RimI, YjaB, YiaC and PhnO[63–66]. All bacterial KATs show homologies to the Gcn5-related N-acetyltransferase (GNAT)-family of mammalian KATs[60,66,67]. Moreover, in bacteria, the high-energy molecule acetyl-phosphate, extensively produced under conditions of overflow metabolism by the reversible activities of acetate-kinase (AckA) and phospho-transacetylase (Pta), was shown to result in non-enzymatic lysine acetylation, which can also occur in a site-specific manner to influence protein function (Fig. 1d)[68–70]. Lysine-deacetylases (KDACs) remove the acetyl group from the ε-amino group of lysine side chains. In *E. coli* only one KDAC, i.e. the sirtuin CobB, was identified (Fig. 1d)[44,71–74]. Recently, the presence of a structurally and mechanistically unrelated lysine-deacetylase, YcgC, was suggested to constitute another unrelated KDAC in *E. coli*[36]. In two additional studies, RutR was identified to be a substrate for YcgC[75,76]. Our data suggest, that YcgC does not show the proposed lysine-deacetylase activity towards RutR, and RutR is exclusively deacetylated by CobB[37].

We and others identified various lysine acetylation sites in RutR within the N-terminal DNA-binding domain including the HTH motif as well as within the C-terminal LBD[36,37,75,76]. This suggests that lysine acetylation might be a regulatory system controlling fundamental functions of RutR: DNA-binding, protein-protein interaction, uracil binding or other functions such as protein stability. We postulate that lysine acetylation is a regulatory mechanism to adjust RutR function to the cellular metabolic state, not only as response to fluctuations in the nitrogen supply but more generally to metabolic challenges such as conditions of starvation or metabolic fuel switching. Our data on RutR highlight how bacteria might use post-translational lysine acetylation of transcriptional regulators to translate changes sensed in the cellular metabolic state into alterations in gene expression programmes, a mechanism functionally similar to histone-acetylation in eukaryotes.

## Results

### RutR is constitutively expressed in *E. coli* BW30270 and *E. coli* U65

RutR was reported to be an important regulator for pyrimidine metabolism[19]. To analyze how RutR transcriptional regulator activity is adjusted, we initially analyzed the expression of *rutR* during bacterial growth. To this end, we genomically inserted a DNA sequence encoding for a FLAG-tag downstream of *rutR* in two distinct *E. coli* K12 strains: BW30270 and U65 (Supplementary Table 1). Following this strategy, the *rutR* gene is under control of its genuine promoter allowing the physiological expression of *rutR* and detection of RutR-FLAG from cell lysates by immunoblotting. Notably, neither the genomic deletion of *rutR* nor the expression of *rutR* as C-terminal FLAG-fusion protein in *E. coli* BW30270 or U65 interferes with bacterial growth in complex medium (Fig. 2a)[19]. Earlier supporting studies on RutR protein levels during *E. coli* growth have shown that RutR protein levels are almost constant during lag-phase, exponential growth phase, and stationary phase as likewise shown here for *E. coli* BW30270 cells (Fig. 2b)[19]. This suggests that *rutR* is constitutively expressed in different *E. coli* strains under the conditions analyzed, i.e. conditions of high and low nutrient availability. However, this does not rule out that RutR protein levels might vary under different physiological conditions not analyzed here. Still, these findings fed our hypothesis of a protein level-independent RutR activity regulatory mechanism such as post-translational modifications (PTMs). RutR acetylation at several lysines within the DNA-binding domain (DBD) (K21, K52, and K62) and the ligand-binding domain (LBD) (K95, K150) was reported in several studies demonstrating this kind of lysine modification potentially being of high functional importance (Fig. 1c)[36–38,75,77]. We aimed at elucidating the regulatory role of post-translational lysine acetylation to modulate RutR function. This PTM might allow a precise coordination of RutR function dependent on the metabolic state in addition to the reported regulatory role of uracil.

### Preparation of site-specifically lysine-acetylated RutR proteins using genetic code expansion

To study the impact of lysine acetylation on RutR function, we prepared site-specific lysine-acetylated RutR proteins (RutR AcK) by genetic code expansion: RutR AcK21, AcK52, AcK62, AcK95 and AcK150[36–38,75,77]. Studies performed in our laboratory show that the real impact of lysine acetylation cannot be mimicked in all cases by mutations such as Lys to Gln (K to Q) to neutralize the positive charge at the lysine side chain or Lys to Arg (K to R) to conserve the positive charge at the side chain[78]. To nevertheless allow mechanistic investigation, we genetically encoded acetyl-L-lysine (AcK) for site-specific incorporation into RutR as a response to an amber-stop codon. We used a synthetically evolved, orthogonal acetyl-lysyl-tRNA-synthetase (AcKRS3)/*Mbt*RNA$_{CUA}$ pair from *Methanosarcina barkeri*[79–81]. We constructed a single-plasmid system based on pRSFDuet1 encoding for the acetyl-lysyl-tRNA-synthetase/*Mbt*RNA$_{CUA}$-pair and additionally carrying the *rutR* gene under

# a
# b

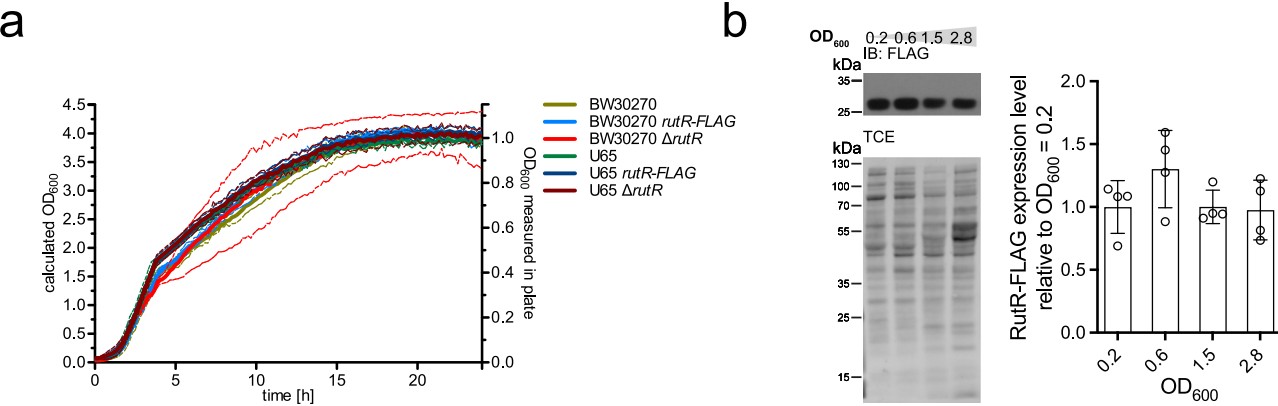

**Fig. 2 | Endogenous RutR-FLAG protein levels are constant during *E. coli* growth. a** Genomic insertion of a FLAG-tag encoding region downstream of *rutR* or genomic deletion of *rutR* does not affect growth behavior of *E. coli* BW30270 and *E. coli* U65 in complex medium (LB). The experiment was performed in duplicates and the graph depicts the means (solid lines) ± standard deviations (dashed lines) of both recorded replicates (*n* = 2). Source data are provided as Source Data file. **b** Endogenous RutR-FLAG protein level is constant during *E. coli* BW30270 growth.

Left panel: immunoblotting (IB) of RutR-FLAG with anti-FLAG antibody (IB: FLAG). 2,2,2-trichloroethanol staining (TCE) served as loading control. Right panel: Quantitative analysis of the RutR-FLAG expression level was performed by analysis of immunoblottings using ImageJ software. One exemplary out of four replicates is shown and bars depict means ± standard deviations. Significance was tested by t-tests and no significant differences were detected by application of a significance level of $p < 0.05$ (*n* = 4). Source data are provided as Source Data file.

the control of a T7 promoter with an amber-stop codon placed at the site selected for acetyl-L-lysine (AcK) incorporation[79]. Applying this technology, we expressed the acetylated proteins RutR AcK21, AcK52, AcK62, AcK95 and AcK150 in *E. coli* BL21 (DE3). All proteins were expressed and purified to yield protein in quality and purity sufficient for biochemical and biophysical experiments (Supplementary Fig. 1a, b). All RutR proteins and acetylated variants eluted as dimers from the analytical size-exclusion chromatography (SEC)-column suggesting that the acetylation does neither interfere with protein folding nor its oligomeric state (Supplementary Fig. 1a). Moreover, as a quality control, the correct site-specific incorporation of acetyl-L-lysine was shown by immunoblotting using a specific anti-acetyl-L-lysine antibody (Supplementary Fig. 1b). The immunoblotting shows that not all acetylation sites are recognized to the same extent by the antibody, supporting previous observations that suggest a sequence bias of the antibody[82]. Therefore, additional molecular mass determination by electrospray-mass spectrometry (ESI-MS) was performed and showed the correct molecular weights of all RutR proteins and LC-MS/MS experiments confirm the site-specific acetyl-lysine incorporation (Supplementary Fig. 2; Supplementary Fig. 3). These acetylated RutR proteins were analyzed further to assess a potential impact of the RutR acetylation on DNA-binding.

## RutR acetylation at K52 and K62 within the HTH DNA-binding domain abolishes binding to both *rutAG* and *carAB* promoter

We analyzed the impact of RutR acetylation on DNA-binding by a dual semi-quantitative and quantitative approach consisting of electrophoretic mobility shift assays (EMSAs) and isothermal titration calorimetry (ITC) experiments, respectively (Fig. 3). For the interaction studies, we selected the RutR DNA-binding regions located upstream of the *rutAG*-operon (box_{rutA}), and of the *carAB*-operon (box_{carA}), the major targets of RutR (Supplementary Table 2). EMSA assays show that acetylation of K52 and K62 in RutR drastically impairs DNA-binding to both box_{rutA} and box_{carA} to a similar extent, while the other acetylated RutR proteins (RutR AcK21, RutR AcK95, RutR AcK150), as well as non-acetylated RutR (RutR WT), bind to the dsDNA fragments (Fig. 3a). We also prepared the mutant RutR K52Q, used as charge-neutralizing acetylation mimic mutant, and the mutant RutR K52R, used to conserve the non-acetylated state, to further analyze if these mimicking mutants can be used to study the impact of lysine acetylation also in vivo. Interestingly, while RutR K52R behaves as RutR WT in EMSAs, the

RutR K52Q mutant does not completely switch-off DNA-binding to both, box_{rutA} and box_{carA}, suggesting that it does not perfectly mimic K52-acetylation (Supplementary Fig. 4; Supplementary Fig. 5). In a homology model, the RutR acetylation sites K52 and K62 are located within the HTH motif contacting a DNA base and the phosphate backbone respectively explaining their role in DNA-binding[20]. We also assessed the impact of uracil on the interaction of RutR and DNA by EMSA experiments. The addition of uracil impaired RutR-DNA binding. For the acetylated RutR variants that show reduced DNA-binding, addition of uracil additively impaired DNA-binding suggesting acetylation and uracil acting through a different mechanism (Supplementary Fig. 6).

To confirm these results and for a quantitative analysis of the impact of RutR K52- and K62-acetylation on DNA-binding as well as to characterize their impact on the binding mechanism, we analyzed the interactions of acetylated RutR proteins with box_{rutA} and box_{carA} dsDNA by ITC (Fig. 3b, c; Supplementary Fig. 5b, c; Supplementary Table 3). These data elucidate that all interactions are endothermic and solely driven by the change in the reaction entropy. All reactions show a stoichiometry of approximately two, indicating that two molecules of RutR, i.e. a RutR dimer, bind to one molecule of dsDNA. While non-acetylated RutR binds to both box_{rutA} and box_{carA} dsDNA with a high nanomolar affinity ($K_D$(box_{rutA}) = 44 nM; $K_D$(box_{carA}) = 28 nM), acetylation of RutR at K52 completely switches-off RutR box_{rutA} and box_{carA} dsDNA-binding. In contrast, RutR AcK62 completely abolishes box_{rutA} dsDNA-binding, while retaining some binding towards box_{carA} dsDNA. RutR AcK62 shows 40-fold reduced binding towards box_{carA} compared to non-acetylated RutR (Fig. 3b, c; Supplementary Fig. 5c; Supplementary Table 3). Considering that there are only 150-300 molecules of RutR per cell corresponding to an intracellular concentration of approximately 0.5 μM RutR, this residual binding observed for RutR AcK62 to box_{carA} dsDNA might not be of physiological relevance[19]. Notably, RutR Ack21, AcK95, and AcK150 did not affect binding towards either box_{rutA} or box_{carA} dsDNA as judged by ITC (Supplementary Table 3, Supplementary Fig. 7). EMSA and ITC experiments agree in showing that RutR K52R binds with a similar nanomolar affinity to both dsDNA-fragments compared to non-acetylated RutR WT. However, RutR K52Q, in contrast to RutR AcK52, retains some DNA-binding affinity towards both, box_{rutA} and box_{carA} dsDNA (Fig. 3b, c; Supplementary Fig. 5; Supplementary Table 3). These data suggest that with respect to

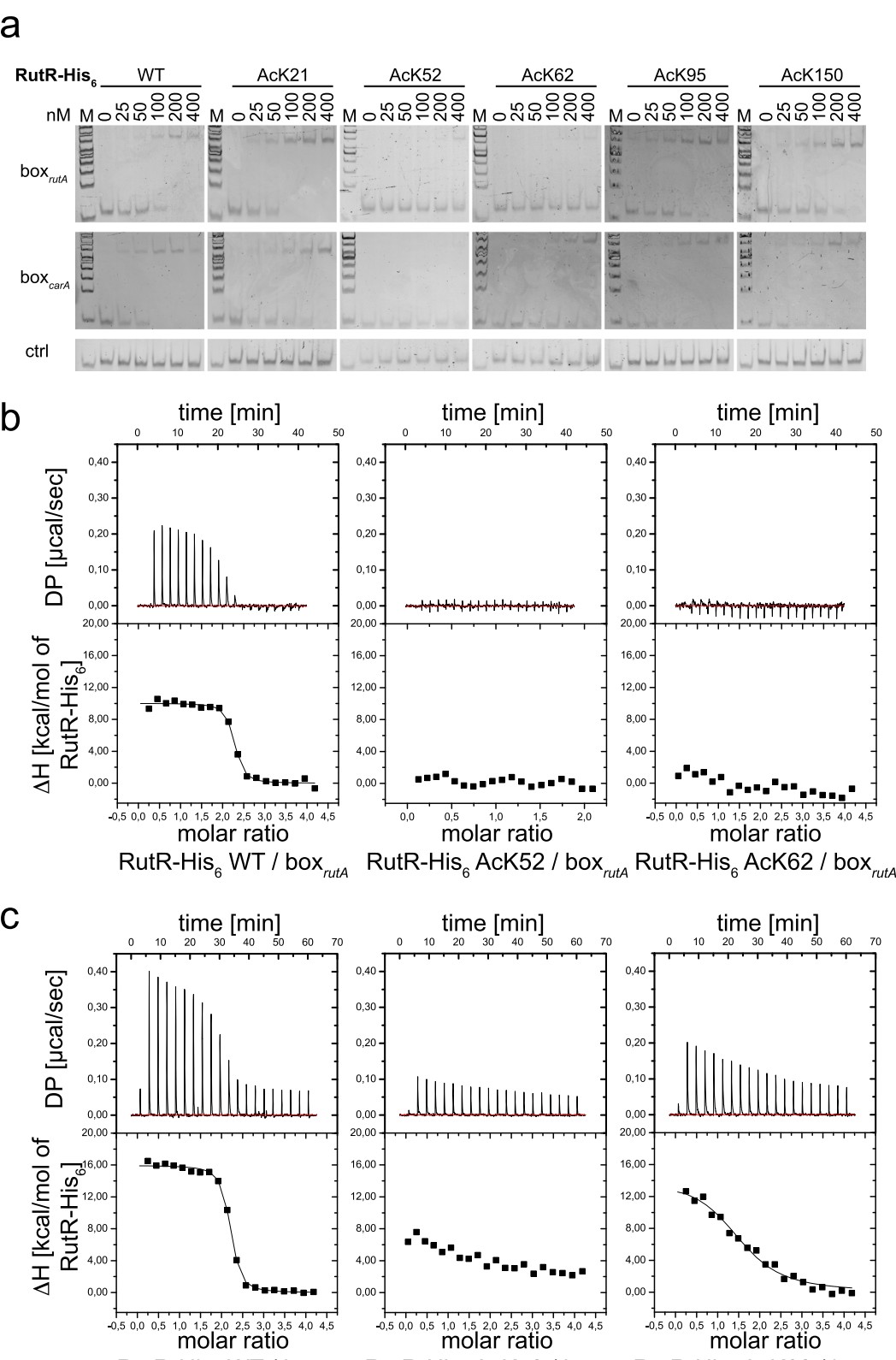

DNA-binding RutR K52R can be used in vivo to conserve the non-acetylated, positively-charged state (Supplementary Fig. 5b, c). However, RutR K52Q does not completely abolish binding to both DNA-fragments and is therefore not a perfect mimic for K52-acetylation. Further factors such as the cellular concentration of the RutR K52Q protein determine whether it can be used to study all aspects of RutR lysine acetylation in vivo.

## Transcriptional effect of RutR K52-acetylation is imperfectly mimicked by RutR mutations at the molecular level

To assess if RutR K52Q and RutR K52R are mutants are able to mimic the impact of K52-lysine acetylation in vivo, we constructed the strain *E. coli* U65 with genomic deletion of *rutR* and insertion of $P_{carA}$-*lacZ* (*E. coli* U65 Δ*rutR* $P_{carA}$-*lacZ*) allowing expression of *lacZ* encoding for β-galactosidase under the control of the endogenous *carAB*

**Fig. 3 | Acetylation of RutR at K52 and K62 abolishes DNA-binding shown by electrophoretic-mobility shift assays (EMSAs) and isothermal titration calorimetry (ITC). a.** The protein-dsDNA interactions between RutR variants and box$_{rutA}$ and box$_{carA}$ promoter dsDNA fragments were studied in EMSAs. Acetylation of RutR at K52 and K62 strongly impairs dsDNA-binding. One exemplary result of at least three replicates is shown ($n$ = 3). Source data are provided as Source Data file. **b.** K52- and K62-acetylation in RutR abolishes binding towards box$_{rutA}$. Interaction between non-acetylated RutR WT, and K52-/K62-acetylated RutR and promoter box$_{rutA}$ DNA analyzed by ITC. Shown are exemplary ITC traces (DP: differential power). All interactions were determined at least in three biologically independent experiments and the values are given as means ± standard deviations ($n \geq 3$; Supplementary Table 3). Source data are provided as Source Data file. **c.** K52- and K62-acetylation in RutR impairs binding towards box$_{carA}$ as shown by ITC. Shown are exemplary ITC traces (DP: differential power). All interactions were determined at least in three biologically independent experiments and the values are given as means ± standard deviations ($n \geq 3$; Supplementary Table 3). Source data are provided as Source Data files.

promoter. This enables us to assess the transcriptional regulator activity of RutR by measuring the β-galactosidase activity in cells ectopically expressing non-acetylated RutR WT, K52Q, or K52R mutants. To this end, *E. coli* U65 Δ*rutR* P$_{carA}$-*lacZ* was transformed with pRSFDuet-1 encoding for RutR WT, RutR K52Q, or RutR K52R to analyze the extent a lysine acetylation can be analyzed by a K to Q mutant (mimic for acetyl-lysine) or by a K to R mutant (to conserve the non-acetylated state, i.e. a charge-conserving mutant) (Fig. 4a). As controls, *E. coli* U65 P$_{carA}$-*lacZ* and *E. coli* U65 Δ*rutR* P$_{carA}$-*lacZ* were transformed with the empty vector pRSFDuet-1. For ectopically expressed RutR WT, RutR K52Q, and RutR K52R, we obtained a similar protein level upon addition of 10 μM IPTG as judged by anti-His$_6$ immunoblotting (Fig. 4a, left panel). Although *E. coli* U65 does not contain a genomic insertion of the DE3 lysogen encoding for T7-DNA polymerase, we observed that ectopic expression of *rutR* from pRSFDuet-1 shows a dependence on the concentration of IPTG added (Supplementary Fig. 8). We found that RutR WT is able to significantly de-repress, i.e indirectly induce, expression of *lacZ*, as indicated by the higher β-galactosidase activity obtained in comparison to the empty vector controls (Fig. 4a, right panel). For RutR K52Q, we observed a β-galactosidase activity similar to the empty vector control suggesting that in this assay under these experimental conditions RutR K52Q is a reliable mimic for RutR K52-acetylation and that charge neutralization constitutes an important mechanism to switch-off RutR transcriptional regulator activity (Fig. 4a). Furthermore, we can conclude that presence of RutR at the endogenous level is sufficient to de-repress β-galactosidase expression as the β-galactosidase activity measured for *E. coli* U65 P$_{carA}$-*lacZ* (WT-empty) is significantly increased compared to *E. coli* U65 Δ*rutR* P$_{carA}$-*lacZ* (Δ*rutR*-empty) (Fig. 4a). Intriguingly, our data show that ectopic expression of *rutR* K52R does not increase β-galactosidase activity to the level obtained for ectopically expressed non-acetylated RutR WT. Instead, RutR K52R showed a significantly reduced β-galactosidase activity compared to RutR WT, indicating that in vivo this mutant does not perfectly mimic the non-acetylated state at the molecular level (Fig. 4a, right panel). As we show that RutR K52R does bind in vitro to box$_{rutA}$ and box$_{carA}$ dsDNA with a similar high nanomolar affinity as RutR WT (Supplementary Fig. 3; Supplementary Table 3) other unknown mechanisms must exist in vivo that explain these differences observed for the RutR K52R and RutR WT transcriptional regulator activity (Fig. 4a, right panel). However, while RutR K52R does not perfectly mimic the non-acetylated state in vivo, we observed a significantly reduced β-galactosidase activity for RutR K52Q compared to RutR K52R, although not reaching the RutR WT level, showing that in principle these mutants can be used to decipher the role of RutR K52-acetylation in vivo. This suggests that acetylation at K52 might exert a steric mechanism, additionally to the main electrostatic mechanism, represented by RutR K52Q, to interfere with RutR transcriptional regulator activity. These data show that while in vitro the RutR K52Q mutant does not fully recapitulate the full impact of RutR K52-acetylation on DNA-binding, in vivo the RutR K52R-mutant does not fully mimic the non-acetylated state. To unravel the full impact of K52-acetylation of RutR on its transcriptional regulator activity we next applied genetic code expansion to incorporate acetyl-lysine into RutR in vivo.

## Using genetic code expansion in vivo to assess how the transcriptional control of gene expression is affected by RutR K52-acetylation

Next, we examined whether acetylation of RutR at K52 (RutR AcK52) affects its capacity to function as a transcriptional regulator. Our data on RutR-binding to box$_{carA}$ and box$_{rutA}$ dsDNA suggest that the mutation of RutR K52Q does not perfectly mimic all aspects of K52-acetylation, i.e. retaining some binding affinity towards box$_{rutA}$ and box$_{carA}$ dsDNA in contrast to RutR AcK52. Moreover, RutR K52R does not perfectly mimic the non-acetylated state in vivo. To this end, we aimed at site-specific incorporation of acetyl-lysine into RutR in vivo to assess the real impact of K52-acetylation on RutR transcriptional regulator function. Thus, we again used the strain *E. coli* U65 Δ*rutR* P$_{carA}$-*lacZ* with genomic deletion of *rutR* and insertion of P$_{carA}$-*lacZ* allowing expression of *lacZ* encoding for β-galactosidase under the control of the endogenous *carAB* promoter. This strain was transformed with pRSFDuet-1/*rutRK52$_{amber}$*/*ackRS3*/*MbpylT* allowing to genetically encode acetyl-L-lysine at position K52 in RutR. This enables to produce K52-acetylated RutR protein within living *E. coli* cells as described above. To express non-acetylated RutR WT, *E. coli* U65 Δ*rutR* P$_{carA}$-*lacZ* was transformed with pRSFDuet-1/*rutR*/*ackRS3*/*MbpylT*. As a control, *E. coli* U65 Δ*rutR* P$_{carA}$-*lacZ* was transformed with the empty vector pRSFDuet-1/*ackRS3*/*MbpylT*. To obtain a similar expression level of ectopically expressed *rutR* WT and of *rutRK52$_{amber}$* we adjusted the IPTG concentrations used for induction of expression at OD$_{600}$ of 0.6 (*rutR* WT: 0, 3, 5, 10, 15 μM IPTG; *rutRK52$_{amber}$*: 1 mM IPTG) (Fig. 4b). This revealed that upon addition of 10 μM IPTG and 1 mM IPTG a similar protein level was obtained for non-acetylated and K52-acetylated RutR, respectively (Fig. 4b).

This time, we performed reporter assays with *E. coli* U65 Δ*rutR* P$_{carA}$-*lacZ* to characterize the transcriptional regulator activity of non-acetylated RutR WT and K52-acetylated RutR by measurement of β-galactosidase activity (Fig. 4c). To show that the IPTG concentration does not affect β-galactosidase background expression or activity in absence of RutR, we compared *E. coli* U65 Δ*rutR* P$_{carA}$-*lacZ* transformed with the empty vector (control 1, with 10 μM IPTG) and *E. coli* U65 Δ*rutR* P$_{carA}$-*lacZ* transformed with the empty vector (control 2, with 1 mM IPTG). These experiments showed no statistically significant difference in β-galactosidase activity suggesting that the variation in IPTG concentration did not affect β-galactosidase expression (Fig. 4c). For the K52-acetylated RutR we observed that it significantly represses β-galactosidase activity compared to the *E. coli* U65 Δ*rutR* P$_{carA}$-*lacZ* ectopically encoding for non-acetylated RutR WT to a level similar to the empty vector controls (Fig. 4c). This shows that RutR K52-acetylation completely switches-off its transcriptional regulator activity. To compare the protein levels of ectopically expressed RutR WT, RutR AcK52 and endogenous RutR, we expressed these as C-terminal FLAG-tag fusion proteins to allow immunodetection by staining with an anti-FLAG antibody (Supplementary Fig. 8a). These data show that endogenous RutR protein levels are significantly reduced compared to ectopically expressed RutR WT and RutR AcK52 at all IPTG concentrations analyzed. Importantly, also analyzing these ectopically expressed RutR-FLAG fusion proteins shows that K52-acetylation in RutR significantly represses β-galactosidase expression from the *carAB*-promoter (Supplementary Fig. 8b). Importantly, as

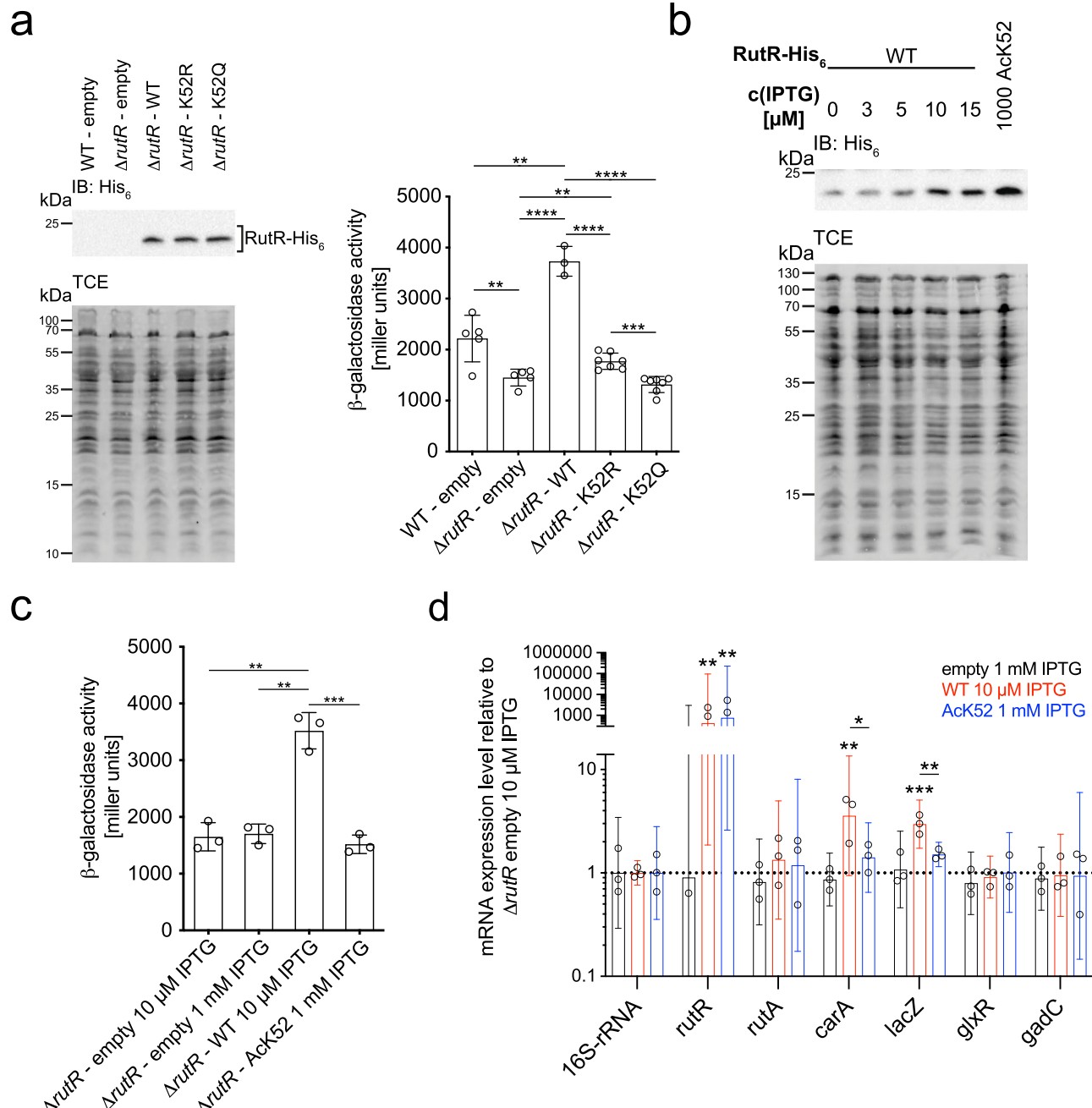

**Fig. 4 | Applying the genetic code expansion concept in vivo in *E. coli* to elucidate that RutR AcK52 affects its transcriptional regulator activity.**
**a** Acetylation mimetic mutations K52R and K52Q are imperfect to study RutR K52-acetylation. Left panel: SDS-PAGE gel shows a similar expression level of RutR wild-type and the K52Q and K52R mutants. Right panel: transcriptional reporter β-galactosidase assays in the absence and presence of *rutR* as well as upon ectopic expression of non-acetylated and mutated RutR. At OD$_{600}$ = 0.6 cells were analyzed by *lacZ* reporter assay. Experiments were performed in three biologically independent experiments (n = 3). Bars depict means ± standard deviations of determined miller units. Statistical significance (**: p ≤ 0.01; ***: p ≤ 0.001; ****: p ≤ 0.0001) was tested using t-tests. Source Data are provided as Source Data files. **b** Adjustment of similar protein levels of non-acetylated RutR and acetylated RutR AcK52. *E. coli* U65 Δ*rutR* P$_{carA}$-*lacZ* was transformed with pRSFDuet-1/*rutR* or pRSFDuet-1/*rutRK52*$_{amber}$, respectively. At OD$_{600}$ = 0.6 cultures were harvested and 30 µg of proteins from cell lysates were analyzed in immunoblots (IB), probed with anti-His$_6$-AB, and related to total protein staining with TCE. Source Data are

provided as Source Data files. **c** Acetylation of K52 in RutR impairs β-galactosidase expression. Transcriptional reporter β-galactosidase assays were conducted upon ectopic expression of non-acetylated and acetylated RutR. *E. coli* U65 Δ*rutR* P$_{carA}$-*lacZ* was transformed with pRSFDuet1 empty, pRSFDuet1/*rutR*, or pRSFDuet-1/*rutRK52*$_{amber}$ as indicated. Cultures were harvested at OD$_{600}$ = 0.6 and analyzed in *lacZ* reporter assays. Bars depict means ± standard deviations of determined miller units. Experiments were performed in three biologically independent experiments (n = 3) and statistically analyzed using t-tests (**: p ≤ 0.01; ***: p ≤ 0.001). Source Data are provided as Source Data files. **d** Gene expression analysis by qRT-PCR upon ectopic expression or non-acetylated and acetylated RutR. *E. coli* U65 Δ*rutR* P$_{carA}$-*lacZ* was transformed with pRSFDuet-1 empty vector (black bars), pRSFDuet-1/*rutR* (red bars), or pRSFDuet-1/*rutRK52*$_{amber}$ (blue bars) as indicated. Significance (*: p ≤ 0.05; **: p ≤ 0.01; ***: p ≤ 0.001) level was tested between ΔCt values of the indicated samples and ΔCt values of the corresponding 16S-rRNA control using t-tests. Experiments were performed in three biologically independent experiments (n = 3). Source Data are provided as Source Data file.

K52-acetylation completely abolishes binding of RutR towards both box$_{carA}$ and box$_{rutA}$ dsDNA, the effect does not depend on the protein level and in fact would be even larger when RutR is acetylated at the endogenous protein level, which is much lower compared to ectopically expressed RutR protein (Supplementary Fig. 8a).

Next, we analyzed the impact of RutR acetylation on the expression of known target genes by qRT-PCR (quantitative reverse transcriptase PCR) (Fig. 4d). To this end, we analyzed the impact of RutR K52-acetylation on the expression of the following reported RutR target genes: *rutA* (as representative for the *rutAG*-operon), *carA* (for *carAB*-operon), *glxR* (for *gcl-hyi-glxR*-operon) and *gadC* (for *gadBC*-operon). Both, the acetylation of RutR at K52 and the absence of RutR protein in the *E. coli* U65 Δ*rutR* P$_{carA}$-*lacZ* empty vector control do not affect the expression of *rutA* (Fig. 4d). This suggests that *rutA* expression is low in the exponential growth phase even under conditions of absence of the transcriptional repressor RutR indicating that *rutAB* expression is repressed by mechanisms independent from RutR. This is similar also for the expression of *glxR* and *gadC* (Fig. 4d). For *rutR* expression we observed a high mRNA level for both, the RutR WT and the RutR AcK52 expressing cells. This is due to the ectopic expression of *rutR* that is detected by qRT-PCR. Intriguingly, for both, the transcription of *carA* and the P$_{carA}$-*lacZ* reporter, we observed that acetylation of RutR at K52 results in a statistically significant repression of expression compared to non-acetylated RutR (Fig. 4d). This is due to loss of RutR activator function on *carAB* expression upon RutR K52-acetylation. For expression of the *carAB* operon it is known that RutR indirectly induces the expression by abolishing the binding of the repressor PepA to the *carAB* promoter as RutR and PepA use partially overlapping binding sites on the *carAB*-promoter (Fig. 1a). These data show that K52-acetylation of RutR is a mechanism to control gene expression in vivo by reducing RutR dependent de-repression of *carAB* expression.

## Structure of K52-acetylated RutR shows that acetylation uses a steric and electrostatic mechanism to abolish DNA-binding

To unravel how acetylation of RutR at K52 mechanistically abolishes binding towards DNA, we solved the crystal structure of K52-acetylated RutR (RutR AcK52) at 2.25 Å resolution by X-ray crystallography (Fig. 5; Supplementary Table 4; Supplementary Fig. 9). RutR AcK52 protein was prepared by using the genetic code expansion concept as described above. RutR AcK52 crystallized in a complex with uracil that was co-purified, which shows that acetylation at K52 does not interfere with uracil binding (Fig. 5a, upper closeup; Supplementary Fig. 9). The complex RutR AcK52•uracil crystallized in the orthorhombic space group P2$_1$2$_1$2$_1$ with one RutR AcK52•uracil dimer per asymmetric unit. The overall conformation of the complex is almost unaltered with an r.m.s. deviation (r.m.s.d.) value of 0.62 Å for C-alpha atoms compared to the structure of non-acetylated RutR reported earlier (PDB: 4JYK). RutR AcK52 is composed of an N-terminal DNA-binding domain encompassing the two α-helical HTH-motif (α2-α3). The helices α2 and α3 are arranged nearly perpendicular to each other (Fig. 5a, b). It was shown in crystal structures of TetR-family members in complex with DNA, that residues of α3 contact the bases in the major groove of the DNA creating sequence specificity[16,83]. K52 in RutR is located at the N-terminus of α3 and would contact a guanine base in the *carAB*-box as suggested by comparison with the structure of the highly similar TetR-family transcriptional regulator QacR in complex with the IR1 operator DNA (Fig. 5a, lower closeup; Fig. 5b)[20]. In order to deduce the mechanism how AcK52 abolishes DNA binding, we superposed the N-terminal DNA-binding domain including the HTH-motif from the structure solved here (aa19-66; chain A) onto the corresponding region of QacR (aa4-50)[16,20,83,84] (Figs. 1b and 5b). These regions show a high degree of structural similarity represented by an r.m.s.d. value for the C-alpha atoms of 0.83 Å. In QacR•IR1, the homologous K36 is in

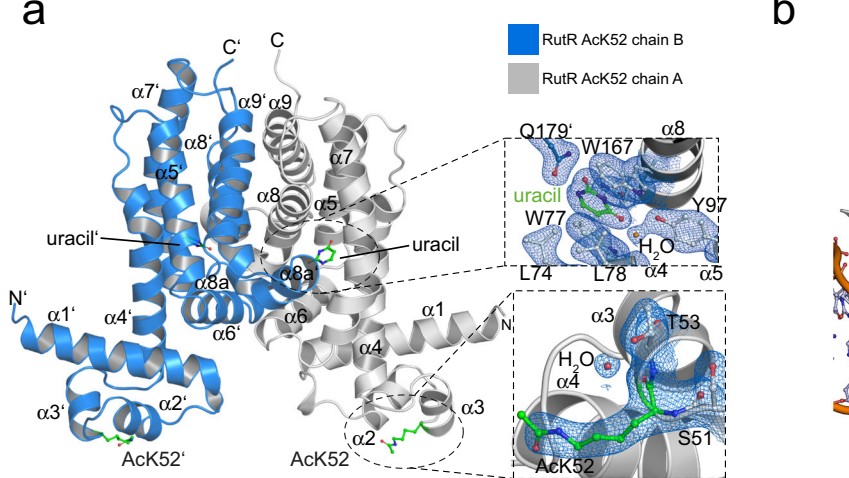

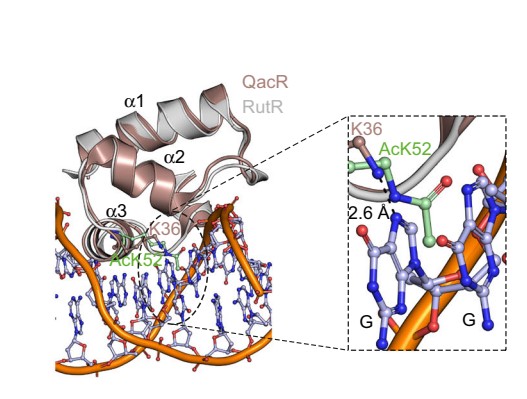

**Fig. 5 | Crystal structure of the RutR AcK52•uracil complex (PDB ID: 6Z1B).** **a** Overview of the RutR AcK52•uracil structure in cartoon representation. RutR is an all-helical protein consisting of an N-terminal domain encompassing helices α1-α3 containing the HTH-motif (α2-α3) needed for DNA-binding and a C-terminal ligand-binding domain (LBD), encompassing α-helices α4-α10. A short α-helix, α9, connects α8 and α10, as described for the non-acetylated structure (PDB: 4JYK). RutR forms a dimer with one uracil-molecule bound to the LBD in each monomer. Grey: chain A; blue: chain B (dark color: DNA-binding domain, light color: ligand-binding domain). Acetyl-L-lysine 52 of both chains is shown as stick-representation in green. Upper closeup: The uracil binding site in RutR AcK52. Uracil (green) forms hydrogen bonds with side chains of Q171 (below W167) and a bridging water molecule (red sphere) bound to Y97 of chain A (CA; grey). W167 and W77 form stacking interactions with uracil on both sides of the pyrimidine ring. Uracil is in hydrogen contact distance to Q179' of the other monomer (chain B, CB: blue). The K52-acetylation does not directly interfere with uracil binding. Shown in blue is the 2F$_o$-

F$_c$ electron density map contoured at 1σ. Lower closeup: of the 2F$_o$-F$_c$ electron density in blue contoured at 1σ obtained for the acetyl-L-lysine (AcK52) and the neighboring residues S51 and T53 from chain A (CA) of the RutR AcK52•uracil structure presented here (PDB: 6Z1B). **b** K52-acetylation in RutR abolishes DNA binding exerting a steric and electrostatic mechanism. Upper panel: Superposition of the HTH-motif from the structure of the TetR-repressor QacR in complex with IR1 operator DNA (PDB: 1JT0) and RutR AcK52•uracil (PDB: 6Z1B)[84]. Both HTH-regions (QacR: aa4-50; RutR aa19-66) superpose well with r.m.s.d. value of 0.83 for C-alpha atoms. closeup: AcK52 (green) electrostatically and sterically affects dsDNA binding. The N-(ε)-amino group of K36 in QacR is in interaction distance to N7 of a guanine base (2.6 Å; red dashed line) creating sequence specificity. This interaction is abolished by K52-acetylation in RutR. A model of RutR in compex with the *carAB* promoter suggests that K52 interacts with the guanine base at analogous position[20]. dark brown: HTH-motif from QacR; bright brown: HTH-motif from RutR.

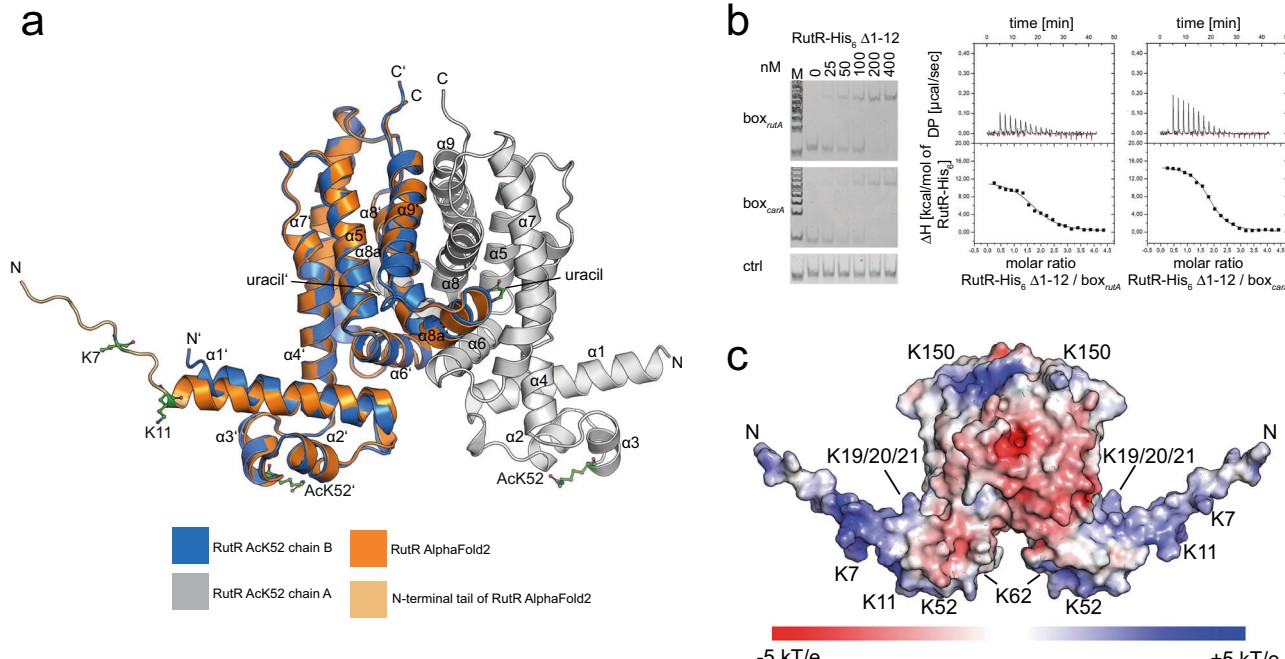

**Fig. 6 | RutR contains an unstructured histone-like N-terminal tail contributing to dsDNA-binding. a.** Localization of K7 and K11 in the histone-like N-terminal tail of RutR in the AlphaFold2 model[85,86]. The AlphaFold2 model was superimposed with chain B of the structure of K52-acetylated RutR. The structures are highly similar with an overall r.m.s.d. of 1.15 Å. The N-terminal tail is highly flexible supported by the finding that no electron density was obtained for the N-terminal 12 (chain A) or 11 (chain B) residues in the RutR AcK52 structure solved here (PDB: 6Z1B). The AlphaFold2 model supports that these N-terminal residues are unstructured represented by the low model confidence score (pLDDT < 50). **b** Deletion of the unstructured histone-like N-terminal tail in RutR Δ1-12 impairs dsDNA-binding. Left panel: RutR Δ1-12 binds to box$_{rutA}$ and box$_{carA}$ dsDNA as shown by EMSAs. box$_{rutA}$, box$_{carA}$, and control dsDNA fragments are incubated with increasing concentrations of RutR proteins as indicated. Experiments were performed in duplicates. Right panels: RutR Δ1-12 impairs binding towards both, box$_{rutA}$ and box$_{carA}$, dsDNA as shown by ITC (DP: differential power); box$_{rutA}$: left

diagram; box$_{carA}$: right diagram. All interactions were conducted in three biologically independent experiments ($n = 3$) and the values in the table are given as means ± standard deviations (Supplementary Table 3). Source data are provided as Source Data files. **c** Electrostatic surface representation of the AlphaFold2 model of RutR. The positions of acetylated lysines in RutR are indicated. The unstructured N-terminal tails encompassing the α-N-terminus, K7, and K11 are positively charged as visible by the blue color. Acetylation of the α-amino groups and of the ε-amino groups of K7 and K11 would neutralize this positive charge affecting electrostatic steering of RutR to box$_{rutA}$ and box$_{carA}$ dsDNA. The basic patch consisting of 19-KKK-21 in α1 is shown and K52 in the HTH of the DNA-binding domain and K62 in the N-terminus of α4 are also contributing to the positive surface potential in this region of RutR. K150 is located at the top of the LBD. The electrostatic potential was calculated by the APBS plugin in PyMOL (with k: Boltzmann constant, T: temperature, e: unit charge). The figure was prepared by PyMOL[144].

hydrogen bond distance of 2.6 Å with N7 of a guanine base (Fig. 5b, closeup). This contributes to sequence specificity in binding of QacR to DNA[20]. The superposition suggests that lysine acetylation at K52 in RutR exerts an electrostatic as well as steric mechanism to abolish DNA-binding supporting our data shown above. More precisely, K52-acetylation would abrogate the hydrogen bond towards N7 of the guanine base and it is incompatible with DNA-binding sterically clashing with DNA-bases (Fig. 5b, closeup). Notably, we did not observe electron density for the N-terminal residues preceding the RutR DNA-binding domain, suggesting that this part is flexible. This is supported by the recently reported AlphaFold2 model of RutR that shows very low per-residue confidence score for most of this region (< 50 pLDDT), where values of <50 pLDDT are indicative for regions being unstructured in isolation (Fig. 6a)[85,86]. To assess whether this flexible region contributes to DNA-binding, we produced the RutR protein lacking the first twelve N-terminal residues, RutR Δ1-12. For other TetR-family members it is reported that the N-terminal region preceding the HTH-motif contributes to DNA-binding[16]. We observed binding of RutR Δ1-12 towards box$_{rutA}$ and box$_{carA}$ dsDNA in EMSA (Fig. 6b, left panel). However, thermodynamic characterization of the interactions by ITC showed that the binding of RutR Δ1-12 towards box$_{rutA}$ (742 nM) and box$_{carA}$ (317 nM) dsDNA was approximately 17- and 11-fold reduced compared to RutR WT (box$_{rutA}$: 44 nM; box$_{carA}$: 28 nM). This shows that these N-terminal residues contribute to DNA-binding (Fig. 6b, right panels; Supplementary Table 3). Analyzes of the primary sequence of

this N-terminal tail reveals two conserved lysine side chains, namely K7 and K11 (Fig. 6c; Supplementary Figs. 10, 11; Supplementary Data 1). For many TetR-family members, a positively-charged flexible N-terminal region was shown to precede the HTH-motif[16]. This suggests that this unstructured N-terminal tail in RutR and other TetR-family members might play a functionally similar role in mediating DNA-binding as observed for the N-terminal tail in eukaryotic histones. In contrast to K52-acetylation of RutR, which results in clashes with DNA bases, histone-DNA-binding was shown to be mediated via electrostatic interactions of lysines with the negatively-charged sugar-phosphate backbone.

## RutR is enzymatically acetylated at K7 and K11 in the RutR flexible N-terminal tail

Until recently, the Gcn5-related N-terminal acetyltransferase (GNAT)-family member PatZ/YfiQ was the solely identified KAT in *E. coli*[57]. A recent study reported the presence of four additional KATs in *E. coli*: RimI, YjaB, YiaC, and PhnO[63]. RimI and YjaB were structurally characterized by X-ray crystallography and NMR, respectively[64,65]. We recombinantly expressed and purified all five known *E. coli* KATs and tested these for acetylation activity towards RutR. We incubated non-acetylated RutR protein with the KAT enzyme in vitro and analyzed RutR lysine acetylation by immunoblotting using an anti-acetyl-lysine antibody (Fig. 7). As controls, we also prepared the catalytically inactive mutant enzymes and performed reactions in absence of the acetyl

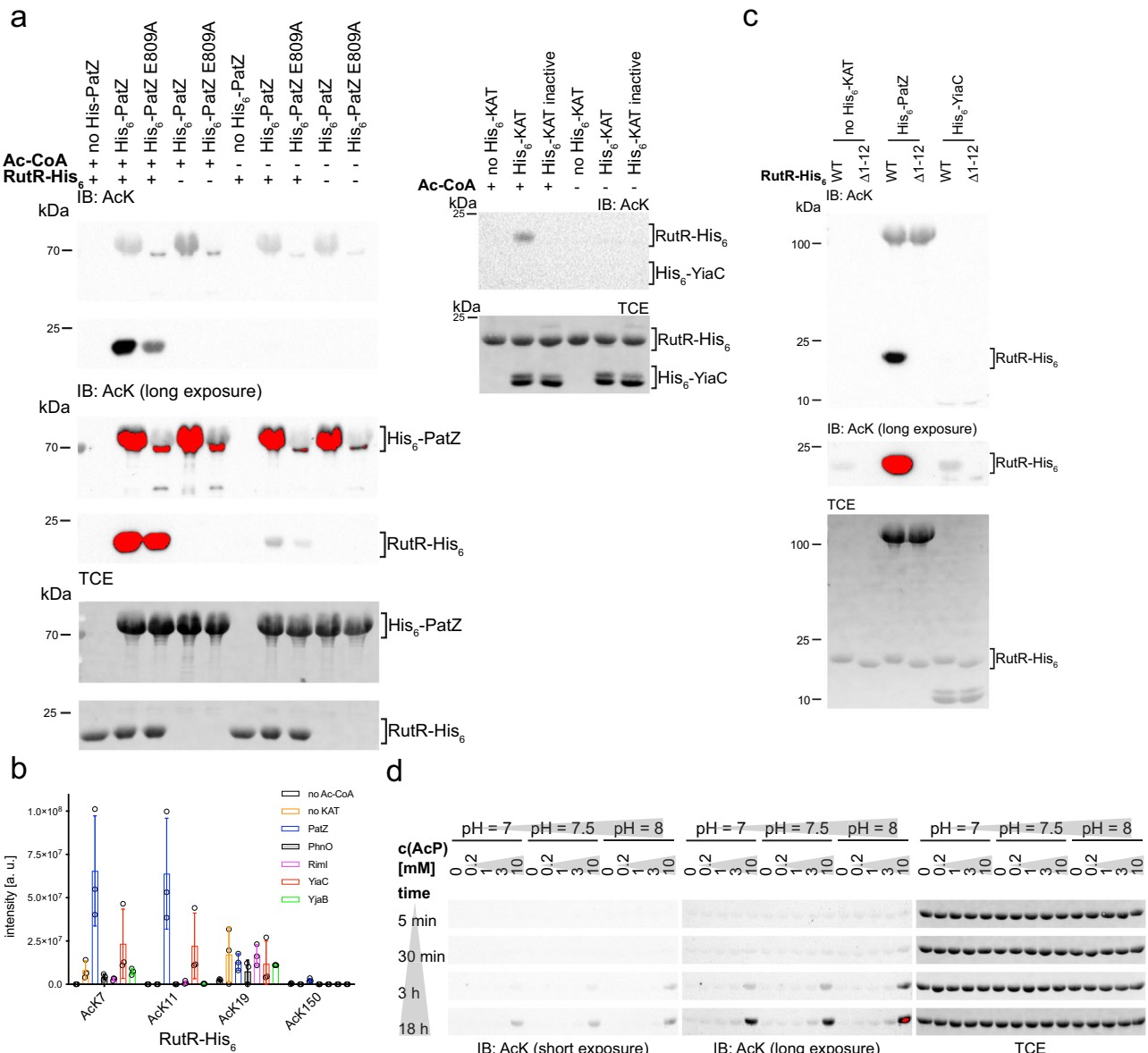

**Fig. 7 | RutR is enzymatically acetylated by PatZ/YfiQ and YiaC in the histone-like N-terminus and non-enzymatically by acetyl-phosphate. a** Identification of PatZ/YfiQ and YiaC as RutR KATs. Purified RutR was incubated with the KAT enzymes (active/inactive) in presence/absence of acetyl-coenzyme A (Ac-CoA). The samples were analyzed by immunoblotting with anti-AcK AB (IB: AcK). Total protein staining using 2,2,2-trichloroethanol (TCE) was used as loading control. All KATs, except YiaC show an autoacetyltransferase activity (Supplementary Fig. 5a). The KATs PatZ/YfiQ (left panels) and YiaC (right panels) lysine-acetylate RutR. Inactive KATs: PatZ/YfiQ E909A, YiaC Y115A. The results were confirmed in at least three replicates ($n \geq 3$). Long exposure served to visualize weak signals (red: over-saturation of signal). Source data are provided as Source Data files. **b** K7 and K11 in the RutR N-terminal are acetylated enzymatically by KATs PatZ/YfiQ and YiaC as shown by LC-MS/MS. Both can simultaneously acetylate RutR at K7 and K11. Shown are the calculated overall intensities summed up from single peptides. Presented are the means ± standard deviations from three biologically independent experiments ($n = 3$). a.u.: arbitrary units. black: no acetyl-CoA; orange: no KAT; blue: PatZ/ YfiQ; black/grey-shaded: PhnO; pink: RimI; red: YiaC; green: YjaB (Supplementary Data 1). Source data are provided as Source Data files. **c** PatZ/YfiQ and YiaC act as KATs acetylating RutR exclusively in the N-terminal tail. RutR Δ1-12 is not acetylated by PatZ/YfiQ or YiaC (long exposure) suggesting that the acetylation sites reside in the N-terminal region in RutR. Immunoblotting (IB) was conducted with anti-AcK-AB and total protein staining using TCE served as loading control. One exemplary result is shown from three independent replicates ($n = 3$). Red indicates over-saturation of signal. Source data are provided as Source Data files. **d** RutR is non-enzymatically lysine-acetylated by acetyl-phosphate (AcP) conducted at different pH-values and time ranges. All samples were analyzed in immunoblots (IB) stained with anti-AcK AB. TCE staining served as loading control. LC-MS/MS analyzes shows that RutR is acetylated non-enzymatically by AcP at K7, K11, the triple K-motif (19-KKK-21), K52, K95 and K150 (Supplementary Data 2; Supplementary Table 5; Supplementary Fig. 6). One exemplary result out of two replicates is shown ($n = 2$). Red indicates oversaturation of signal. Source data are provided as Source Data files.

group donor acetyl-CoA (Ac-CoA). Our studies revealed that, with exception of YiaC, all KATs show auto-acetyltransferase activity (Fig. 7a; Supplementary Fig. 12a). PatZ/YfiQ and YiaC were active in acetylating RutR, while RimI, YjaB and PhnO were not (Fig. 7a; Supplementary Fig. 12a). Using RutR K52Q as a substrate for PatZ/YfiQ and YiaC shows no reduced acetylation level in immunoblottings

compared to RutR WT. This observation signifies that either K52 is not acetylated by PatZ/YfiQ or YiaC or it is acetylated at a low stoichiometry or that other sites are stronger epitopes for the antibody and therefore are better detected (Supplementary Fig. 12b). Previously, we observed that the antibody shows strong bias in the recognition of different acetylation sites[82]. However, we later clearly show that indeed

K52 is not enzymatically acetylated but only acetylated non-enzymatically by acetyl-phosphate. In order to identify the specific sites targeted by PatZ/YfiQ and YiaC, we performed mass-spectrometry on enzymatically acetylated RutR samples (Fig. 7b). These data showed that K7 and K11 in the histone-like unstructured N-terminal tail of RutR are the major acetyl group acceptor sites in PatZ/YfiQ and YiaC enzymatically acetylated RutR (Fig. 7b; Supplementary Data 2). We also observed acetylation at K19. However, this is part of a basic patch (19-KKK-21) known to be prone to non-enzymatic acetylation due to the decrease of the side chains' $pK_A$ value[69]. As a support, we observed similar intensities for the control lacking KAT, suggesting that this patch is indeed prone to non-enzymatic acetylation. Notably, PatZ/YfiQ and YiaC are not only capable to catalyze simultaneous acetylation of K7 and K11, but they also both act as RutR N-terminal acetyltransferase (NAT) acetylating the N-($\alpha$)-amino group of RutR (Supplementary Data 2). To confirm these results and to rule out that other sites are acetylated enzymatically but just not detectable by LC-MS/MS, we analyzed the N-terminally deleted RutR protein (RutR $\Delta$1-12) and used this as a substrate for PatZ/YfiQ and YiaC catalyzed acetylation (Fig. 7c). These results confirm that K7 and K11 in the N-terminal tail of RutR are the acetyl group acceptor sites as the N-terminally truncated RutR is acetylated by neither PatZ/YfiQ nor YiaC to an extent that is detectable by immunoblotting (Fig. 7c). We thus identified K7 and K11 in the N-terminal tail of RutR as bona fide substrates for PatZ/YfiQ and YiaC. The strong signal we observed in immunoblotting particularly for K7 and K11 acetylation by PatZ/YfiQ suggests that these are accumulating to high stoichiometries. To clarify if RutR is also targeted by non-enzymatic acetylation we next performed studies to assess RutR acetylation by acetyl-phosphate, the major molecule responsible for non-enzymatic acetylation in bacteria (Fig. 1d).

### Acetyl-phosphate drives non-enzymatic acetylation of RutR

After identifying the lysines that are targeted by enzymatic acetylation catalyzed by the KATs PatZ/YfiQ and YiaC, we next asked if RutR is also acetylated non-enzymatically by the high-energy molecule acetyl-phosphate. It was shown before that non-enzymatic acetylation by acetyl-phosphate can even occur in a site-specific manner[68,69,87,88]. To inspect whether RutR is also a target of acetylation by acetyl-phosphate, we incubated recombinantly expressed and purified RutR WT protein with acetyl-phosphate in vitro (Fig. 7d). To show the sensitivity of RutR WT for acetylation driven by acetyl-phosphate and to evaluate if this can be of physiological significance, we first assessed the conditions under which RutR acetylation by acetyl-phosphate occurs. To this end, we varied the pH-value (pH: 7, 7.5, 8), the incubation time (5 min, 30 min, 3 h, 18 h) and the acetyl-phosphate concentration (0/0.2/1/3/10 mM). As a readout, we used immunoblotting with an anti-acetyl-lysine antibody. We observed that acetyl-phosphate results in stronger acetylation of RutR compared to the control lacking acetyl-phosphate at concentrations of 3-10 mM acetyl-phosphate after 30 min of incubation at pH 7.5 and pH 8 (Fig. 7d). Differences become more pronounced at longer incubation times, higher acetyl-phosphate concentrations and a more basic pH. Notably, this does not rule out that RutR is acetylated by acetyl-phosphate at lower concentrations. The detection limit of this assay is determined by the sensitivity of the immunoblotting readout and the recognition of the individual sites by the antibody. To narrow down which acetylation sites are major targets in RutR by non-enzymatic acetylation, we selected conditions for analyzes by mass-spectrometry (pH 7.5, incubation time: 3 and 18 h, 10 mM acetyl-phosphate). Our data revealed acetyl-phosphate-mediated acetylation above the control level for K7 and K11 in the RutR N-terminal tail, K19, K20 and K21 in a basic patch (19-KKK-21) in $\alpha$1, K52 in the HTH-motif important for DNA-

binding and K95 as well as K150 within the LBD (Supplementary Fig. 13; Supplementary Data 3; Supplementary Table 5). However, this does not exclude that other lysines in RutR are also targeted by non-enzymatic acetylation but were not accessible for detection by mass spectrometry. This means due to the generation of very small or very large peptides or non-sufficient ionization efficiencies some peptides might not be suitable for LC-MS/MS analyzes. We identified K7 and K11 in the RutR unstructured N-terminal tail as acetyl group acceptor sites for enzymatic and non-enzymatic acetylation. We next investigated how acetylation at K7 and/or K11 affects RutR function.

### Acetylation in the RutR N-terminal tail affects DNA-binding affinity and the mechanism of binding

We prepared RutR AcK7, RutR AcK11, and double-acetylated RutR AcK7/11 proteins applying the genetic code expansion concept as described above. To increase the yield for the double-acetylated RutR AcK7/11 protein, we included a plasmid (pTech-*ackRS*-tRNA$^{Pyl-opt}$) encoding for an optimized amber suppressor tRNA, *Mbt*RNA$_{opt}$ for the expression[89–92]. The proteins behaved similar, i.e. eluting as dimer, in analytical size-exclusion chromatography experiments as non-acetylated RutR showing that the acetylation in the N-terminal tail does neither interfere with protein folding nor with oligomerization (Supplementary Fig. 1a). To assess the role of these acetylation sites for RutR function, we performed semi-quantitative EMSAs to analyze the effect on binding to box$_{rutA}$ and box$_{carA}$ dsDNA (Fig. 8a; Supplementary Fig. 4). Both single-acetylated RutR AcK7 and AcK11 bind to both dsDNA fragments similarly to non-acetylated RutR. However, EMSAs suggest reduced binding of double-acetylated RutR AcK7/11 to both dsDNA fragments (Fig. 8a; Supplementary Fig. 7).

To analyze whether acetylation in the RutR N-terminal tail affects the binding mechanism for the interactions with the dsDNA fragments and to quantify the affinities, we performed ITC experiments (Fig. 8b; Supplementary Table 3). These data revealed that acetylation at the individual sites, K7 and K11, resulted in a two- to threefold decrease in binding affinity to both box$_{rutA}$ and box$_{carA}$ dsDNA (box$_{rutA}$ AcK7: 94 nM, AcK11: 95 nM; box$_{carA}$: AcK7: 73 nM, AcK11: 82 nM) compared to non-acetylated RutR (box$_{rutA}$: 44 nM, box$_{carA}$: 28 nM; Fig. 8b; Supplementary Table 3). However, the binding mechanism is not affected by single acetylation at either K7 or K11. As observed for non-acetylated RutR, the endothermic reactions, driven exclusively by the change in the reaction entropy, show a molar ratio of 2:1 (Fig. 8b; Supplementary Table 3). We also prepared K7- and K11-double-acetylated RutR as we identified simultaneous acetylation of both sites enzymatically by PatZ/YfiQ and YiaC. Supporting the EMSA results, quantitative thermodynamic analyzes by ITC revealed for the double-acetylated RutR AcK7/11 a sixfold reduced binding affinity to box$_{rutA}$ dsDNA and a 11-fold reduced binding to box$_{carA}$ dsDNA compared to non-acetylated RutR (box$_{rutA}$: WT 44 nM, AcK7/11 251 nM; box$_{carA}$: WT 28 nM, AcK7/11 300 nM) (Fig. 8b; Supplementary Table 3). That means acetylation of RutR at K7 and K11 reflects a similar dsDNA-binding affinity reducing effect as seen for the N-terminal deletion mutant RutR $\Delta$1-12 ($K_D$: box$_{rutA}$ 741 nM; box$_{carA}$ 317 nM). However, in terms of the thermodynamic profile, for which RutR $\Delta$1-12 behaves as non-acetylated full-length RutR WT, the double acetylation of RutR at K7 and K11 affects the binding mechanism to the box$_{rutA}$ and box$_{carA}$ dsDNA. RutR AcK7/11 shows a decreased overall change of favorable reaction entropy, which is compensated by a decrease in unfavorable change in the reaction enthalpy for the interactions with both DNA-fragments compared to non-acetylated RutR WT (Fig. 8b; Supplementary Table 3). The ITC data suggest additivity of the changes in the free energies ($\Delta\Delta G$) for RutR AcK7 and AcK11 dsDNA-binding, i.e. they independently affect binding to both, box$_{rutA}$ and box$_{carA}$ dsDNA (box$_{rutA}$: $\Delta\Delta G$ AcK7 + $\Delta\Delta G$ AcK11: 1.93 kJmol$^{-1}$ + 2.11 kJmol$^{-1}$ = 4.04 kJmol$^{-1}$ versus $\Delta\Delta G$

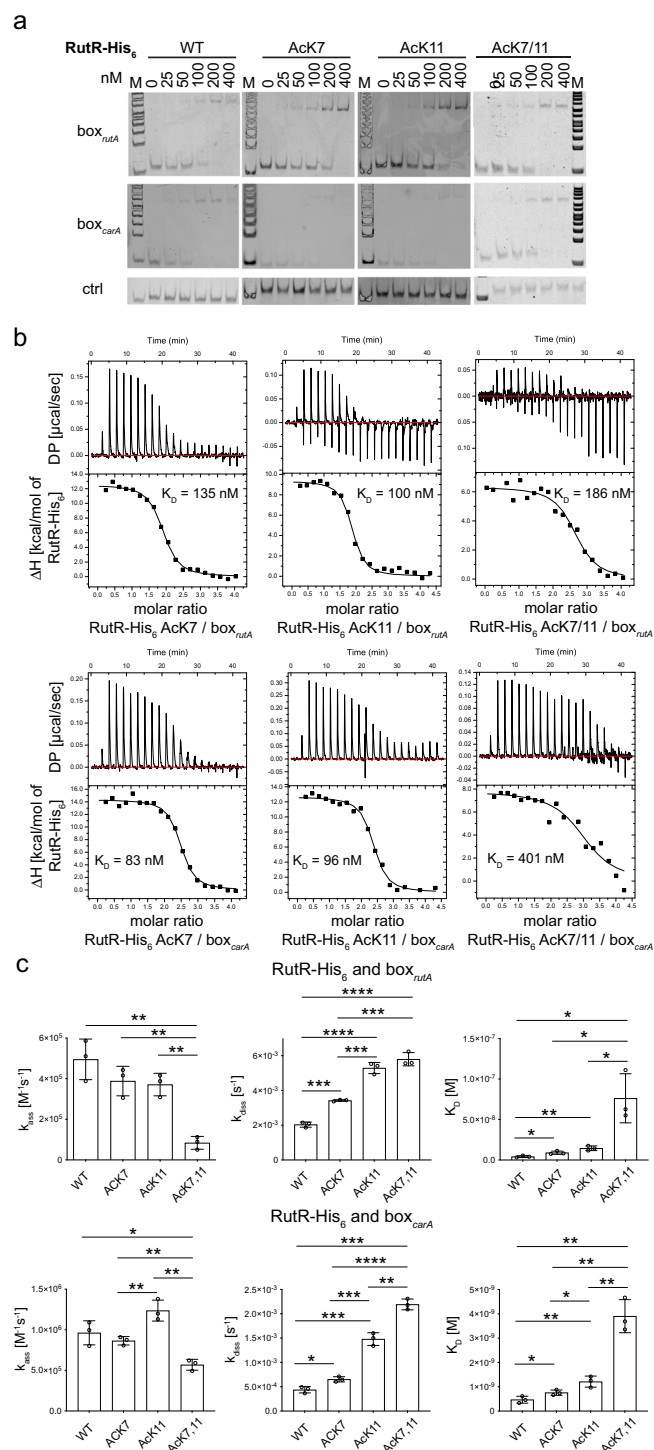

**Fig. 8 | Acetylation of RutR at K7, K11 and K7/11 affects binding to dsDNA thermodynamically and kinetically as determined by EMSAs, ITC and surface plasmon resonance (SPR). a** EMSAs show that RutR AcK7, AcK11, and AcK7/11 bind to both dsDNA fragments as indicated (box$_{rutA}$: upper panels; box$_{carA}$: middle panels; lower panels: control dsDNA). EMSAs indicate that double-acetylated RutR AcK7/11 impairs binding towards dsDNA fragments more pronounced than the single acetylated proteins. EMSAs were conducted as described above. Source data are provided as Source Data file. **b** Acetylation of RutR in the N-terminal domain impairs binding towards box$_{rutA}$ (upper panels) or box$_{carA}$ (lower panels) dsDNA as shown by ITC. All reactions show a molar ratio of 2:1 showing that one RutR dimer binds to one dsDNA molecule. The reactions are endothermically driven exclusively by the change in the reaction entropy. Upon acetylation the unfavorable change in the enthalpy is decreased, which is compensated by a decrease in favorable change in reaction entropy. The interaction of double-acetylated RutR towards box$_{rutA}$ is fivefold reduced (box$_{rutA}$: WT 44 nM, AcK7/11 239 nM; box$_{carA}$: WT 28 nM, AcK7/11 300 nM) and towards box$_{carA}$ 11-fold reduced compared to non-acetylated RutR. All ITCs were conducted in three biologically independent experiments ($n = 3$). Source data are provided as Source Data file. **c** Acetylation of RutR in the N-terminal domain decreases the association rate constants and increases the dissociation rate constants towards box$_{rutA}$ (upper panels) or box$_{carA}$ (lower panels) dsDNA as shown by surface plasmon resonance (SPR) measurements. The association rate constants ($k_{ass}$) (left panels), dissociation rate constants ($k_{diss}$) (middle panels) and the derived dissociation equilibrium constants ($K_D$) were plottet as bar graphs. All measurements were done in three biologically independent experiments ($n = 3$). The bars represent the means ± standard deviations. Source data are provided as Source Data file.

important for binding, steering and molecular recognition of interactions to allow a productive association of proteins or biomolecules[93–95]. It was shown earlier for other DNA-binding proteins such as histones that electrostatic interactions between lysines and the negatively charged DNA sugar-phosphate backbone contributes to DNA-binding. To analyze the impact of RutR K7- and K11-acetylation and K7/K11 double-acetylation on the dynamics of the interaction with box$_{rutA}$ and box$_{carA}$ dsDNA, we performed surface plasmon resonance (SPR) experiments (Fig. 8c; Supplementary Fig. 14a). To this end, we prepared biotin-labeled dsDNA box$_{rutA}$ and box$_{carA}$ oligonucleotides that coupled to an SPR sensor chip via an immobilized streptavidin tag. Subsequently, RutR variants (RutR WT, AcK7, AcK11, AcK7/11) were injected. Association rate constants, $k_{ass}$, were determined from the first five injections performed with increasing RutR concentrations (6.25/12.5/25/50/200 nM RutR). The dissociation rate constants, $k_{diss}$, were derived from the final dissociation phase (Fig. 8c; Supplementary Fig. 14a). Overall, the acetylation of RutR at K7 and K11 resulted in an increase of the dissociation rate constant, i.e. a decrease in RutR dsDNA-binding affinity, compared to non-acetylated RutR. This effect is largest for the double-acetylated RutR AcK7/11, suggesting that neutralization of the positive charges at the lysine side chains in the N-terminal tail enables faster dissociation from the DNA (Supplementary Table 7). Along that line, RutR AcK7/11 reduces association kinetics towards both box$_{rutA}$ and box$_{carA}$ dsDNA suggesting that acetylation of K7 and K11 in the N-terminal tail affects electrostatic steering of RutR to dsDNA (Fig. 8c; Supplementary Table 7). Both effects, an increase in the dissociation rate constant and decrease in the association rate constant, result in an overall decrease in the binding affinity, which is almost 8-fold reduced for interaction of RutR AcK7/11 towards box$_{carA}$ dsDNA and 18-fold reduced for the binding towards box$_{rutA}$ dsDNA. For the interaction of RutR AcK7/11 and box$_{carA}$ this reduction in the affinity is mainly due to the approximately fivefold increase in the dissociation rate constant (WT: $4.38*10^{-4}\,s^{-1}$; AcK7/11: $2.19*10^{-3}\,s^{-1}$), while for the interaction with box$_{rutA}$ this is mainly due to the approximately sixfold decrease in the association rate constant (WT: $4.95*10^5\,M^{-1}s^{-1}$; AcK7/11: $8.42*10^4\,M^{-1}s^{-1}$). These data confirm the tendency to reduce dsDNA-binding affinity upon RutR acetylation at K7 and/or K11. The analysis of the differences in the change in Gibbs free energy ($\Delta\Delta G$) shows that under these experimental conditions the effect of

AcK7/11: 4.33 kJmol$^{-1}$; box$_{carA}$: $\Delta\Delta G$ AcK7 + $\Delta\Delta G$ AcK11: 2.26 kJmol$^{-1}$ + 2.58 kJmol$^{-1}$ = 4.84 kJmol$^{-1}$ versus $\Delta\Delta G$ AcK7/11: 5.58 kJmol$^{-1}$) (Supplementary Table 6). After showing that acetylation of K7 and K11 within the N-terminal tail of RutR affects dsDNA-binding affinity and the DNA-binding mechanism, we studied the impact on kinetics of the interactions to further elucidate the underlying mechanism.

**K7- and K11-acetylation in the RutR N-terminal tail affects the interaction kinetics with dsDNA**

As acetylation of lysines results in neutralization of the positive charge this might affect the kinetics of association and dissociation of RutR to the dsDNA. Electrostatic interactions act on longer ranges and are

double-acetylated RutR AcK7/11 on dsDNA-binding is more than additive compared to single lysine acetylations in RutR at K7 or K11 (Supplementary Table 6). This suggests that under these conditions the double-acetylation at K7 and K11 results in an additional structural effect upon dsDNA-binding that is not observed when RutR is acetylated at the individual sites. Our data show that the N-terminal tail in RutR contributes to dsDNA-binding by interfering with the association and dissociation kinetics.

### Lysine acetylation at K7 and K11 in the RutR N-terminal tail impairs RutR DNA-binding in vivo

Our data on RutR acetylation shows that K7 and K11 within the RutR N-terminal tail impairs binding towards both, box$_{rutA}$ dsDNA and box$_{carA}$ dsDNA. As our data suggest other unknown mechanisms might exist, which are needed to initiate expression from *rutA* promoter apart from removing the RutR transcription repressor, we used the strain *E. coli* U65 with genomic deletion of *rutR* and insertion of P$_{carA}$-*lacZ* (*E. coli* U65 Δ*rutR* P$_{carA}$-*lacZ*) to assess whether RutR acetylation at K7 and/or K11 affects RutR transcriptional regulator activity in vivo. We initiated studies using the genetic code expansion concept to incorporate AcK at positions K7 and K11 in RutR to assess the impact of acetyl-lysine in vivo as described for RutR AcK52. However, we were not able to titer the system to obtain similar protein levels for RutR AcK7, RutR AcK11, RutR AcK7/11 and non-acetylated RutR WT. In fact, we obtained no protein for the N-terminally acetylated RutR proteins upon expression of *rutRK7$_{amber}$*, *rutRK11$_{amber}$* or *rutR7/K11$_{amber}$* in *E. coli* U65 Δ*rutR* P$_{carA}$-*lacZ*. This made an evaluation of the data impossible. To tackle this issue, we used the RutR K7Q, RutR K11Q and RutR K7Q/11Q mutants and the corresponding R-mutants to analyze the impact on RutR transcriptional regulator activity using the β-galactosidase reporter activity as a readout. The N-terminally

mutated RutR-proteins followed the same trend observed also for the N-terminally acetylated RutR-protein. While we obtained protein for the N-terminal K to Q and K to R single mutants (group 2, Fig. 9, left panel, red bars) the protein level was reproducibly lower compared to RutR WT and RutR K52Q (group 1, Fig. 9, left panel, black bars). For the double-mutants RutR K7Q/K11Q and K7R/K11R (group 3, Fig. 9, left panel, blue bars) the protein level was even more reduced compared to the single mutated RutR. Notably, for the N-terminal deletion mutant RutR Δ1-12 we could not detect any protein upon expression in *E. coli* U65 Δ*rutR* P$_{carA}$-*lacZ* (Fig. 9). Although the underlying mechanism needs further investigation, this supports our hypothesis that the sequence encoding for the RutR N-terminus is important for translational efficiency. To analyze the data, we performed statistical comparisons only within these groups of similar expression but not between these groups as the RutR protein amount affects the β-galactosidase expression and as a consequence also the β-galactosidase activity. The comparisons within group 1 reveal that RutR K52Q and the empty vector control show a statistically significant reduction in β-galactosidase activity compared to non-acetylated RutR WT. Within group 2, i.e. RutR single mutants, no statistically relevant alterations in β-galactosidase expression was observed (Fig. 9, right panel). Importantly, within group 3 the acetylation mimetic double mutant RutR K7Q/11Q shows a statistically significant reduction in β-galactosidase activity compared to the RutR K7R/K11R charge-conserving mutant (Fig. 9, right panel). To validate if the conclusions drawn from the double mutants are transferable to the effects of the double-acetylated RutR AcK7/11 the interactions of the double mutants RutR K7Q/K11Q and K7R/K11R towards box$_{rutA}$ and box$_{carA}$ dsDNA we analyzed by ITC (Supplementary Table 3, Supplementary Fig. 7; Supplementary Fig. 14b). The double mutant RutR K7R/K11R binds to both DNA-fragments with a similar affinity compared to non-acetylated

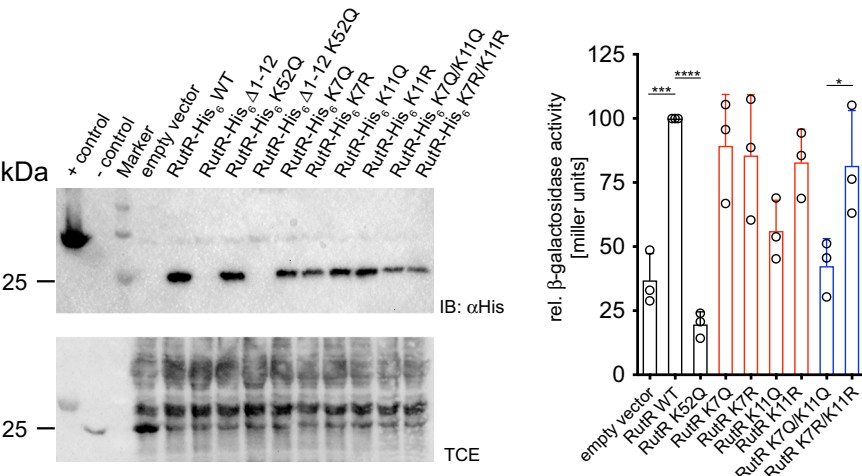

**Fig. 9 | Acetylation in the RutR N-terminus modulates its transcriptional regulator activity in vivo.** *E. coli* U65 Δ*rutR* with a genomic P$_{carA}$-*lacZ* fusion was transformed with pRSFDuet-1 empty vector, pRSFDuet-1/*rutR*, pRSFDuet-1/*rutR* K7Q, K7R, K11Q, K11R and the double mutants K7Q/K11Q and K7R/K11R as indicated. Cultures were harvested at OD$_{600}$ of app. 0.6 and analyzed in a *lacZ* reporter assay. Left panel: Cell lysates of the *E. coli* U65 cells expressing RutR wild-type and the mutants were analyzed by immunoblotting (IB) using an anti-His$_6$-antibody (IB: αHis). Neither RutR Δ1-12 nor RutR Δ1-12 K52Q were expressed. For the single mutants (red bars) and the double mutants (blue bars) of the N-terminal lysines we observed a reduced expression even when normalizing for the protein loading and/or different optical densities of cultures. To this end, we perform statistical analyzes only between samples of similar expression, i.e. single mutants to each other and double mutants to each other. Positive control for IB: Sirt1$_{225-664}$-His$_6$; negative control: CobB. TCE staining served as loading control. Right panel: transcriptional reporter β-galactosidase assays in the absence (empty vector) and presence of *rutR*

upon ectopic expression of RutR-WT and mutated RutR. As expected, RutR K52Q results in a statistically significant reduction in β-galactosidase activity compared to RutR WT and the empty vector control (black bars). Comparing the single mutants (red bars) shows no statistically relevant alterations in β-galactosidase activity. Comparison of double mutated K7Q/K11Q with K7R/K11R (blue bars), which show a similar expression level, shows that the double Q acetylation mimicking mutant shows a statistically significant reduction in β-galactosidase activity compared to the double R-mutant. All RutR K to R mutants did not impair β-galactosidase expression suggesting that at K7 and K11 electrostatic quenching is the major molecular mechanism by which acetylation impairs DNA-binding. Experiments were performed in three biologically independent experiments and bars depict means ± standard deviations of determined relative β-galactosidase activity in miller units ($n$ = 3). Statistical significance (*: $p ≤ 0.05$; ***: $p ≤ 0.001$; ****: $p ≤ 0.0001$) was tested using t-tests. Source data are provided as Source Data file.

RutR WT (box$_{rutA}$: WT 44 nM, K7R/K11R: 47 nM; box$_{carA}$: WT 28 nM, K7R/K11R: 17 nM) revealing that this double mutant is a reliable mimic for the non-acetylated state. The double mutant RutR K7Q/K11Q shows a similar trend reducing the affinity in binding towards box$_{rutA}$ and box$_{carA}$ dsDNA as the double acetylated RutR AcK7/11 (box$_{rutA}$: AcK7/11 251 nM, K7Q/K11Q: 540 nM; box$_{carA}$: AcK7/11 300 nM, K7Q/K11Q: 153 nM). Our data confirm that acetylation of RutR in the N-terminal tail results in a phenotype of physiological importance modulating RutR transcriptional regulator activity by impairing RutR DNA-binding.

## RutR acetylation at functionally important sites is reversed by CobB-mediated deacetylation

Next, we wondered whether the acetylation of RutR at the known acetylation sites can be reversed by enzymatic deacetylation. So far, the only reported deacetylase in *E coli* is the NAD$^+$-dependent sirtuin deacetylase CobB. We reported earlier that RutR AcK52 and RutR AcK62 are bona fide substrates for the bacterial NAD$^+$-dependent sirtuin deacetylase CobB, with RutR AcK52 being a slightly better substrate compared to RutR AcK62[37]. There are reports stating that the protein YcgC is a second deacetylase encoded in *E. coli*. However, we showed earlier that YcgC does not show RutR-lysine deacetylase activity[36,37]. Moreover, it is also not a hydrolase with protease activity[37]. Therefore, we tested whether CobB is able to deacetylate acetylated RutR variants in an NAD$^+$-dependent manner. Here, we confirm that RutR AcK52, which is situated within the helix-turn-helix (HTH)-motif, and to a lesser efficiency also RutR AcK62 are deacetylated in an NAD$^+$-dependent manner by CobB (Fig. 10a, left panels). Moreover, we identified RutR AcK21 in helix α1 preceding the HTH as an additional CobB substrate (Fig. 10a, left panels). The two acetylation sites in the flexible N-terminal tail preceding the RutR DNA-binding domain, i.e. AcK7 and AcK11, are most efficiently deacetylated by CobB. We observed furthermore that CobB catalyzed AcK7, AcK11, and AcK7/11 deacetylation is not inhibited by 10 mM of the non-competitive inhibitor nicotinamide (Supplementary Fig. 15a), which was sufficient for inhibition of all other RutR acetylation sites. Instead, 50 mM nicotinamide was nessecary to inhibit CobB catalyzed deacetylation at AcK7, AcK11, and AcK7/11 (Fig. 10a, right panels). This suggests that these sites in the RutR histone-like N-terminal tail, i.e. AcK7 and AcK11, and even the double-acetylated RutR AcK7/11 are highly efficiently deacetylated by CobB. To evaluate the efficiency of CobB catalyzed deacetylation, we performed kinetic time-course experiments with catalytic concentrations of CobB (100 nM) and 150-fold molar excess of acetylated RutR (15 μM) as substrates (Fig. 10b). These data show that under the assay conditions, the N-terminal sites RutR AcK7, AcK11 and AcK7/11, are more than 25% deacetylated already after 2 min, while AcK52 was only approximately 50% deacetylated after 120 min (Fig. 10b). To confirm that PatZ/YfiQ and YiaC-mediated acetylation of RutR can be removed enzymatically by CobB, we analyzed PatZ/YfiQ and YiaC acetylated RutR for CobB deacetylation (Supplementary Fig. 15b). We found that PatZ/YfiQ and YiaC-mediated acetylation is completely reversed upon addition of CobB and NAD$^+$ as judged by immunoblotting (Supplementary Fig. 15b). These data show that all lysine acetylation sites in RutR with functional importance are deacetylated by CobB in an NAD$^+$-dependent manner. Compared to AcK7, AcK11 and AcK7/11 in the unstructured N-terminal tail, AcK52 in the HTH-motif and AcK62 in α4 are far less efficiently deacetylated by CobB. These results unveil that CobB-mediated deacetylation occurs with two kinetic profiles appropriate to the kinetics of its acetylation, i.e. fast enzymatic acetylation/deacetylation at AcK7, AcK11, and AcK7/11 and slow non-enzymatic-acetylation/enzymatic deacetylation at AcK52 and AcK62.

## Discussion

RutR is a transcription factor, which is a master regulator of pyrimidine metabolism[19]. RutR binds to DNA-promoter sequences to repress the expression of the *rutAG* operon important for pyrimidine degradation and pyrimidine cellular transport[19]. Additionally, it binds to the promoter region of the *carAB*-operon indirectly activating transcription by abolishing the binding of the transcriptional repressor PepA to the *carAB*-promoter region[20,29,34]. This enables the expression of genes important for the synthesis of carbamoyl phosphate, the precursor for pyrimidine and arginine synthesis. Thereby, pyrimidine biosynthesis and degradation might be adjusted to the cellular conditions.

Our studies showed that the RutR protein level stays constant during bacterial growth. Modulation of RutR activity by post-translational lysine acetylation enables a tight regulation dependent on the cellular metabolic state. RutR was shown previously to be lysine-acetylated at several sites within the N-terminal DNA-binding domain (DBD) and the C-terminal ligand-binding domain (LBD)[36,37]. Lysine acetylation is a post-translational modification that is tightly connected to the cellular metabolism[43,67,96]. Lysine acetyltransferases use acetyl-CoA as acetyl group donor molecule. Besides, sirtuin deacetylases use NAD$^+$ as a stoichiometric co-substrate for deacetylation. Acetyl-CoA, acetyl-phosphate and NAD$^+$ are strong indicators for the cellular metabolic state[53,59,68–70,97,98]. Thereby, the cellular metabolic state determines the activities of these enzymes and in turn the acetylation status of their substrates. This enables a precise regulation of protein function adjusted to the cellular metabolic state.

We identified the KATs PatZ/YfiQ and YiaC to catalyze lysine acetylation of RutR at K7 and K11 in the unstructured histone-like N-terminal domain involved in RutR DNA-binding. Additionally, PatZ/YfiQ and YiaC are capable of simultaneously acetylating RutR at K7 and K11 and furthermore to act as N-(α)-acetyltransferases. This enables a regulation of RutR DNA-binding that resembles the interaction of the basic unstructured histone-tails with DNA in eukaryotes. Our data show that the highly positively-charged N-terminus contributes to DNA-binding and is important for electrostatic steering of RutR to the DNA-binding site. Upon neutralization of the positive charges by acetylation, long range electrostatic interactions with the negatively charged sugar-phosphate backbone are impaired. This is reflected by the observed decreased association rate constants for the interactions of K7/K11-acetylated RutR and box$_{rutA}$ and box$_{carA}$ dsDNA. Together with the increased dissociation rate constants, this results in the observed decrease in DNA-binding affinity. While it is not investigated how YiaC expression and/or activity is regulated, PatZ/YfiQ expression was shown to be under the control of catabolite repression by the cAMP-activated global transcriptional regulator CRP. In glucose medium PatZ/YfiQ expression was reported to be high during the stationary phase and constant during all growth phases in a medium containing acetate as the sole carbon source[99]. Other reports stated that the expression decreases in the stationary phase in complex medium[99–101]. Notably, genetically deleting *patZ/yfiQ* in various *E. coli* strains does not result in systemic decline of lysine acetylation suggesting that PatZ/YfiQ shows a narrow substrate profile and is a non-promiscuous, specific enzyme[68]. Additionally, the cytosolic acetyl-CoA concentration in glucose-fed *E. coli* in exponential growth phase was determined to be 610 μM as quantified by LC-MS/MS, which means that PatZ/YfiQ is almost saturated with acetyl-CoA considering a $K_M$ value of PatZ/YfiQ for acetyl-CoA of 110 μM[57,102]. Our data on PatZ/YfiQ-catalyzed K7/11-acetylation suggests that it occurs at rather high stoichiometry as we observed a strong signal in immunoblotting. This is supported by studies in eukaryotes which revealed that site-specific enzymatically catalyzed acetylation accumulates to higher median stoichiometries compared to non-enzymatic acetylation[103]. The enzymatic acetylation in the RutR N-terminal tail allows a dynamic and fast regulation of the RutR acetylation status dependent on the cellular metabolic state.

Next to this fast and dynamic enzymatically catalyzed acetylation, another source of acetylation, albeit with a slower kinetic profile, is represented by non-enzymatic lysine acetylation driven by the

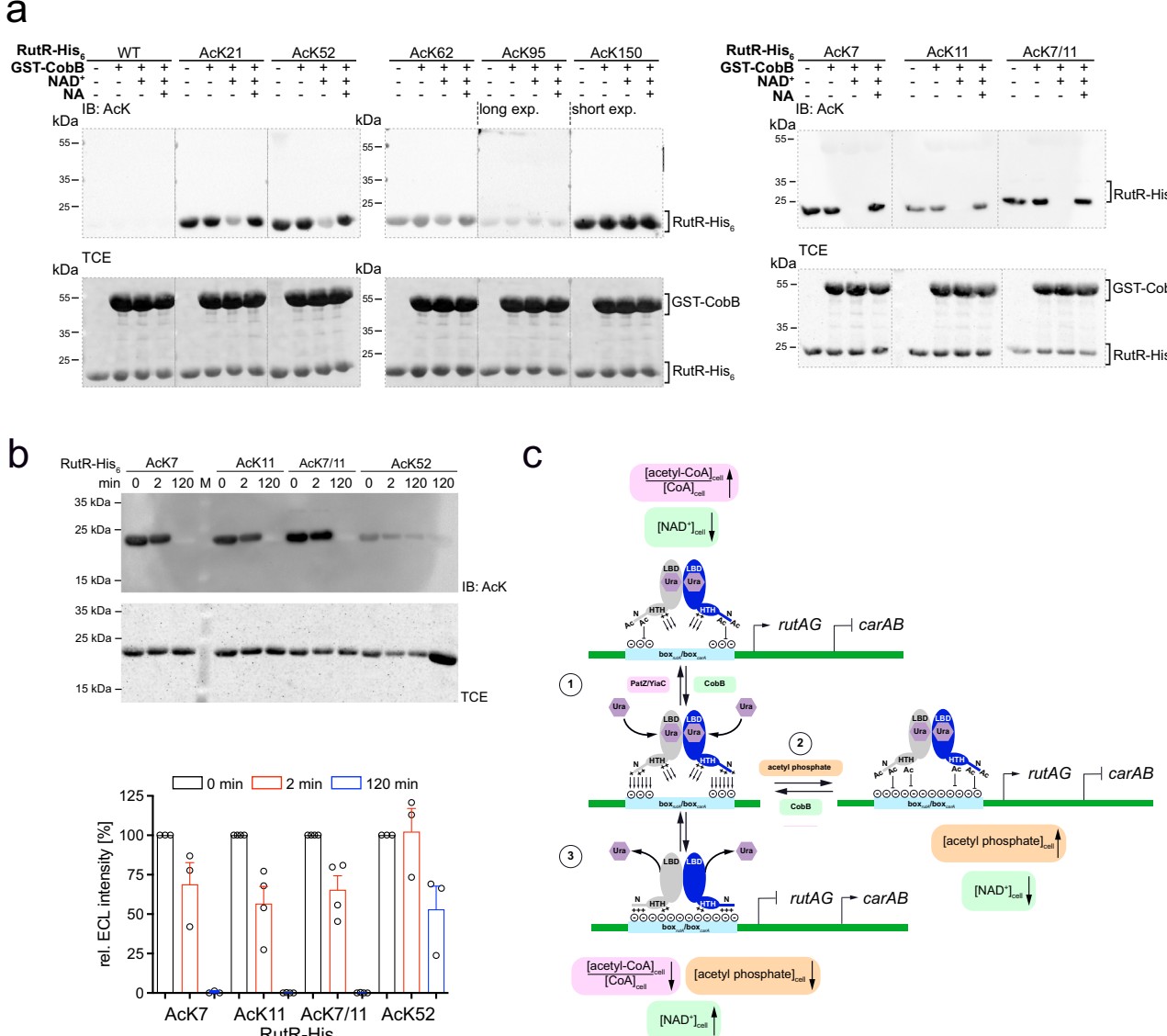

**Fig. 10 | The NAD⁺-dependent sirtuin CobB deactylates RutR at all functionally important sites but with different efficiency (a) and model for regulation of RutR function by lysine acetylation (b). a** CobB deacetylases RutR at all functionally important sites: AcK7, AcK11, AcK7/11 (right panel), AcK52, AcK62 (left panels). CobB also catalyzes deacetylation of RutR AcK21 within the basic patch (19-KKK-21). Immunoblotting (IB) using anti-AcK-AB (IB: AcK) was used to detect RutR deacetylation. TCE staining served as loading control. 10 mM NA is sufficient to inhibit CobB-catalyzed deacetylation at RutR AcK21, AcK52, and AcK62 but 50 mM NA is needed to inhibit deacetylation at AcK7, AcK11, and AcK7/11 (Supplementary Fig. 9A). One exemplary out of two biological replicates is shown (*n* = 2). Source data are provided as Source Data file. **b** Time-course experiment showing efficient deacetylation of RutR AcK7, AcK11 and double-acetylated AcK7/11 and less efficient deacetylation at AcK52. After 2 min (red bars), RutR is more than 25% deacetylated at AcK7, AcK11, and at AcK7/11, while after 120 min AcK52 is 50% deacetylated (blue bars) compared to the sample at 0 min (black bars). The experiment was performed in three/four biologically independent experiments for RutR AcK7/RutR AcK52

(*n* = 3)/RutR AcK11/RutR AcK7/11 (*n* = 4). Bars depict means ± standard deviations (*n* ≥ 3). Source data are provided as Source Data files. **c** We propose a model after which RutR is enzymatically lysine-acetylated in the unstructured N-terminal tail at K7 and K11 by the KATs PatZ/YfiQ and YiaC (1). Acetylation in the RutR N-terminal tail modulates the association and dissociation to dsDNA. Acetylation in the N-terminal tail can be efficiently reversed by CobB. The positively-charged N-terminal tail of RutR allows an electrostatic steering to the negatively-charged sugar-phosphate backbone of dsDNA. Accumulation of acetyl-phosphate results in non-enzymatic acetylation of RutR at K7, K11, and at K52 (2). K52-and K62-acetylation abrogates binding to dsDNA and can also be reversed by CobB. (3) RutR DNA-binding can be modulated by uracil (Ura) binding to the LBD. Switching-off RutR DNA-binding by lysine acetylation contributes to the activation of transcription of the *rutAG* operon. RutR acetylation at K52 and K7/K11 indirectly impairs the transcription of the *carAB* operon. Overall, the RutR lysine acetylation state is regulated enzymatically and non-enzymatically. This enables the adaptation of RutR transcriptional regulator activity to the cellular metabolic state.

gradient accumulation of acetyl-phosphate in bacteria. It was shown for *E. coli* that systemic acetylation gradually increases upon transition from the exponential growth phase to the stationary phase due to the accumulation of acetyl-phosphate up to millimolar concentration[61,63,66,95–98,104]. Growth in glucose-rich medium results in the accumulation of acetate, even under aerobic conditions through glycolysis, that peaks in the exponential growth phase[105]. This process

is known as overflow metabolism[106,107]. Acetate can be reversibly converted into acetyl-CoA by acetyl-CoA synthetase or into acetyl-phosphate by acetate kinase (AckA) if high intracellular acetate concentrations are achieved in the stationary phase (Fig. 1d)[68,108,109]. The average systemic acetylation stoichiometries were repeatedly confirmed to be very low suggesting that acetylations occurring at higher stoichiometry are enzymatically catalyzed[68,72]. Several reports

show that this overall accumulation of lysine acetylation is further increased upon nitrogen starvation[68]. This observation creates a functional link of lysine acetylation to RutR, whose primary role is the regulation of nitrogen supply. As we did not observe enzymatic acetylation of RutR at other lysines than K7 and K11 in the N-terminal tail, we show here for some sites and assume for all other identified acetylation sites in RutR, including AcK52 and AcK62, that these are acetylated due to a non-enzymatic, acetyl-phosphate-mediated mechanism. Another option is enzyme-catalyzed acetylation by so far unknown KATs. Published data suggest a high consistency between acetyl-phosphate-mediated sites detected in vivo in cell lysates and those detected using recombinant proteins in vitro. This suggests that the sites identified in vitro are also of physiological significance[68,98,102,110]. For RutR AcK52 we discovered that it switches off RutR's transcriptional repressor function. Our structural analyzes of RutR AcK52 revealed that acetylation abolishes DNA-binding mechanistically by electrostatic as well as steric effects.

We incorporated acetyl-lysine into a protein in *E. coli* in vivo by applying the GCEC. The incorporation of AcK52 in RutR shows that it modulates RutR transcriptional regulator activity. Notably, we were not able to incorporate acetyl-lysine at the N-terminus in vivo, neither AcK7, AcK11 nor both simultaneously. Moreover, the protein yield was even lower for the K7Q, K7R, K11Q, K11R and the double mutants thereof when expressed in vivo in *E. coli* U65 (Fig. 9). To improve the incorporation efficiency of unnatural amino acids (UNAA) at single and multiple sites by the GCEC *E. coli* strains were developed which carry genomic deletion of release factor 1 (RF1) or in which all amber stop codons are reassigned to ochre[111–113]. Moreover, evolved amber suppressor tRNAs were reported to substantially increase the incorporation efficiency[89]. In our study, however, the lower RutR protein yield in vivo of RutR modified at residues in the N-terminus is likely not due to a low incorporation efficiency or production of premature translational truncation products since we obtain RutR AcK7, AcK11 and AcK7/11 and the respective K to Q and K to R mutant proteins when expressed recombinantly in *E. coli* BL21 (DE3). Therefore, other mechanisms such as impaired mRNA stability might result in the observed lower or even absent protein levels as described also for the related transcriptional regulator TetR[114].

We also analyzed the deacetylation of RutR by the *E. coli* NAD⁺-dependent sirtuin deacetylase CobB. Our data show that all sites of functional importance, RutR AcK7, AcK11, AcK7/11, AcK52 and AcK62, are substrates of CobB. Primary sequence alignment comparing RutR proteins from different bacterial species shows that K7, K11, K52, and K62 in *E. coli* RutR are highly conserved (Supplementary Fig. 10). Data on expression of CobB showed that it is constantly expressed in the exponential growth phase and the stationary phase in glucose-fed *E. coli*[105]. This indicates that the activity of CobB is mostly regulated by the availability of the co-substrate NAD⁺. For glucose-fed *E. coli* in the exponential growth phase, cytosolic concentrations of NAD⁺ of 2.6 mM were reported, suggesting high activity of CobB[102]. CobB activity was shown to be regulated also by other mechanisms, e.g. it is inhibited by nicotinamide and by cyclic-diGMP (c-di-GMP)[38,115]. The cytosolic concentrations of nicotinamide were reported to reside between 30 and 90 μM during exponential growth phase and is lower in stationary phase[102,115]. Considering that nicotinamide shows a non-competitive inhibitory mechanism, these concentrations would still be of physiological importance even if NAD⁺ concentrations are more than one order of magnitude higher[71,102,115].

In agreement with the suggested kinetic profiles for RutR acetylation, we found similar kinetic profiles for the CobB-catalyzed deacetylation. The enzymatically acetylated sites, AcK7, AcK11, and AcK7/11, were efficiently deacetylated by CobB suggesting a regulatory role for these sites. In contrast, the non-enzymatically acetylated sites, such as AcK52 and AcK62, are deacetylated by CobB with a much slower kinetic efficiency. Besides a regulatory role for these acetylation events

slowly switching-off RutR transcriptional repressor activity with gradual increase in cellular acetyl-phosphate, this might indicate that CobB acts as a detoxifying enzyme removing lysine acetylation, which would otherwise result in an unwanted decline in RutR activity due to gradual accumulation of cellular acetyl-phosphate. This raises the question of molecular determinants of CobB substrate specificity. It was reported that CobB shows a strong preference for aromatic residues (Trp, Phe, Tyr) or arginine neighboring the target acetyl-lysine and disfavobring proline and acidic residues in +1 position[116]. For RutR we identified the following sequences as targets for CobB deacetylation: $_6$V(AcK)T$_8$, $_{10}$G(AcK)R$_{12}$, $_{20}$K(AcK)A$_{22}$, $_{51}$S(AcK)T$_{53}$, $_{61}$S(AcK)E$_{63}$. Thr and Ser residues are overrepresented at −1 and/or +1 position. Other determinants besides the primary sequence, such as the three-dimensional structure seem to be important for CobB substrate specificity as we showed also for mammalian sirtuins[116,117]. Inspection of the localization of the acetylation sites in the RutR structure suggests that their presence in regions with no secondary structure (AcK7 and AcK11), in regions allowing conformational flexibility (AcK62 at the N-terminus of α4) or in regions that are structurally optimal for CobB-recognition (AcK52 within α3 of the HTH-motif) promote CobB-mediated deacetylation efficiency.

Next to acetylation and deacetylation kinetics, the stoichiometry of site-specific acetylation is important for its physiological function. For PTMs such as lysine acetylation or Ser/Thr/Tyr phosphorylation, an appropriate stoichiometry is needed to exert a specific physiological function. If exerting a loss-of-function effect on a protein function, a high PTM stoichiometry is needed, whereas for a PTM resulting in a gain-of-function effect a lower stoichiometry is sufficient to facilitate a physiological effect. For RutR, we show that the sites of functional importance can be regulated enzymatically by the KATs PatZ/YfiQ and YiaC, i.e. K7, K11, and K7/K11, with the potential to accumulate to high stoichiometries, and non-enzymatically amongst others at the functionally important sites K7, K11, K52 and K62 by acetyl-phosphate likely only gradually increasing to higher stoichiometries. Importantly, these lysine acetylations could occur simultaneously at different sites and stoichiometries affecting the outcome.

Our data on *rutAG* expression suggest that additional events are needed besides the inactivation of the RutR transcriptional repressor to switch-on *rutAG* operon expression at least during the exponential growth phase. However, considering the role of RutR as a transcriptional repressor for expression of the *rutAG* operon, RutR lysine acetylation can be considered to exert a gain-of-function effect as lower PTM stoichiometries are sufficient to activate or at least to contribute to activation of gene expression of *rutAG* resulting in degradation of pyrimidines as nitrogen source. These low to moderate acetylation stoichiometries might be obtained by non-enzymatic acetylation in the late exponential growth phase or stationary phase when acetate is excreted under conditions of high ratios of carbon:nitrogen or carbon:magnesium and as a consequence cellular acetyl-phosphate increases[70,104,118]. For expression regulation of *carAB*, RutR acetylation can be an indirect mechanism to switch-off the expression of genes involved in pyrimidine biosynthesis by facilitating PepA repressor binding to the *carAB* promoter. This represents a loss-of-function effect of RutR lysine acetylation on *carAB* expression. Such effects require higher acetylation stoichiometries as obtained by enzymatic acetylation at K7 and/or K11. Thereby, biosynthesis of pyrimidines can be coordinated by the presence of KATs and the availability of cellular acetyl-CoA.

As a summary, for RutR, a major regulator of pyrimidine metabolism and nitrogen supply, we describe the regulation of its function by post-translational lysine acetylation with different kinetic profiles, i.e. fast and dynamic enzymatically catalyzed (by KATs) acetylation as well as slow and gradually increasing non-enzymatic (by acetyl-phosphate) acetylation. Acetyl-phosphate emerged very early during evolution enabling non-enzymatic acetylation as well as phosphorylation

of molecules[119]. In a primordial metabolism this enabled adjustment of cellular processes to the metabolic state before the emergence of catalysts for acetylation or phosphorylation. Notably, this non-enzymatic acetyl-phosphate-driven acetylation has been conserved until today and exists side-by-side with the KAT-catalyzed acetylation allowing to precisely adjust cellular processes according to their respective kinetic requirements. Lysine acetylation is a PTM that acts as a cellular sensor for the metabolic state. The expression levels of the KATs PatZ/YfiQ and YiaC and of the deacetylase CobB as well as the intracellular acetyl-CoA:CoA-ratio, acetyl-phosphate, $NAD^+$, nicotinamide and c-di-GMP concentrations determine the enzymatic and non-enzymatic acetylation state and acetylation stoichiometry of RutR (Fig. 10c). This enables the adjustment of the RutR transcriptional repressor activity and the metabolism of pyrimidines with different kinetic profiles to the cellular metabolic conditions. The enzymatic acetylation in the unstructured RutR N-terminal tail is functionally related to acetylation of flexible N-terminal histone tails in eukaryotes, both allowing a fast and dynamic regulation of gene expression. Sequence and structure comparisons of TetR-related transcriptional regulators show the presence of positively charged histone-like lysine-rich unstructured regions preceding the DNA-binding HTH domain in several members (Supplementary Figs. 10 and 11a, b; Supplementary Data 1). This suggests a general theme of regulation of TetR-family proteins by post-translational lysine acetylation. In fact, it was shown earlier that many TetR-family members contain a positively charged flexible N-terminal tail preceding the HTH motif contributing to DNA-binding (Supplementary Fig. 11)[16]. Many TetR-family transcriptional regulators are involved in metabolic adaptation of gene expression programs to altered conditions. This suggests that besides interacting with ligands, post-translational lysine acetylation might be an additional more general regulatory system to modulate TetR-family transcriptional regulator activity. We applied the genetic code expansion concept in vivo to reveal how lysine acetylation of RutR at endogenous levels affects its transcriptional regulator activity. However, it remains to be elucidated under which exact physiological conditions this mechanism of regulation plays a role in vivo. Our data conclusively show that also in bacteria, in analogy to histone acetylation in eukaryotes, DNA-binding and transcription is an important target for lysine acetylation to adapt gene expression programs to the cellular metabolic state.

## Methods

### Expression and purification of proteins

Non-acetylated RutR proteins, the KATs PatZ/YfiQ, PhnO, RimI, YiaC, YjaB, in their wild-type and catalytically dead form, were expressed as $His_6$-tagged fusion proteins (pRSFDuet-1; Merck Biosciences) in *E. coli* BL21 (DE3) cells (Supplementary Data 4). CobB and its catalytically dead variant were additionally expressed as GST-fusion using the pGEX-4T5/Tev vector derived from pGEX-4T1 (GE Healthcare). The protein expressions were conducted in 6-10 L TB or LB media. In brief, cells were cultivated to an $OD_{600}$ of 0.6 (37 °C; 160 rpm). Expression was induced by the addition of 0.1-0.3 mM of isopropyl-β-D-thiogalactopyranoside (IPTG) and was performed for 12-16 h (18 °C; 180 rpm). The cells were harvested by centrifugation (4000 g, 20 min) and resuspended in resuspension buffer (RutR proteins: 100 mM $K_2HPO_4/KH_2PO_4$ pH 6.4, 150 mM NaCl, 5 mM $MgCl_2$, 2 mM β-mercaptoethanol; KATs: 30 mM Tris/HCl pH 8.0, 100 mM NaCl, 2 mM β-mercaptoethanol; GST-CobB: 20 mM HEPES pH 7.5, 50 mM NaCl, 2 mM β-mercaptoethanol) containing 0.2 mM Pefabloc protease inhibitor cocktail. Cell lysis was done either by sonication or by cell disruptor, the cleared lysate (20000 g, 45 min) was applied to the equilibrated Ni-NTA- or GSH-affinity-chromatography column (Ni-NTA: resuspension buffer plus 10-20 mM imidazole; GSH: resuspension buffer). Washing was done with high-salt buffer (equilibration buffer with 500 mM NaCl). Elution from the Ni-NTA-column was performed over a gradient

of 20–500 mM imidazole. GST-CobB, was eluted from the GSH-column with elution buffer (20 mM HEPES pH 7.5, 500 mM NaCl, 2 mM β-mercaptoethanol, 30 mM glutathione (GSH)). After elution, the protein was concentrated by ultrafiltration and applied to an appropriate size-exclusion chromatography column (HiLoad 16/600 Superdex 75 or 200 pg; GE Healthcare). The concentrated fractions were shock frozen in liquid nitrogen and stored at −80 °C. Protein concentrations were determined measuring the absorption at 280 nm using the protein´s extinction coefficient.

### Incorporation of N-(ε)-acetyl-lysine

The site-specific incorporation of N-(ε)-acetyl-L-lysine was conducted by addition of 10 mM N-(ε)-acetyl-lysine (CHEM-IMPEX INT´L LNC) and 20 mM nicotinamide to block the *E. coli* CobB deacetylase to the *E. coli* BL21 (DE3) culture at an $OD_{600}$ of 0.6 (37 °C). Protein expression was induced by addition of 0.1-0.3 mM IPTG following a cultivation of additional 30 min (160 rpm, 37 °C). Acetylated RutR proteins were expressed from a pRSFDuet-1 vector additionally encoding the synthetically evolved *Methanosarcina barkeri* $Mb$tRNA$_{CUA}$ and the acetyl-lysyl-tRNA-synthetase (AcKRS3) as described previously[79,81,120]. Briefly, the pRSFDuet-1 vector was modified for genetic encoding of acetyl-lysine[79]. Using restriction cloning, the coding sequence *acKRS* of the acetyl-lysyl-tRNA synthetase of *Methanosarcina barkeri* (variant *AcKRS4* representing *AcKRS3* with a S123G mutation) including *glnS* promoter and terminator was inserted into an *Sph*I restriction site. The coding sequence *pylT* for tRNA$_{CUA}$ including *lpp* promoter and *rrnC* terminator was inserted into an *Xba*I restriction site. Additionally, the vector contains a G to P mutation C-terminal of the Start-M codon to prevent α-N−6-phosphogluconoylation at the N-terminus, which would interfere with mass spectrometric determination of protein molecular weights[121]. Moreover, two *Eco*RI restriction sites outside the multiple cloning site were removed by site-directed mutagenesis. For the expression of GST-tagged proteins, the vector pGEX-4T5/TEV was used. This vector is a modified version of pGEX-4T1 (GE healthcare) with a tobacco etch virus (TEV)-protease cleavage site inserted between the thrombin cleavage sequence and the multiple cloning site. The incorporation of N-(ε)-acetyl-L-lysine in *E. coli* is done co-translationally as a response to an amber stop codon. For expression of the double-acetylated RutR AcK7/11, we used a dual plasmid system including the pRSFDuet-1 vector encoding AcKRS3 und $Mb$tRNA$_{CUA}$ and a pUC57 vector encoding for the optimized $Mb$tRNA$_{opt}$[89].

### Plasmids, enzymes and antibodies

For expression in bacterial cells pRSFDuet-1 (Merck/Sigma-Aldrich, Novagen) and a vector derived from pGEX-4T-1 (GE Healthcare) were used and modified as described above (Supplementary Data 4). Mutations were introduced by site-directed mutagenesis according to QuickChange protocol (Supplementary Table 8). For cloning, Phusion-DNA-polymerase, Taq-DNA-ligase, T5 exonuclease, and restriction enzymes were used (New England Biolabs). Anti-AcK-, anti-$His_6$-, anti-FLAG-primary antibodies and suitable HRP-coupled secondary antibodies were purchased from Abcam, Cell Signaling Technologies and Invitrogen (rabbit anti-AcK-AB: abcam, ab21623/dilution: 1:1000 in 3-5% (w/v) milk; mouse anti-$His_6$-AB: abcam, ab18184/dilution: 1:000 in 3-5% (w/v) milk; rabbit anti-FLAG-AB: Cell Signaling Technology, CST14793/dilution: 1:1000 in 3–5% (w/v) milk; mouse anti-FLAG-AB: Invitrogen, FG4R/dilution: 1:2000 in 3% (w/v) milk; goat anti-rabbit-HRP-AB: abcam, ab6721/dilution: 1:10000 in 3-5% milk; rabbit anti-mouse-HRP-AB: abcam, ab6728/dilution: 1:10000 in 1-5% (w/v) milk).

### Analysis of endogenous expression of RutR-FLAG

Growth curves for *E. coli* BW30270 and U65 with genomic insertion of a FLAG-tag C-terminal of *rutR* (*rutR-FLAG*) or deletion of *rutR* (Δ*rutR*) were analyzed (Supplementary Table 1). 6 mL LB medium was inoculated to an $OD_{600}$ of 0.05 with unmodified, *rutR*-FLAG or Δ*rutR* variants

of the indicated strains. Growth was monitored at 37 °C for 24 h in a six-well plate. A region encoding for a FLAG-tag was genomically inserted downstream of *rutR* in a BW30270 background (BW30270 *rutR-FLAG*). 500 mL LB was inoculated to an $OD_{600}$ of 0.05 and growth at 37 °C was monitored. Samples were analyzed at indicated $OD_{600}$ values (the last sample was collected after 8 h) and 20 µg protein from cell lysates was subjected to immunoblotting (IB). RutR-FLAG was stained with anti-FLAG antibody (IB: FLAG; CST14793). Total protein staining with 2,2,2-trichloroethanol (TCE) served as loading control. Signal intensities were correlated to the respective total protein staining intensities and normalized to the expression level at $OD_{600}$ of 0.2.

## Immunoblotting of RutR-FLAG

For immunoblotting of C-terminally FLAG-tagged RutR protein, bacterial cells were sonicated in lysis buffer (100 mM $K_2HPO_4$/$KH_2PO_4$ pH 7.4, 150 mM NaCl, 1 mM EDTA, 20 mM nicotinamide, 2.5% Triton X-100, 5 mM AEBSF, 5 mM BA). Lysates were resolved by SDS-PAGE and analyzed by immunoblotting using an anti-FLAG-AB (Invitrogen, FG4R) primary antibody and a rabbit anti-mouse-HRP secondary AB (abcam, ab6728). Immunoblotting was conducted using a standard protocol. Detection was done by using enhanced chemiluminescence (Roth). Immunoblot analysis was done by measuring mean grey intensities using ImageJ software (http://rsbweb.nih.gov/ij). For statistical analyzes, a two tailed sudent's t-test was applied.

## In vitro acetylation/deacetylation assay

For in vitro acetylation, 10 µM of purified RutR protein was incubated with/without 50 µM acetyl-CoA and of 10 µM recombinantly expressed and purified acetyltransferase (PatZ, RimI PhnO, YiaC, YjaB) for 18 h at 37 °C in assay buffer (50 mM Tris/HCl pH 8.0). Reactions were stopped by addition of SDS-sample buffer and boiling the samples for 5 min at 95 °C. For the non-enzymatic acetylation or RutR with acetyl-phosphate at indicated concentrations, 10 µM or 20 µM of the purified RutR-protein was incubated with different concentrations of acetyl-phosphate in 100 mM $K_2HPO_4$/$KH_2PO_4$ buffer at different pH and for different incubation times. In vitro deacetylase assays with GST-CobB were performed in deacetylase buffer (50 mM Tris/HCl pH 8, 100 mM NaCl, 5 mM $MgCl_2$). A total of 15–25 µM of acetylated RutR protein was incubated with 30 µM GST-CobB in the presence or absence of 7.5 mM $NAD^+$. As controls, reactions containing 10 mM nicotinamide were performed. For double acetylated RutR AcK7/11, a reaction with 50 mM nicotinamide was also performed to inhibit deacetylation. Reactions were conducted for 2 h at 37 °C. Reactions were stopped by addition by SDS-sample buffer and subsequent boiling for 5 min at 95 °C. The reaction products were analyzed by immunoblotting (rabbit anti-AcK-AB: abcam, ab21623; goat anti-rabbit-HRP-AB: abcam ab6721). Uncropped and unprocessed scans of the most important blots are provided in the Source Data file.

## Isothermal titration calorimetry (ITC) measurements

The interactions of non-acetylated and acetylated RutR proteins with $box_{rutA}$ and $box_{carA}$ dsDNA were thermodynamically characterized by isothermal titration calorimetry on an $ITC_{200}$ instrument (Malvern Panalyical)[122]. Double-strand DNA (dsDNA) fragments of $box_{rutA}$ and $box_{carA}$ were obtained by annealing from ssDNA oligonucleotides (Eurofins) (Supplementary Table 2). The measurements were done in ITC buffer (100 mM $K_2HPO_4$/$KH_2PO_4$ pH 6.4, 150 mM NaCl, 5 mM $MgCl_2$, 2 mM β-mercaptoethanol) at 20 °C or 25 °C as indicated. For a typical reaction 100 µM RutR protein (syringe) was stepwise titrated into a solution containing 5 µM $box_{rutA}$ and $box_{carA}$ (cell). As primary data, the heating power per injection was plotted as a function of time until binding saturation was achieved. A one-site binding model was fitted to the data using the software supplied by the manufacturer. The measurements resulted in the stoichiometry of binding (N), the enthalpy change (ΔH) and the equilibrium-association constant ($K_A$) as

direct readout. ΔG, ΔS, and the equilibrium dissociation constant ($K_D$) are derived from the primary data. Each titration was at least conducted three times and Supplementary Table 3 shows the mean values ± standard deviations (s.d.). We used the standard EDTA-$CaCl_2$ sample test as described by Malvern Panalytical/MicroCal to assess the statistical significance of individual observations. These gave values within the manufacturer's tolerances of ±20% for $K_A$ values and ±10% in ΔH.

## Surface plasmon resonance (SPR) measurements

Surface plasmon resonance (SPR) experiments were carried out using a Biacore T200 (Cytiva) at 25 °C, a flow rate of 50 µL $min^{-1}$ and the data was collected at 10 Hz. Streptavidin capture chips were prepared using amine coupling. Briefly, a CM5 sensor chip was activated using ((3-dimethylaminopropyl)−3-ethylcarbodiimide (EDC)/ N-hydroxysuccinimide (NHS)) followed by a NeutrAvidin (Thermo-Fisher) injection and surface inactivation with ethanolamine. Biotinylated $box_{carA}$, $box_{rutA}$ or control dsDNA was captured to immobilization levels ~ 100 RU (-0.1 ng $mm^{-2}$) (Supplementary Table 2). Due to unspecific binding of RutR to the negatively charged carboxymethylated dextran layer, a 20 s pulse with 1 mg $mL^{-1}$ polyethylenimine (PEI) followed by a 30 s running buffer injection at 100 µL $min^{-1}$ were included for charge inversion of the surface and avoid unspecific binding before each single cycle kinetic experiment.

Concentration series of RutR variants were prepared as two-fold serial dilutions in running buffer (100 mM $K_2HPO_4$/$KH_2PO_4$ pH 6.4, 300 mM NaCl, 0.05% Tween20) covering concentrations from 6.25–100 nM. Single cycle experiments were recorded and double referenced. After each experiment, the surface was regenerated using a 30 s pulse with 1 M NaCl in running buffer. Kinetics were determined using a 1:1 Langmuir model and rate constants were reported as average and standard deviation of three independent experiments.

## Elecrophoretic-mobility shift assay (EMSA)

The protein-DNA interactions between RutR variants and promoter ($box_{rutA}$: 50 bp segment of $Prom_{rutA}$, $box_{carA}$: 46 bp segment of $Prom_{carA}$), as well as control (ctrl) oligonucleotides with or without addition of exogenous uracil, were studied in EMSAs (Supplementary Table 2). Double-strand DNA (dsDNA) fragments of $box_{rutA}$ and $box_{carA}$ were obtained by annealing from ssDNA oligonucleotides (Eurofins). Their concentrations were adjusted according the synthesis scale. Binding of RutR proteins and $box_{rutA}$ and $box_{carA}$ dsDNA was conducted in a final volume of 10 µL in 1x EMSA binding buffer (20 mM Tris-HCl (pH 7.5), 100 mM KCl, 2 mM DTT, and 10% glycerol). To form RutR-dsDNA complexes, increasing concentrations of purified RutR protein was added to 25 nM of the dsDNA fragment as indicated. For the studies assessing the impact of uracil on dsDNA-binding, 200 nM RutR variants were used. The binding reaction was incubated for 20 min at 30 °C, and then 10 µl of each sample was separated next to a DNA size ladder (New England Biolabs) on 9% non-denaturing polyacrylamide gels (acrylamide/bisacrylamide, 29:1; 0.5× TBE) that were run under cold conditions (4 °C). For visualization of DNA, the gels were stained with ethidium bromide. For the $box_{carA}$ dsDNA we observed a fainter signal compared to $box_{rutA}$ dsDNA. This might be due to lower AT-content in $box_{carA}$ dsDNA compared to $box_{rutA}$ dsDNA[123]. Results were recorded using a gel documentation system (ChemDoc, Biorad or INTAS). After quantification in ImageJ, the data was evaluated by fitting the following quadratic equation in GraphPad Prism: $Y = ((Ymax\text{-}Ymin)*((A + X + K)\text{-}sqrt((sqr(A + X + K))−4*A*X))/(2*A))+Ymin$ (with: $X$: concentration of RutR [nM]; $Y$: concentration of free oligonucleotide [%]; $A$: concentration of DNA [nM]; $K$: equilibrium dissociation constant $K_D$, constraints: $Ymax = 0$; $Ymin < 100$, $A = 25$).

## Bacterial strain generation

**λ Red-mediated genomic recombination.** λ Red-mediated homologous recombination was performed according to the established

procedure[124]. Linear DNA fragments carrying at both ends short homologies to the desired genomic locus and a FRT (FLP recombination target) site flanked antibiotic resistance gene were generated using the indicated oligonucleotides and plasmids (Supplementary Tables 8 and 10). Acceptor strains were transformed with the temperature-sensitive plasmid pKD46, which encodes the enzymatic λ Red system. It allows homologous recombination between linear DNA fragments and the chromosomal locus that is targeted by flanking sequences on the DNA fragment. Cells were grown at 28 °C in the presence of 10 mM L-arabinose to induce expression of the λ Red recombinase. Electrocompetent cells were then prepared and transformed with at least 100 ng of the linear DNA fragment in 1 μL H$_2$O. Selection was performed at 37 °C for loss of pKD46. The resulting colonies were restreaked on LB agar plates with appropriate antibiotics. Successful recombinants grew on the antibiotic that the encoded resistance targets, but not on ampicillin due to the loss of pKD46. Colony PCR confirmed the expected insertion size with the indicated oligonucleotides (Supplementary Table 9). For removal of the antibiotic resistance marker, the recombinants were transformed with the temperature-sensitive plasmid pCP20 encoding the site-specific FLP recombinase and ampicillin resistance. At 28 °C, the transformants could grow on LB agar plates containing ampicillin. The recombinase was expressed at 42 °C and removed the resistance cassette, which resulted in deletion of the antibiotic resistance gene and one remaining FRT scar. After restreaking on LB agar at 42 °C, pCP20 could not be propagated anymore and all antibiotic resistances were lost. The resulting mutants were confirmed by colony PCR and sequencing of the resulting DNA fragment. Bacterial strains grown in LB medium were stored at −80 °C upon addition of 30% glycerol.

**P1vir phage-dependent transduction.** Transduction of gene loci was performed with a P1*vir* phage[125]. First, P1*vir* lysates were prepared from the respective donor strain. 5 mL LB medium containing 2.5 mM CaCl$_2$ was inoculated and grown to an OD$_{600}$ of 0.5. Then 100 μL of P1*vir* lysate, previously prepared with any other strain, was added. Cells were further cultivated at 37 °C for 3 to 4 hours until lysis of the cultures became apparent by clearing of the medium and formation of cell debris. After addition of 40 μL chloroform, cell debris was removed by centrifugation at 4000 x *g* and the supernatant P1*vir* lysate was transferred to a fresh tube. The extraction of the supernatant with chloroform was repeated twice. A few drops of chloroform were added for storage of the viral lysate at 4 °C. The recipient strain was grown in LB medium containing 2.5 mM CaCl$_2$ to an OD$_{600}$ of 0.8. Then 100 μL P1*vir* lysate propagated on the donor strain was added. The mixture was incubated for 30 minutes at 37 °C before 100 μL 1 M trisodium citrate was added to stop further phage adsorption. The infected cells were pelleted and washed with 1 mL 50 mM trisodium citrate three times. Afterwards, the cells were resuspended in 100 μL 50 mM trisodium citrate and plated on selective LB agar plates. Transductants were analyzed as recombinants.

**Chromosomal integration of transcriptional Prom$_{carA}$-lacZ fusion into attB site.** Transcriptional *lacZ* reporter fusions were integrated into the λ attachment site *attB*[126]. The fusion of the *carA* promoter fragment Prom$_{carA}$ and *lacZ* was generated in pKESL253 with the indicated oligonucleotides (Supplementary Table 10). The acceptor strain was transformed with the temperature sensitive plasmid pLDR8 coding for the λ integrase and a kanamycin resistance. The transformants were grown at 28 °C on LB agar plates containing kanamycin. Colonies from those plates were used for inoculation of LB with kanamycin. Cultures were incubated over night at 28 °C before dilution into LB with kanamycin and growth for 90 minutes at 37 °C. At 37 °C, the λ integrase was expressed and pLDR8 was not propagated. Chemocompetent cells were prepared from this culture. Prom$_{carA}$-pKESL253 was digested with *Bam*HI and purified

from an agarose gel. 10 ng of the resulting linear fragments were religated to create origin-less DNA circles containing the transcriptional fusion and a spectinomycin resistance cassette. The circles were transformed in the prepared acceptor strain, in which expressed λ integrase was present. The integrase performed the integration of the *attP* site encoded in the origin less circle into the chromosomal *attB* site. Integrants were selected on LB agar plates with spectinomycin at 42 °C. After restreaking on spectinomycin, as a selection for successful integration, and kanamycin, as indication of loss of pLDR8, the integrants were analyzed by the following colony PCRs and sequencing of the genomic locus. To analyze the chromosomal integration into the *attB* site PCR reactions were conducted (Supplementary Table 11).

### RNA isolation
RNA isolation was performed according to the manufacturers' manuals of the RNAprotect Bacteria Reagent and the RNeasy Mini Kit (Qiagen). Specifically, cultures of the indicated strains were diluted from overnight cultures into LB medium (containing AcK, NA and IPTG as applicable) to an OD$_{600}$ of 0.05. A specific culture volume (1 mL at OD$_{600}$ of 0.6; 0.2 mL at OD$_{600}$ of 3.0) was added to the doubled volume of RNAprotect Bacteria Reagent at the respective OD$_{600}$ value and treated as described in the manual. The manufacturer's manual of the RNAeasy Mini Kit was followed for subsequent RNA extraction according to the procedure for lysis with lysozyme and proteinase K. An on-column DNaseI digestion was performed as described for the RNase-Free DNase Set. RNA was eluted in 20 μL H$_2$O. Subsequently, an additional DNaseI digestion step was performed. 17 μL RNA, 2 μL 10x RDD buffer from the RNase-Free DNase Set and 1 μL DNaseI were incubated for 10 minutes at 37 °C. Then 0.2 μL 5 M EDTA in DEPC-H$_2$O were added and incubated at 75 °C for 10 minutes to stop the reaction. RNA concentration was determined by determination of the absorption at 260 nM using a NanoDrop.

### Urea-PAGE
To analyze the quality of the isolated RNA, a denaturing urea-polyacrylamide gel electrophoresis (PAGE) was performed. 250 ng RNA in 2.5 μL H$_2$O was mixed with 2.5 μL 2x RNA Loading Dye and incubated at 70 °C for 10 minutes. Subsequently, the urea-PAGE was performed in 0.5x TBE at 200 V for 90 minutes. Afterwards, the gel was stained using ethidium bromide (0.5 μg/mL in 0.5x TBE). The quality of the RNA was considered satisfactory when bands corresponding to the 23 S rRNA and 16 S rRNA were detected without appearance of degraded RNA.

### Reverse transcription
RNA was transcribed into cDNA by reverse transcriptase (RT) according to the manufacturer's protocol of the kit (SuperScript III First-Strand Synthesis System, Invitrogen). Briefly, equal RNA amounts of each sample in 4 μL solution were mixed with 4 μL random hexamer oligonucleotides (50 ng/μL) and 2 μL of dNTPs (10 mM) and incubated for 5 minutes at 65 °C. After cooling down on ice for at least 1 minute, 2 μL of 10x RT buffer, 4 μL of 25 mM MgCl$_2$, 2 μL of 0.1 M DTT, 1 μL of RNase OUT (40 U/μL), and 1 μL of Superscript III RT (200 U) were added. The reaction mixture was incubated for 10 minutes at 25 °C, 1 hour at 50 °C, and 5 minutes at 85 °C. To remove the template RNA, it was degraded by addition of 1 μL RNase H and incubation for 20 minutes at 37 °C. Additionally, a control reaction of RNA pooled from all samples without addition of RT was performed accordingly. A fraction of each reaction, equaling 50 ng of RNA that was used for reverse transcription, was analyzed in a PCR using oligonucleotides for the endogenous control genes (Supplementary Table 12). PCR products were resolved in agarose gels. Equal intensities were confirmed for samples while the negative control was supposed to show no reaction product.

## Quantitative PCR

Quantitative PCR (qPCR) was performed according to the manufacturer's protocol of the kit (GoTaq qPCR Master Mix, Promega). The amount of cDNA that was used for the reaction was based on the known RNA concentration that was used for reverse transcription. cDNA synthesized from 50 ng of RNA in 2 μL volume was used as template. The cDNA was diluted 1:10000 and 1:10 for the detection of 16 S rRNA and $rpoD$ gene expression, respectively. Additionally, 6 μL 2x buffer, 0.12 μL CXR Reference Dye (if measured in StepOne Real-Time PCR System), and each 0.6 μL forward and reverse oligonucleotide (10 μM) (Supplementary Table 12) were added. The final volume of 12 μL was achieved by the addition of $H_2O$. Controls were performed with the following samples instead of cDNA: equal amounts of the reverse transcription reaction without RT to show genomic DNA contaminations and $H_2O$ to reveal general impurities e.g. from pipette tips. The reaction was performed in 96-well plates closed with adhesive films in technical duplicates. The following cycling program was applied: one step at 95 °C (3 minutes), 40 steps of 95 °C (5 seconds), 60 °C (30 seconds), and measurement of a melting curve (60 °C (1 minute) to 95 °C (15 seconds) in steps of 0.3 seconds). Cycle threshold ($C_t$) values for each sample were obtained from the respective software, Applied Biosystems StepOne Software or Bio-Rad CFX Maestro, and melting curves were inspected visually to ensure reliability of these values. $C_t$ values were normalized to those of the house keeping gene 16S-rRNA or $rpoD$ as endogenous control resulting in $\Delta C_t$ values. Comparison among samples was performed by calculating $\Delta\Delta Ct$ ($\Delta C_t$ sample 1 – $\Delta C_t$ sample 2) values. The mRNA level fold changes were calculated as $2^{\wedge}\Delta\Delta C_t$.

## Determination of β-galactosidase activity by *lacZ* reporter assay

The transcriptional reporter assay was performed according to a published standard procedure[125]. In brief, *E. coli* U65 Δ*rutR* with a genomic Prom$_{carA}$-*lacZ* fusion (U65 Δ*rutA* Prom$_{carA}$-*lacZ*) was transformed with pRSFDuet-1 empty, pRSFDuet-1/*rutR*, pRSFDuet-1/ *rutR* K52 or *rutR* K52Q or pRSFDuet-1/*rutRK52$_{amber}$* as indicated for each experiment. Overnight cultures were used to inoculate 30 mL LB medium containing 10 mM acetyllysine and 20 mM nicotinamide to an $OD_{600}$ of 0.05 in the presence of indicated concentrations of IPTG. Cultures were harvested and analyzed in a *lacZ* reporter assay at an $OD_{600}$ of 0.6. Cultures were harvested and lysed in the presence of 5 mM benzamidine (BA) and 4-(2-aminoethyl) benzenesulfonyl fluoride (AEBSF). Bacterial cultures were harvested and diluted in Z-buffer (60 mM $Na_2HPO_4$, 40 mM $NaH_2PO_4$, 10 mM KCL, 1 mM $MgSO_4$, 100 μg/ mL chloramphenicol) to a final volume of 1 mL. Cells were lysed by the addition of a drop of 0.1% SDS and chloroform. 30 μg of proteins from cell lysates were analyzed in immunoblots (IB), probed with anti-His$_6$-AB (mouse anti-His$_6$-AB: abcam, ab18184; rabbit anti-mouse-HRP-AB: abcam, ab6728), and related to total protein staining with 2,2,2-trichloroethanol (TCE). 400 μL of o-nitrophenyl-β-galactoside (2 mg/mL) was added to start the β-galactose-dependent reaction and after incubation for a duration that depended on each specific dilution, the reaction was stopped by the addition of 500 μL of 1 M $Na_2CO_3$. Absorption at 420 nm ($A_{420}$) was recorded in cuvettes with d = 1 cm. Miller units were calculated according to the following equation: β-galactosidase activity [miller units] = ($A_{420}$*dilution factor*1000)/ ($OD_{600}$*reaction time in min). Values represent means ± standard deviation (s.d.) of three independent experiments. *$p < 0.05$, **$p < 0.005$, ***$p < 0.001$ indicates a statistically significant difference of the samples compared as determined by two-sided student's t-test.

## Crystallisation, data collection, phasing, model building and refinement

The RutR AcK52•uracil crystals were obtained by the sitting drop vapor diffusion method in Morpheus II screen in the condition containing 50% precipitation mix 6 (25% w/v PEG 4000, 40% w/v 1,2,6-

Hexanetriol), 0.1 M buffer system 4 (MOPSO, Bis-Tris pH 6.5), 100 mM amino acids mix 2 (0.2 M DL-Arginine hydrochloride, 0.2 M DL-threonine, 0.2 M DL-histidine monohydrochloride monohydrate, 0.2 M DL-5-hydroxylysine hydrochloride, 0.2 M trans-4-hydroxy-L-proline). 30% (w/v) D-glucose was used as cryoprotectant. The crystals belonged to the orthorhombic space group P2$_1$2$_1$2$_1$ with one RutR AcK52•uracil dimer per asymmetric unit. The native data set was collected at the ESRF Grenoble/France at 100 K on beamline ID30B at a wavelength of 1 Å using a Dectris Pilatus 6 M detector. The oscillation range was 0.1° and 1800 frames were collected. The program XDS[127] was used for indexing and integration. Scaling and merging was performed with Aimless 0.7.4[127,128]. Initial phases were determined using the program Phaser 2.8.3 within the Phenix program suite 1.17.1_3660 and the non-acetylated RutR-structure (PDB: 4JYK) structure as search model[129,130]. Refinement was done using the program phenix.refine[131]. Restraints for the uracil were obtained by the program eLBOW in Phenix[132]. Coot was used for model building into the 2F$_o$-F$_c$ and F$_o$-F$_c$ electron density maps in iterative rounds of refinement with phenix.refine[133,134]. Finally, 100% of all residues are in the allowed regions of the Ramachandran plot as judged by the program MolProbity[135,136]. All structure figures were made with PyMOL 2.3.4[137]. Data collection and refinement statistics are given in Supplementary Table 4. R$_{work}$ is calculated as follows: $R_{work} = \sum |F_o - F_c|/ \sum F_o$. F$_o$ and F$_c$ are the observed and calculated structure factor amplitudes, respectively. R$_{free}$ is calculated as R$_{work}$ using the test set reflections only. The structure is deposited in the PDB database (http://www.rcsb.org) under the accession number 6Z1B.

## Mass spectrometric determination of molecular masses of intact proteins by electrospray-ionization mass-spectrometry (ESI-MS)

Proteins were prepared in RutR storage buffer at concentrations of 1 μg/μL for intact mass determination. All samples were analyzed on a Triple TOF 6600 (Sciex) mass spectrometer equipped with a DuoSpray Ion Source that was coupled to a Shimadzu LC30AD HPLC System (Shimadzu). Approximately 100 ng of intact proteins (in 0.1% formic acid in water) were loaded onto a Jupiter C4 column (1 mm × 150 mm, 5 μm, 300 A, Phenomenex) and chromatographically separated at a constant flow rate of 100 μL/min using the following gradient: 20-60% solvent B (0.1% formic acid in acetonitrile) within 6 minutes followed by 85% solvent B for 1 minute and 20% B for 3 minutes. MS1 scans were acquired from 600-1600 m/z with 250 msec accumulation time. MS1 scans were summed across the chromatographic protein peak and the in summary spectra were exported in simple txt format for further analysis in the MagTran 1.02 deconvolution software[138].

## Mass spectrometric detection of acetylated peptides by liquid chromatography tandem mass spectrometry (LC-MS/MS)

After completion of protein purification or respective assays, proteins were prepared for mass spectrometric detection of lysine acetylation sites by tryptic digestion. After the removal of all solvents in a vacuum concentrator, samples were resuspended in 20 μL 8 M urea. DTT was added to a final concentration of 5 mM for reduction before incubation for 1 hour at 25 °C. Alkylation was performed by chloroacetamide (CAA) addition to a concentration of 40 mM and incubation in the dark at room temperature for 30 minutes. A pre-digestion step with lysyl endopeptidase was performed, in which lysyl endopeptidase was added in an enzyme-to-substrate ratio of 1 to 75 and incubated at 25 °C for 4 hours. Subsequently, 50 mM Tris/HCl (pH 8) was added to a final volume of 100 μL resulting in a urea concentration of 1.6 M. Trypsin digestion was performed for 18 hours at 25 °C with an enzyme to substrate ratio of 1 to 50. Formic acid was added to a final concentration of 1%. After stage tipping, the samples were analyzed by liquid chromatography coupled to tandem mass spectrometry, also called data-dependent acquisition. All samples were analyzed on a Q Exactive Plus Orbitrap (Thermo Scientific) mass spectrometer that was

coupled to an EASY nLC (Thermo Scientific). Peptides were loaded with solvent A (0.1% formic acid in water) onto an in-house packed analytical column (50 cm–75 μm I.D., filled with 2.7 μm Poroshell EC120 C18, Agilent). Peptides were separated chromatographically at a constant flow rate of 250 nL/min using the following gradient: 3–5% solvent B (0.1% formic acid in 80% acetonitrile) within 1 minute, 5-30% solvent B within 42 minutes, 30–40% solvent B within 8 minutes, 40-95% solvent B within 1 minute, followed by washing and column equilibration. The MS1 survey scan was acquired from 300-1750 m/z at a resolution of 70,000. The top 10 most abundant peptides were isolated within a 2.0 Th window and subjected to higher-energy collisional dissociation fragmentation at a normalized collision energy of 27%. The automatic gain control target value was set to $5 \times 10^5$ charges, allowing a maximum injection time of 55 ms. Product ions were detected in the Orbitrap at a resolution of 17,500. Precursors were dynamically excluded for 10.0 s. Data were acquired in centroid mode. Mass spectrometric raw data were processed with MaxQuant (version 1.5.3.8) using default parameters[139]. Briefly, MS2 spectra were searched against either the UniProt *E. coli* database, including a list of common contaminants for analysis of recombinant RutR variants or a fasta file containing recombinant sequences for RutR and His-lysine acetyltransferases for analysis of in vitro assays[140]. False discovery rates on protein and peptide spectrum match level were estimated by the target-decoy approach to 1%. The minimal peptide length was set to 7 amino acids and carbamidomethylation at cysteine residues was considered as a fixed modification. Acetylation of protein N-termini (for analysis of in vitro assays), methionine oxidation, and lysine acetylation were included as variable modifications. The match between runs option was disabled and label-free quantification was enabled using default settings. Further analysis of detected peptides and acetylation sequence windows was performed in Perseus[141].

Alternatively, acetylated peptides after in vitro acetylation with acetyl-phosphate were analyzed using LTQ Orbitrap XL - mass spectrometry. Stained gel-bands of protein were tryptic digested for 14 hours at 37 °C. Tryptic peptides were separated and measured online by ESI-mass spectrometry using a Proxeon Easy nLCII-system (Thermo) coupled to an Thermo Scientific LTQ Orbitrap-XL mass spectrometer. Peptides were separated chromatographically at a constant flow rate of 300 nL/min using the following gradient: Buffer A = 0.1% formic acid in Water Optima LC/MS; Buffer B = 0.1% formic acid in 99.9% acetonitrile Optima LC/MS (Fisher Scientific); 1–5% solvent B within 1 minute, 5–35% solvent B within 30 minutes, 35–75% solvent B within 4 minutes, 75–95% solvent B within 3 minutes, followed by washing and column equilibration. Data were acquired on a LTQ Orbitrap-XL Mass Spectrometer using an in-house packed analytical column 0.1 × 200 mm column with C18 Aeris Peptide (Phenomenex) at a flow rate of 300 nl per minute. For MS and MS/MS analyzes a full survey scan in the OrbitrapXL with a mass range (m/z 300-2000) and with a FT-resolution of 30,000 was followed by data dependent fragmentation experiments of the 5 most intense ions. The spectra acquired in the LTQ via CID. The parameters for the dynamic Exclusion list are as follows: Repeat count = 1, Repeat duration = 30 s, Exclusion List size=500 and Exclusion duration=30. For data base search the Mascot search engine Version: 2.6.2 (Matrix Science Ltd, London, UK) with a specific sequence database from *E. coli* including a list of common contaminants and the sequence for RutR was used. The Mascot search was carried out considering the following parameters: parent ion mass tolerance of 10 ppm, fragment ion mass tolerance of 0.50 Da, Acetylation (K) (+ 42.01056 Da) was set as variable modifications.

## Data analysis and visualization

Fiji (ImageJ 2.0.0-rc-68/1.52 h) was used for quantitative analysis of EMSAs and immunoblots[142]. Raw data from most experiments was processed using Microsoft Excel 2011. Data was visualized and statistically analyzed in GraphPad Prism 5. Fitting of data was also performed in GraphPad Prism 5. SnapGene Viewer 5.1.4.1 was employed for DNA sequence handling and generation of plasmid maps (SnapGene software from Insightful Science; available at snapgene.com). PyMOL 2.3.4 was used to generate visual representations of protein and DNA structures[143]. Adobe Photoshop 22.3.1 and Adobe Illustrator 25.4.1 was used to create figures.

## Statistics and reproducibility

All assays were performed in independent replicates as indicated resulting in similar results. For bar graphs, the standard deviations (s.d.) and mean values were depicted. No statistical method was used to predetermine the sample size. No data were excluded from the analyzes. Unpaired, two-tailed student's t-tests were performed to assess statistical significance with significance levels as indicated.

## Reporting summary

Further information on research design is available in the Nature Portfolio Reporting Summary linked to this article.

## Data availability

The X-ray structure of K52-acetylated RutR•uracil including structure factors and coordinates generated in this study have been deposited in the PDB database (http://www.rcsb.org) under accession code 6Z1B. The structural data for PDB 4JYK and PDB 1JT0 are deposited in the PDB database. Source data underlying the findings of this study are provided with this article and its Supplementary Information. Source data are provided with this paper.

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

## Acknowledgements

We thank Dr. Stefan Müller for support in proteomic experiments and Astrid Wilbrand-Hennes and Ursula Cullman for technical assistance (CECAD Proteomics Facility). We thank Linda Baldus for expert technical assistance. We thank Prof. O'Donoghue for sending the pTech-*ackRS*-tRNA$^{Pyl-opt}$ construct. We thank the staff of the beamline ID30B at the ESRF, Grenoble for their assistance during data collection. Crystals were grown in the Cologne Crystallisation facility (http://C2f.uni-koeln.de), supported from the German Research Foundation (DFG, Deutsche Forschungsgemeinschaft) grant No. INST 216/682-1 FUGG (U.B.). We thank HZB/BESSY, Berlin, and EMBL/DESY, Hamburg for continuous support in X-ray data collection. This work was supported by the German Research Foundation (DFG, Deutsche Forschungsgemeinschaft) grants No. INST 292/156-1 FUGG (M.L.), INST 292/154-1 FUGG (M.L.), LA2984-5/1 (Project: 389564084) (M.L.) and LA2984-6/1 (Project: 449703098) (M.L.).

## Author contributions

M.K. and Sa.S. performed most biochemical experiments, N.E. contributed to biochemical experiments, F.N. performed SPR experiments and analyzed the data, R.V. performed deacetylation assays, L.B., B.D., B.G., J.H. conducted expression and purification of proteins, D.A. conducted proteomics measurements, Su.S. supervised proteomics experiments, M.D. supervised SPR experiments, U.B. and G.P. supervised crystallographic data collection, K.S. supervised the bacterial strain construction and qRT-PCR experiments, M.L. initiated, designed and supervised the study, performed the crystallization and solved the structure. M.K., Sa.S. and M.L. wrote the manuscript. All authors contributed to data analysis and gave comments on writing the manuscript.

## Funding

## Competing interests

The authors declare no competing interests.
