## [Peer Review File NEW · Nature Communications]

Bacteria employ lysine acetylation of transcriptional regulators to adapt gene expression to cellular metabolismReviewer #1 (Remarks to the Author):

In this manuscript, the authors tried to elucidate the physiological role of lysine acetylation on RutR, a transcriptional regulator in *E. coli*. They found that RutR could be acetylated enzymatically (by PatA and YiaC) and non-enzymatically (by AcP), and multiple acetylation sites were identified including some new sites. Subsequently, by applying genetic code expansion, they found that acetylation at specific lysine residues (Lys52, Lys7, Lys11) affected the function of RutR both in vitro and in vivo. In addition, they elucidated the fine mechanism how acetylation of Lys52 affects the DNA binding activity of RutR through protein structural analysis. Finally, they tried to claim that bacteria apply lysine acetylation of transcriptional regulators as sensors of the cellular metabolic state. Though a new method for probing the role of lysine acetylation on protein function has been introduced, and lots of works have been done, the following concerns need to be addressed.

- 1) Since the authors already identified the enzymes or metabolite responsible for reversible lysine acetylation in *E. coli*, why not trying to increase or decrease the overall acetylation level of RutR in vivo by constructing gene knockout mutants, like many other works in this field, then test their effects on RutR function?
- 2) From the data presented, RutR has multiple acetylation sites, and several of them (like Lys52, Lys62, Lys 7 and Lys11) seem to be important for its function. Is it suitable to focus on one specific residue like Lys52? This seems to be what the new method could achieve, the authors met with problems when tried to expand the new method to double sites. Is it possible to apply the new method to multiple sites? The limitations and future optimizations of the new method should be discussed.
- 3) The title of the manuscript is inappropriate, and the conclusion of the manuscript is not sufficiently supported. To take RutR as an example, what kind of cellular metabolic state could the bacteria sense through RutR acetylation? Or more specifically, which metabolite? Uracil? Acetylation of Lys52 did not affect uracil binding to RutR. AcP? No experimental data showing that changing the bacterial cellular AcP level could affect the function of RutR.

Reviewer #2 (Remarks to the Author):

In this submission, the authors describe their study of lysine acetylation on the transcription factor RutR. They show that RutR becomes acetylated through both enzymatic and non-enzymatic mechanisms, and CobB that can deacetylate relevant RutR acetylations regardless of mechanism. This team also employ the genetic code expansion method in vivo, which to my knowledge has not been done before, but if done well here could be promising for future work. This work will represent another clear indication that acetylation is a means by which the cell can integrate metabolic state with protein function in a dynamic way.

The manuscript describes a well-designed, well-controlled, well-written, and important contribution to the merging field of bacterial protein acetylation. However, it should be improved. It has a few issues that I hope the authors take seriously.

Major comments:

1. The authors used BL21 as the vehicle for RutR purification. In this background, they either mimicked acetylation at lysine residues, or incorporated acetyllysine directly. If I understand correctly, however, BL21 possessed an intact Pta-AckA pathway and patZ gene. I want to know how the authors are account for the fact that their observed results may depend upon other acetylations that occur at physiological stoichiometry. They should purify acetyl-mimic/acetyllysine incorporated RutR's in a BL21 triple mutant lacking pta, ackA, and patZ (i.e., a strain where background acetylation is dampened). They may also consider deleting yiaC as well, considering that they showed it may have relevant activity. If they achieve the same results regarding K7/K11 and K52 then that's great. However, if things change, then clearly other acetylations elsewhere on

the protein are also relevant. If they cannot repeat these experiments, then the discussion must be expanded to reflect this possible alternative explanation of your results (see specific comments below).

2. In some experiments, the authors use IPTG to induce expression of RutR constructs acetylated at a particular residue via the genetic code expansion method. In one case, different concentrations of IPTG were used such that expression of a WT copy of RutR was balanced with the derivatives to be tested (Line 254). I would like to see this experiment repeated, including wild-type cells as a control. Furthermore, I would like to see a comparison of the amount of RutR between all three (wild-type, expressing construct with unacetylated RutR, expressing construct with K52-acetylated RutR). I would like to know whether the amount of expressed RutR is comparable to the wild-type condition (see specific comments below).

Overinterpretation:

324: This statement is too strong. It's an overinterpretation. Earlier, the authors suggested the possibility based upon their data. That is how it should be stated here. Or rather, start with that proposal and then show that it is likely true given the structural data.

Grammar:

121. Whilst the manuscript is relatively well written and organized in a logical fashion, little grammatical problems diminish readability and thus the impact of this very nice work. Some sentences are grammatically incorrect. Others have commas in very weird places. There are other types of mistakes. I highlight this early example because these small grammatical issues are consistent throughout. For improved clarity, I strongly recommend that the authors get editorial help.

157: Here's another example: As written, the sentence as written, as the qualifying phrase at the end of the sentence seems to refer to BW30270 and U65, which do not carry a FLAG-tag. Re-write with something like the following: "To this end, we inserted a FLAG-tag encoding region downstream of rutR into two distinct E. coli K12 strains: BW30270 and U65."

297: And another. This sentence should be rewritten. It is convoluted and thus not entirely clear upon a first read.

503. This sentence is also a bit off. I think the authors are trying to say that the hypothesis that the N-terminus is important for translation based on the lack of expression of the N-terminal deleted protein is supported by the observation that the genetically encoded acetyllysine variants did not express well. If so, then the authors should rewrite this sentence.

505: I would change "before, we confirmed..." to "previously, we showed that...." My initial thought was "before what?" And then I realized what the authors meant.

520: Here is an example of a comma in the wrong place. This is a common problem throughout. A little bit of editing would increase clarity.

There are other issues; these were just a few examples.

Citations:

There are several oversights and errors in referencing.

126: With reference to acetylation of CRP, the authors should also cite Davis et al 2018 PMID: 29105190.

135: References 60 and 61, while relevant to acP, are inappropriate to the actual statement. Also, reference 42 is only a commentary on reference 62. Instead, the authors should cite Kuhn et al.,

2014 PMID: 24756028 and Schilling et al., 2019 PMID: 30782634.

180. Are these citations correct? I ask because ref 65 and 67 discuss IDH and tyr-tRNA synthetase.

368: The authors should cite Kuhn 2014. This paper specifically talks about the molecular features that permit this type of reaction

387: Again, the authors should cite Kuhn.

601: Also cite Klein et al. 2007 PMID: 17545286. These authors actually measured acetyl-P. Also, they might consider citing Christensen et al. 2017 PMID: 28062462. These authors linked acetyl-P-dependent behaviors to carbon excess (due to magnesium limitation) as cells enter stationary phase.

605: Cite Wolfe 2005 PMID: 15755952.

646: Abouelfouth 2015 PMID: 25417765 also made this point.

Other comments:

67 and elsewhere. Prokaryote is actually an anachronism. It literally means not eukaryote. Many authors use prokaryote when they actually mean bacteria. This manuscript is about bacteria and not archaea. However, citation 9 has "prokaryotes" in the title so it is possible that the authors intend to include archaea but, if not, this should be corrected to bacteria.

129. "Next to" implies actual physical proximity within the genome. Consider omitting this phrase, as the 5 Gcn5-like acetyltransferases are not necessarily close within the genome.

138. To my knowledge, it has since been shown that YcgC is not a deacetylase at all, but rather a hydrolase.

183. This is true but the limitation of using evolved amber-codon recognizing tRNAs is that the protein may be expressed poorly. How did the authors account for/circumvent this problem? See major comment above regarding the necessity of a WT strain as a control for wild-type RutR levels.

190. What is the nature of this BL21 background? How did the authors account for in vivo acetylation of non-mutated sites? It is possible that, while your acetyl-mimic RutR carries the genetic code expansion acetyllysine, other non-mutated lysines on the protein may have been acetylated in vivo and are acetylated on your purified protein. For all the authors know, the proteins may all be acetylated the same way and to the same degree. What if the authors had used BL21 deleted for pta ackA and yfiQ (if not multiple KATs that target RutR)? See above major comment regarding the use of WT BL21.

195. Is this antibody specific for that exact residue? On the surface, this appears to be an unacceptable method of confirming the acetylation if the antibody binds any other acetylated lysine residues.

197. The antibody can bind to other sites, albeit with different affinities, but then it begs the question why do this experiment this way? The authors can argue that 100% of the RutR proteins are acetylated at the contrived site and then acetylated at normal stoichiometry at other sites, but if one of those other sites is of consequence and a phenotype is observed at normal stoichiometry then the phenotype is not necessarily linked to the residue they are testing. Had a strain deleted for pta ackA yfiQ yiaC been used to generate the RutR, it could be compared to RutR purified from an otherwise wild-type strain and an important comparison could have been made.

242. At this point, inclusion of Fig. 3A is disorienting, but I do really like the inclusion of this

experiment to demonstrate how the authors ensure a comparable induction of the Ack52 RutR, given the amber suppression will always be incomplete. But it might be best to draw attention to this figure when the authors are actually discussing what it demonstrates.

249. How much RutR is expressed in this strain with this construct as compared to a WT strain expressing RutR? Is there less? Are the authors concerned with the relevance of their observations given that the cell may express less or more RutR than a normal WT cell?

294. So earlier the K-to-Q wasn't good in vivo but here instead the K-to-R isn't good in vivo and the K-to-Q result is emphasized. I understand why the authors have chosen to say what they have said and do what they did but I think it is seemingly contradictory and might be confusing for a reader who is not familiar with the use of K-to-R and K-to-Q mutations. This is an issue throughout the paper. Yes, it is important to point out that the mimics are imperfect but consider communicating it differently so as not to a) undercut your own results and b) avoid confusing the reader about the legitimacy of this approach.

300. Again, the constant background conversation of whether these mimetics are adequate undercuts the results and as a reader leaves me confused as to whether I should trust results with these mimetic mutants.

344. Change to "to" not "do"

361. So then doesn't this contradict your earlier experiment where you used the antibody to confirm incorporation of the acetyllysine at the amber codon? (See comments for line 195)

403. AcP acetylates K52, a residue of consequence based on your earlier results. Is this residue acetylated by YfiQ/PatZ and YiaC?

500. I don't think I saw any K-to-A mutants in this paper, which is an oversight. It is possible that the lysine is critical, and its acetylation is of no consequence. The fact that both K-to-R and K-to-Q deviate from WT is a textbook example of when a K-to-A mutation is necessary to tease out the difference between the structural import of the lysine and its acetylation status.

503. Why is it not shown given all the other supplemental data?

525. And those reports were in error, as YcgC was shown later to be a hydrolase with some activity on lysines if I'm not mistaken. This should be mentioned. As currently stated, this line is misleading and implies that YcgC is a deacetylase.

544: Reverted has a very specific genetic meaning. I would use the term "reversed."

568: AcP is also a central metabolite that is an indicator of metabolism. it should be stated here too - with the appropriate references.

597. This fast and slow dichotomy is very intriguing and makes a lot of sense when one considers the relative abundance of a KAT versus the small molecule AcP. Could this be framed as an ancient balancing act that has been evolutionarily conserved?

Figure 2A. These EMSAs are really faint, particularly the carA ones for Ack52 and Ack150, but Ack62 and Ack95 as well. I don't necessarily disbelieve them, but they are a little difficult to assess at times. This is also true of some of the EMSAs in Figure 6A. If this is an issue that is common for the carA reporter, that should be noted in the text.

669: This happens only when carbon:nitrogen or carbon:magnesium ratios are high. Carbon accumulates, acetate is excreted, and acetyl-P accumulates.

alan wolfe

Reviewer #3 (Remarks to the Author):

This paper delivered a major point that lysine acetylation could regulate the transcriptional regulation function of RutR in response to cellular metabolic state. To validate their points, by using the genetic code expansion method which could site-specifically incorporate acetylated lysine into RutR, this study first found that AcK52 could interfere with the activity of RutR in vivo. To further explore the mechanism of how AcK52 affects the function of RutR, this paper solved the crystal structure of K52-acetylated RutR. Remarkably, they found that K52 acetylation could impair the DNA binding ability of RutR through an electrostatic and steric mechanism. In addition to this, this study also noticed an unstructured N-terminal tail could also affect RutR's DNA binding, and it contained two conserved lysine sites, K7 and K11 which were found to be acetylated in later. Different from K52, this study stated that due to the specific location of K7 and K11(N-terminal, affecting translation), only glutamine substitution could be used to investigate the acetylation of these two sites in vivo. Notably, here double acetylation of K7 and K11 could impair the RutR DNA-binding rather than their individual presence. Finally, this study offers a mode that by sensing the level of acetyl-CoA/CoA, acetyl phosphate, and NAD⁺, acetylation could affect the function of RutR via enzymic (PatZ and YiaC) or nonenzymic acetylation (acetyl phosphate), and enzymic deacetylation (CobB). In total, this paper offers an interesting story. However, I do have several concerns or suggestions.

Concerns:

1. As is mentioned in the introduction, this study postulates that lysine acetylation regulates RutR function not only in response to nitrogen supply but also other metabolic challenges, however, this study didn't test those conditions such as fuel switching and their relationship with acetyl-CoA and acetyl phosphate, these important metabolites related to acetylation.
2. Since the acetylation and deacetylation tests in this study are performed in vitro only, I am wondering if you could knock out or overexpress the acetyltransferases (PatZ and YiaC) and deacetylase (CobB) in vivo and detect the acetylation changes of RutR. Here, I just have concerns that in vitro tests can not tell the whole story of enzymic control of RutR acetylation in real cellular conditions.
3. In Figure 3B and Figure 5B, some error bar is too large, showing a high standard deviation in the original data set.
4. Although this study claims that RutR K52-acetylation cannot be mimicked by RutR mutations, figure 3D does show that K52Q has similar effects as AcK52. So, it is not very accurate to draw a conclusion that RutR mutations could not functionally mimic K52 acetylation.
5. I understand that genetic code expansion is not an ideal choice when it comes to K7 and K11 acetylation in vivo, however, I notice that in Figure 7A, the protein level of RutR K7Q/K11Q and RutR K7R/K11R are much lesser than wild type RutR. Maybe the loading control is different, but I would prefer a quantitative analysis of this figure.

Proofreading is needed.

Some examples:

Line 94 "RutR is" should be "RutR was"

Line 145 "functons" should be "functions"

Line 407 "migh" should be "might"

Reviewer #4 (Remarks to the Author):

The authors identified some additional acetylated lysine residues in the transcriptional regulator RutR, which is involved in pyrimidine and purine metabolism in *E. coli*, and found that acetylation regulates RutR transcriptional activity. They further used genetic code expansion system to generate site-specifically lysine-acetylated RutR proteins followed by crystal structure analysis. As expected, acetylation of K52 and K62 impairs RutR DNA-binding ability. They also applied this genetic code expansion system in *E. coli* to show the role of RutR acetylation in vivo. The acetylation of RutR is regulated both enzymatically (i.e. PatZ and YiaC) and non-enzymatically (i.e. AcP), and deacetylation of RutR is mediated by deacetylase CobB. Finally, they proposed a model to illustrate that lysine acetylation of RutR as sensor of the cellular metabolic state to regulate gene transcription in response to environmental changes. Overall, the manuscript is well written, experiments are rigorously performed and techniques and statistics analyses are appropriate. I enjoyed reading this paper. However, I have several concerns about this manuscript.

Major issues:

1. The authors reported that lysine acetylation of RutR adapts gene expression to the cellular metabolic state. Acetylation of RutR at K52 leads to repression of *carAB* mRNA level as shown in Figure 3B. Whether the cellular metabolic state is really changed due to RutR acetylation? For example, the synthesis of pyrimidine, purine or arginine may be weakened when RutR is acetylated on above sites.
2. RutR binds to the *carAB* promoter to activate its expression in the absence of uracil to promote de novo synthesis of pyrimidines. It means *carAB*-operon is repressed by the negative transcriptional regulator PepA in presence of uracil to favor the metabolic process for degradation of pyrimidines. When the RutR Ack52 protein was crystallised in complex with uracil, the authors showed that the acetylation at K52 does not interfere with uracil binding. From Figure 4A, the N-terminal residues of RutR also seems to be far away from uracil binding sites. What's the physiological role of RutR acetylation to regulate *carAB* expression? Based on the authors' results, it seems that the acetylation of RutR can regulate *carAB* expression regardless of concentration of uracil in bacteria.
3. The authors mentioned that *rutAG* operon is expressed under nitrogen limitation in an NtrC-dependent manner (line 83), so whether nitrogen limitation was used in all *rutAG* expression-related experiments? Moreover, what is the condition(s) for *carAB* operon expression?

Minor issues:

1. The authors presented Figure 1A-1D in Introduction section, this writing style is uncommon, so please reorganize Introduction section and Figure 1.
2. Line 125-126, the authors stated that "Such mechanisms were shown for the transcriptional regulators CRP and RcsB". Some key references should be cited here, including PMID: 26943369, PMID: 28329249, PMID: 28118511, PMID: 29899473, PMID: 36700638 and PMID: 30866760.
3. Line 253-"To obtain a similar level of acetylated and non-acetylated RutR, we adjusted the concentration of IPTG used for induction of expression (non-acetylated RutR: 10 μ M IPTG; K52-acetylated RutR: 1 mM IPTG)." While in Figure 3A, the protein level of 1000 μ M Ack52 is higher than 10 or 15 μ M non-acetylated RutR. Please explain this issue.
4. Line 419-"Both single-acetylated RutR-His6 Ack7 and Ack11 bind to both dsDNA fragments similarly to non-acetylated RutR-His6. However, EMSAs suggest reduced binding of double-acetylated RutR-His6 Ack7/11 to both dsDNA fragments." While in Figure 6, it seems single-acetylated RutR-His6 Ack7 and Ack11 impairs the ability of RutR binding to *carA* promoter, and double-acetylated RutR-His6 Ack7/11 restores the binding ability to *carA* promoter.
5. Line 510-"Although the mutation RutR K7Q alone does not result in a significant reduction of RutR", whether it means it is a poor mimic for the acetylated state.

6. Line 543- "18nalysedyzed PatZ/YiaC acetylated" should be a typo.
7. What is the conversation of lysine residues (K7, 11, 52, 62, 95 and 150) of RutR among TetR family?
8. Is acetylation of K21, K95 or K150 involved in regulating RutR DNA-binding ability?
9. Lysine residues are targeted by a particularly high number of PTMs including acetylation, methylation, succinylation and lactylation. So whether other PTMs are involved in regulation of RutR besides acetylation?

**Point-by-point response to the reviewer's comments to the manuscript entitled "Bacteria employ lysine acetylation of**
**transcriptional regulators to adapt gene expression to the cellular metabolic state" by Kremer et al.**

We thank all reviewers for carefully reading our manuscript. We worked thoroughly on all points raised by the reviewers. For us it would be
excellent if all reviewers could be able to also see the comments of the other reviewers and our answers as some points are similar.

**Reviewer #1 (Remarks to the Author):**

We thank reviewer 1 for these constructive comments to our manuscript. We worked on these suggestions and are confident that we
concisely answered all points of concerns.

**Point 1:**

Since the authors already identified the enzymes or metabolite responsible for reversible lysine acetylation in *E. coli*, why not trying to
increase or decrease the overall acetylation level of RutR *in vivo* by constructing gene knockout mutants, like many other works in this field,
then test their effects on RutR function?

Answer:

The enzymes have various substrates apart from RutR so that an overexpression/knock out of the enzymes will not reveal the function of
RutR alone. Our experimental setup has the advantage that we are able to access RutR acetylation even at the resolution of a single
acetylation site without interfering with processes evoked by alteration of the acetylation state of other proteins that are substrates for
YfiQ/PatZ, YiaC and/or CobB.

**Point 2:**

From the data presented, RutR has multiple acetylation sites, and several of them (like Lys52, Lys62, Lys 7 and Lys11) seem to be important
for its function. Is it suitable to focus on one specific residue like Lys52? This seems to be what the new method could achieve, the authors
met with problems when tried to expand the new method to double sites. Is it possible to apply the new method to multiple sites? The
limitations and future optimizations of the new method should be discussed.

Answer:

The reviewer is right in saying that the acetylation sites on Lys52, Lys62, Lys7 and Lys11 are all important for RutR function. However,
acetylation of these lysine side chains is regulated by different mechanisms. While acetylation at K52 and K62 is driven non-enzymatically
by acetyl phosphate that gradually accumulates, (simultaneous) acetylation at K7 and K11 in the RutR N-terminus is driven enzymatically
by the lysine acetyltransferases YfiQ/PatZ and YiaC. These different modes of regulation of those acetylation sites makes it necessary to
individually study their impact on RutR function. Of course *in vivo* conditions might arise under which acetylation at several sites occur
resulting in an additive effect on DNA-binding. However, as we describe here all acetylations at the mentioned lysine impair RutR DNA-
binding. They all have similar consequences but occur either enzymatically catalyzed or by increase of acetyl phosphate.

The fact that we were not able to use the genetic code expansion concept (GCEC) *in vivo* to simultaneously incorporate acetyl-lysine at
positions K7 and K11 is due to the position of these sites in the RutR N-terminus rather than problems with the method to incorporate acetyl-
lysine at two sites. It is known that alterations in the N-terminus of TetR-family proteins affect the amount of soluble protein obtained by
affecting mRNA stability resulting in reduced protein levels (Berens *et al.* (1992) *JBC* 267: 1945-1952). We also observed, that deletion of
the N-terminal residues in RutR Δ 1-12 and also the RutR carrying mutations in the N-Terminus, i.e. K7Q or K7R and K11Q or K11R or the
double mutants thereof, result in a lower protein level compared to RutR WT supporting that modulation of RutR N-terminus affects the
protein level (see Fig. 7 in the revised manuscript). Generally, the efficiency to incorporate acetyl-lysine at multiple sites can be improved by
using *E. coli* strains that carry a genomic deletion of release factor 1 (RF1), which was shown to be non-essential in *E. coli* (Johnson *et al.*
(2012) *ACS Chem. Biol.* 7: 1337-1344). Moreover, the amber stop codon (UAG) is the least used stop codon in *E. coli*. Strains of *E. coli* are
available in which all endogenous UAG stop codons are reassigned to UAA to avoid incorporations at unwanted sites. Notably, in our hands
this is not necessary to express lysine-acetylated proteins as we did not observe that cells, which do not carry a genomic alteration of UAG
to UAA show an impaired growth. However, we found that the yield of proteins double-acetylated at two distinct sites is lower compared to
single acetylated or non-acetylated proteins. This could be improved by genomic deletion of release factor 1 (RF1) and/or by whole genome
alteration of stop codons UAG to UAA. **We included a paragraph in the discussion of the revised manuscript explaining these points:**

**From line 683:**

"For the first time, we incorporated acetyl-lysine into a protein in *E. coli in vivo* applying the GCEC. Incorporation of AcK52 in RutR shows
that it modulates RutR transcriptional regulator activity. Notably, we were not able to incorporate acetyl-lysine at the N-terminus *in vivo*,
neither AcK7, AcK11 nor both simultaneously. Moreover, the protein yield was even lower for the K7Q, K7R, K11Q, K11R and the double
mutants thereof when expressed *in vivo* in *E. coli* U65 (Fig. 7A). To improve the incorporation efficiency of unnatural amino acids (UNAA) at
single and multiple sites by the GCEC *E. coli* strains were developed which carry genomic deletion of release factor 1 (RF1) or in which all
amber stop codons are reassigned to ochre¹⁻³. Moreover, evolved amber suppressor tRNAs were reported that substantially increase the
incorporation efficiency⁴. In our study, however, the lower RutR protein yield *in vivo* of RutR modified at residues in the N-terminus is likely
not due to a low incorporation efficiency or production of premature translational truncation products since we obtain RutR AcK7, AcK11 and
AcK7/11 and the respective K to Q and K to R mutant proteins when expressed recombinantly in *E. coli* BL21(DE3). Therefore, other
mechanisms such as impaired mRNA stability might result in the observed lower or even absent protein levels as described also for the
related transcriptional regulator TetR⁵."

**Point 3:**

3) The title of the manuscript is inappropriate, and the conclusion of the manuscript is not sufficiently supported. To take RutR as an example,
what kind of cellular metabolic state could the bacteria sense through RutR acetylation? Or more specifically, which metabolite? Uracil?
Acetylation of Lys52 did not affect uracil binding to RutR. AcP? No experimental data showing that changing the bacterial cellular AcP level
could affect the function of RutR.

Answer:

To resolve this lack of clarity, we will explain what we mean with sensing of the metabolic state it in the subsequent section. **We also rewrote**
**the abstract in the revised manuscript to make this point clearer:**

**Abstract:**

**From line 56:**

"By detecting acetyl-CoA, NAD⁺ and acetyl-phosphate levels, bacteria apply lysine acetylation of transcriptional regulators to sense the
cellular metabolic state to directly adjust gene expression programs to rapidly change environmental conditions."

Acetyl-CoA/NAD⁺:

The metabolic state is sensed by the enzymes catalyzing acetylation and deacetylation of lysine side chains. Lysine acetyltransferases
(KATs) such as YfiQ/PatZ and YiaC use the central molecule of metabolism, acetyl-CoA, as an acetyl-group donor for acetylation and sirtuin
deacetylases such as CobB use NAD⁺ as stoichiometric co-substrate for deacetylation. KATs and sirtuins are sensors of the cellular
metabolic state by being dependent on these important molecules, i.e. acetyl-CoA and NAD⁺, for acetylation and deacetylation of their
substrates thereby affecting their functionalities and adapting these to the cellular metabolic state. It was reported that KAT activity depends
on the acetyl-CoA/CoA ratio rather than the acetyl-CoA level alone as acetyl-CoA and CoA bind with similar affinity to KATs. As an example,
enzymatic acetylation of substrates is favored in cells, in which high intracellular concentrations of acetyl-CoA (low concentrations of CoA)
and low concentrations of NAD⁺ are present.

Uracil:

Uracil binding to RutR affects the capacity of RutR to bind to DNA and therefore impairs DNA-binding. Mechanistically, binding of uracil to
the ligand-binding domain stabilizes RutR in a conformation that is incompatible with DNA-binding. DNA-binding results in dissociation of
uracil from RutR, i.e. binding of RutR to DNA and to uracil are mutually exclusive. It depends on the cellular uracil concentration to shift the
equilibrium to the RutR conformation that is incompatible with DNA-binding. **We performed additional experiments to show the impact**
**of uracil on binding of acetylated and non-acetylated RutR to DNA (see Supp. Fig. S14).** These data show that both, uracil binding and
RutR acetylation, are affecting RutR-DNA-binding by independent mechanisms, i.e. they have an additive effect. We included this data in
the revised manuscript (Supp. Fig. S14).

Acetyl phosphate:

Acetyl phosphate was reported to be the major driver for non-enzymatic acetylation in bacteria (Weinert *et al.* (2013) *Mol Cell.* 51:265-272).
A gradual increase of acetyl phosphate results in non-enzymatic acetylation at specific acetylation sites. It was shown that there is a good
overlap of sites acetylated by acetyl phosphate *in vitro* and *in vivo* (Weinert *et al.* (2013) *Mol Cell.* 51:265-272). It was reported that
intracellular levels of acetyl phosphate can accumulate up to millimolar concentrations corresponding to the concentrations we applied in our
*in vitro* assays. Protein sequences might be evolved for being prone for non-enzymatic acetylation. As an example, presence of lysine side
chains in poly-basic patches favor non-enzymatic acetylation due to the decrease in the reactivity of lysine side chains by lowering the side
chain's pK_a value. The pH-value can affect the reactivity of lysine side chains and by this also the acetylation is affected by acetyl phosphate.
To this end, we used concentrations of acetyl phosphate and a pH value that are physiologically relevant. Therefore, we are confident that
these results represent the physiological situation. We show that lysine side chains in RutR, which are non-enzymatically acetylated by acetyl
phosphate, i.e. K52 and K62, abolish RutR-DNA-binding. By using the GCEC method and site-specifically acetylated RutR we can concisely
show that this effect is due to acetylation at individual lysine side chains.

**Reviewer #2 (Remarks to the Author):**

We are really grateful to Prof. Wolfe for carefully reading our manuscript and for his constructive comments. As Prof. Wolfe is an expert on
lysine acetylation in bacteria since many years, we are really pleased that he is enthusiastic about our manuscript, which shows how bacteria
sense the cellular metabolic state by lysine acetylation and translate this directly to control transcriptional regulators that compile the signal
into gene expression. As we observe high conservation of several lysines in RutR also in other TetR-related transcriptional regulators we
postulate that post-translational lysine acetylation is a general mechanism to adjust gene expression to the cellular metabolic state. We
worked on all points he suggested and think that our manuscript did strongly improve through this revision. We hope that Prof. Wolfe is
satisfied with our additional work, which we included into the revised manuscript.

**Point 1:**

The authors used BL21 as the vehicle for RutR purification. In this background, they either mimicked acetylation at lysine residues, or
incorporated acetyllysine directly. If I understand correctly, however, BL21 possessed an intact Pta-AckA pathway and patZ gene. I want to
know how the authors are account for the fact that their observed results may depend upon other acetylations that occur at physiological
stoichiometry. They should purify acetyl-mimic/acetyllysine incorporated RutR's in a BL21 triple mutant lacking pta, ackA, and patZ (i.e., a
strain where background acetylation is dampened). They may also consider deleting yiaC as well, considering that they showed it may have
relevant activity. If they achieve the same results regarding K7/K11 and K52 then that's great. However, if things change, then clearly other
acetylations elsewhere on the protein are also relevant. If they cannot repeat these experiments, then the discussion must be expanded to
reflect this possible alternative explanation of your results (see specific comments below).

Answer:

*E. coli* BL21 is a bacterial strain that was developed for recombinant protein expression, i.e. it carries genomic deletions of genes encoding
for proteases such as Lon, genes for DNA-recombination, etc. To this end, the results obtained by studies using BL21 will not represent the
*in vivo* situation. Importantly, *E. coli* BL21 was only used to recombinantly express RutR acetylated proteins. No *in vivo* experiments were
performed with *E. coli* BL21. These proteins expressed in *E. coli* BL21 were subsequently purified and used for the *in vitro* studies. For *in*
*in vivo* studies we selected more original *E. coli* strains such as *E. coli* U65 or *E. coli* BW30270. Our data clearly show no further post-
translational modifications (PTMs) on these recombinantly expressed and purified proteins -at least not at detectable, relevant

stoichiometries. For all proteins, as a quality control, we conducted sensitive ESI-MS experiments to determine their molecular weights.
These data show that there is only one molecular species present, i.e. if there were molecules carrying further lysine acetylations or other
PTMs at relevant stoichiometries, these would be strongly underrepresented at stoichiometries that cannot be detected by ESI-MS. To this
end, from our point of view there is no need to purify the proteins in an *E. coli* strain carrying deletions of *pta*, *ackA*, *patZ*, *yiaC* and *cobB*.
Notably, for the expression of acetylated proteins we added nicotinamide to the culture medium to inhibit the deacetylase CobB. To further
ensure that particularly the acetylations in the RutR N-terminus, which are highly efficiently deacetylated by CobB, are not deacetylated.
During expression we either increased the nicotinamide concentration during cultivation or used an *E. coli* BL21 $\Delta cobB$ deletion strain. Our
data obtained with the purified proteins clearly show the presence/absence of the acetylation at the individual lysine side chains. Apart from
that we always compared the impact of acetylation of the purified RutR protein to the non-acetylated RutR wildtype protein, which would
carry the same -very low stoichiometry- PTMs. Besides, if there were further PTMs present on the proteins, they would also be present in
the cells since proteins are biomolecules and accessible for chemical modification within cells. We showed the site-specific incorporation of
acetyl lysine by 1. DNA-sequencing to confirm presence of the amber stop codon, 2. ESI-MS to confirm correct molecular weights of the
proteins, 3. Immunoblotting by an anti-acetyl lysine antibody. **To confirm the site specific incorporation on the protein level for the**
**recombinantly expressed proteins, we performed additional LC-MS/MS experiments (Supp. Fig. S12). We wrote in the revised**
**manuscript:**

**From line 198:**

"Therefore, additional molecular mass determination by electrospray-mass spectrometry (ESI-MS) was performed and showed the correct
molecular weights of all RutR proteins and LC-MS/MS experiments confirm the site-specific acetyl-lysine incorporation (Supp. Fig. S2 and
S12)."

Concerning the other point, that was raised by Prof. Wolfe concerning the possibility that several acetylations in RutR might be relevant in
different physiological situations we can say from our data at least that we only have two sites that are enzymatically catalyzed, i.e. K7, K11
and K7,K11, and several sites that can be acetylated non-enzymatically by acetyl phosphate, i.e. K7, K11, K21, K52, K95 and K150
(described earlier). He is right in saying that different combinations of acetylations in varying stoichiometries might be possible. However, all
acetylations of functional importance impair RutR-DNA binding suggesting that these have similar physiological consequences. **We**
**performed additional ITC experiments to analyze if acetylation at K21, K95 and K150 affects RutR DNA-binding. Our data show that**
**these acetylations do not affect RutR binding to neither box_{carA} nor box_{rutA} DNA (see Supp. table S3 and Supp. Fig. S15).** We also
performed additional EMSA assays to see how presence of uracil and lysine acetylation affect RutR-DNA-binding. We found that the
presence of uracil impairs DNA-binding and acetylation at K52 and K7, K11 is additive suggesting that binding to uracil and acetylation act
via different mechanisms to impair DNA-binding.

We wrote:

**From line 734:**

"For RutR, we show that the sites of functional importance can be regulated enzymatically by the KATs PatZ/YfiQ and YiaC, i.e. K7, K11 and
K7/11, with the potential to accumulate to high stoichiometries, and non-enzymatically amongst others at the functionally important sites K7,
K11, K52 and K62 by acetyl-phosphate likely only gradually increasing to higher stoichiometries. Importantly, these lysine acetylations could
occur simultaneously at different sites and stoichiometries affecting the outcome."

**Point 2:**

In some experiments, the authors use IPTG to induce expression of RutR constructs acetylated at a particular residue via the genetic code
expansion method. In one case, different concentration of IPTG were used such that expression of a WT copy of RutR was balanced with
the derivatives to be tested (Line 254). I would like to see this experiment repeated, including wild-type cells as a control. Furthermore, I
would like to see a comparison of the amount of RutR between all three (wild-type, expressing construct with unacetylated RutR, expressing
construct with K52-acetylated RutR). I would like to know whether the amount of expressed RutR is comparable to the wild-type condition
(see specific comments below).

Answer:

We used the *E. coli* strain U65 $\Delta rutR$ $P_{carA-lacZ}$ U65 to assess the impact of RutR K52-acetylation on its transcriptional regulator activity.
As requested by Prof. Wolfe, we conducted additional experiments comparing expression of endogenous WT RutR, of ectopically expressed
unacetylated RutR and of ectopically expressed K52-acetylated RutR (Supp. Fig. S13A in revised manuscript). We compared the
transcriptional regulator activity of RutR by measuring the mRNA levels of reported RutR target genes by RT-qPCR and by β -galactosidase
activity, which is expressed under the control of a *carAB* promoter. Notably, for ectopically expressed RutR WT and K52-acetylated RutR
we observe highly corresponding mRNA levels suggesting that these effects show a high degree of comparability. However, we cannot say
how the mRNA level really translates into protein level. Nonetheless, for the identified effect of RutR on expression of *carA* and *lacZ*, which
is under control of *carAB* promoter, we see that K52-acetylation does bring the β -galactosidase expression almost back to the level observed
for *E. coli* strain U65 $\Delta rutR$ $P_{carA-lacZ}$. This shows that acetylation at K52 almost completely switches off RutR function, i.e. acting as indirect
activator for *carAB* expression. The K52-acetylation completely abolishes binding (affinity goes down from app. 30 nM to not detectable)
towards *carAB* DNA and this effect is independent of the amount of protein (assuming no indirect effects dependent on the amount). Even if
there was more RutR protein in the ectopically expressed samples compared to endogenously expressed RutR protein, this complete loss
of DNA-binding would always be observed independent of the protein level. The situation were different if K52-acetylation were to increase
DNA-binding, for which a concentration dependent effect could be present. Along this line, for acetylation sites that only slightly decrease
the affinity towards DNA, the real effect at endogenous RutR levels were even be larger under physiological conditions if ectopic RutR
expression resulted in more protein compared to endogenous expression. Therefore, we postulate that the effect observed is physiologically
important. Another experiment showing the influence of the endogenous level of RutR is shown in Fig. 3C, which presents the impact of the
mimetics K52Q and K52R mutants. In this experiment we see that the endogenous RutR level (WT-empty; wildtype cells transformed with
empty vector to ensure same growth conditions for all samples compared) results in a detectable expression of β -galactosidase and,

importantly, the ectopic expression of RutR WT in the $\Delta rutR$ background results in a statistically significant reduction of β -galactosidase
activity. This represents the control to show the impact of presence/absence of RutR on β -galactosidase expression on the endogenous
level. The ectopic expression of RutR wildtype results in an app. 1.8-fold increase in β -galactosidase activity compared to the endogenously
expressed RutR, showing that the strong increase observed in the mRNA level by qPCR is not transferred to the protein level to the same
extend.

As suggested by Prof. Wolfe we performed experiments to compare the levels of endogenously expressed RutR, ectopically expressed RutR
WT and ectopically expressed K52-acetylated RutR. To this end, we used *E. coli* U65 *PcarA lacZ*, in which the endogenous *rutR* gene was
deleted $\Delta rutR$, i.e. *E. coli* U65 $\Delta rutR$ *PcarA lacZ*. We constructed further plasmids encoding for RutR and RutR Ack52 with C-terminal FLAG-
tag. This allows us to detect all RutR proteins, i.e. the ectopically expressed RutR-FLAG WT, RutR-FLAG Ack52 and the endogenous RutR-
FLAG, by immunoblotting with anti-FLAG antibody to compare the expression levels.

We compared the expression with ectopically expressed RutR WT and K52-acetylated RutR adding different concentrations of IPTG (Supp.
Fig. S13A). To this end, we transformed *E. coli* U65 $\Delta rutR$ *PcarA lacZ* with pRSF-Duet1/*ackRS3/pylT* encoding for RutR WT or K52-
acetylated RutR with C-terminal FLAG-tag. We observed that the K52-acetylated protein is expressed at similar level compared to non-
acetylated RutR wildtype (Supp. Fig. S13A). This is supported by statistical analyses that showed that comparing pairs of RutR WT and K52-
acetylated RutR at same IPTG concentrations are not different in a statistically significant form (Supp. Fig. S13A; Source Data Table).
Moreover, these experiments revealed that the addition of increasing concentrations of IPTG resulted in an increase of ectopically RutR
protein level (Supp. Fig. S13A, right panel). The underlying mechanism is not known and would need additional investigations, which are
not within the scope of this manuscript. *E. coli* U65 does not carry the DE3 lysogen of phage T7 so that there should no T7-DNA-dependent
RNA polymerase be present, that is needed for expression from T7 promoter present at pRSF-Duet1.

Moreover, we also compared these protein levels to the endogenous RutR protein level by comparing it to *E. coli* U65 *rutR-FLAG*. We can
detect RutR protein at the endogenous level, however, the protein level is much lower compared to the ectopically expressed RutR WT and
RutR Ack52, which are expressed at a similar level.

These results suggest that it is important to compare only those samples that show a similar expression level, i.e. only the ectopically
expressed RutR proteins with each other but not ectopically expressed RutR to RutR expressed at endogenous level.

For ectopically expressed RutR Ack52 at the different IPTG concentrations and for the empty vector controls, we observe a statistically
significant reduction in β -galactosidase activity compared to RutR WT (Supp. Fig. S13B). In contrast, ectopically expressed RutR Ack52
does not result in statistically significant differences in β -galactosidase activity compared to the empty vector controls supporting that
acetylation of RutR at K52 completely switches off RutR-transcriptional regulator activity (see Supp. Fig. S13B).

These results have also implications for the experiment shown in Fig. 7, in which the impact of the N-terminal acetylations is shown by
assessing the K7Q, K7R, K11Q, K11R mutants and the double mutants thereof. We found that mutating residues in the RutR N-terminus or
incorporating acetyllysine in the RutR N-terminus results in reduced protein levels. This is likely due to altered mRNA stability rather as
described also for TetR (see answer to point 7; Berens *et al.* (1997) *JBC* 267: 1945-1952). We observed a similar protein level for groups of
RutR proteins: group 1: RutR WT and RutR K52Q, group 2: RutR K7/K11 to Q and K7/K11 to R single mutants, group 3: RutR K7/K11 to Q
and R double mutants. We observe that RutR K52Q switches-off RutR transcriptional regulator activity compared to RutR wildtype (Fig. 7,
right panel, black). The single mutants of K7 and K11 in the RutR N-terminus are not altered in a statistically significant way (Fig. 7, right
panel, red). However, the double mutants in the RutR N-terminus, i.e. K7Q/K11Q and K7R/K11R, are altered statistically significant (Fig. 7,
right panel, blue). We only show statistical evaluation within each of the three groups in the revised manuscript as we observed these different
protein levels. The data showed clearly that acetylation is affecting RutR transcriptional regulator activity. As a further support, we performed
additional ITC experiments, which showed that the double mutant K7Q/K11Q (acetylation mimetic mutant) affects binding towards both DNA-
fragments while K7R/K11R (charge-conserving mutant) binds as RutR wildtype to both DNA-fragments (Supp. Fig. S15, Supp. Table S3).

**Point 3:**

Overinterpretation:

324: This statement is too strong. It's an overinterpretation. Earlier, the authors suggested the possibility based upon their data. That is how
it should be stated here. Or rather, start with that proposal and then show that it is likely true given the structural data.

**Answer:**

We agree with Prof. Wolfe and rephrased the paragraph to be a bit more circumspect:

**From line 369:**

"These regions show a high degree of structural similarity represented by an r.m.s.d. value for the C-alpha atoms of 0.83 Å. In QacR•IR1,
the homologous K36 is in hydrogen bond distance of 2.6 Å with N7 of a guanine base (Fig. 4B, closeup). This contributes to sequence
specificity in binding of QacR to DNA⁶. The superposition suggests that lysine acetylation at K52 in RutR exerts an electrostatic as well as
steric mechanism to abolish DNA-binding supporting our data shown above. More precisely, K52-acetylation would abrogate the hydrogen
bond towards N7 of the guanine base and it is incompatible with DNA-binding sterically clashing with DNA-bases (Fig. 4B, closeup)."

**Point 4:**

121. Whilst the manuscript is relatively well written and organized in a logical fashion, little grammatical problems diminish readability and
thus the impact of this very nice work. Some sentences are grammatically incorrect. Others have commas in very weird places. There are
other types of mistakes. I highlight this early example because these small grammatical issues are consistent throughout. For improved
clarity, I strongly recommend that the authors get editorial help.

**Answer:**

We thank Prof. Wolfe for giving this constructive feedback. We assessed the manuscript for grammatical issues.

We wrote here:

**From line 121:**
"In eukaryotes it is known since the 1960s that acetylation of lysine side chains in the unstructured, flexible N-terminal histone
tails affects RNA synthesis⁷."

**Point 5:**
157: Here's another example: As written, the sentence as written, as the qualifying phrase at the end of the sentence seems to refer to
BW30270 and U65, which do not carry a FLAG-tag. Re-write with something like the following: "To this end, we inserted a FLAG-tag encoding
region downstream of *rutR* into two distinct *E. coli* K12 strains: BW30270 and U65."

Answer:
Corrected:

**From line 158:**
"To this end, we genomically inserted a DNA sequence encoding for a FLAG-tag downstream of *rutR* in two distinct *E. coli* K12 strains:
BW30270 and U65 (see Supp. Table 1 for summary of *E. coli* strains used in this study)."

**Point 6:**
297: And another. This sentence should be rewritten. It is convoluted and thus not entirely clear upon a first read.

Answer:
We rewrote the paragraph to not undercut our own results.

**From line 264:**
"For RutR K52Q, we observed a β -galactosidase activity similar to the empty vector control suggesting that in this assay under these
experimental conditions RutR K52Q is a reliable mimic for RutR K52-acetylation and that charge neutralization constitutes an important
mechanism to switch-off RutR transcriptional regulator activity (Fig. 3A)."

**Point 7:**
503. This sentence is also a bit off. I think the authors are trying to say that the hypothesis that the N-terminus is important for translation
based on the lack of expression of the N-terminal deleted protein is supported by the observation that the genetically encoded acetylysine
variants did not express well. If so, then the authors should rewrite this sentence.

Answer:
We observed that the N-terminally RutR-His₆ Δ 1-12 is not expressed in *E. coli* U65 Δ *rutR P_{carA}-lacZ*. Along this line, also the RutR-His₆
Ack7, Ack11 and double-acetylated Ack7/11 is not expressed. To this end, we analyzed the single RutR-His₆ mutants K7Q, K11Q, K7R,
K11R and the double mutants RutR-His₆ K7Q/K11Q and K7R/K11R. We also see a lower expression for the single mutants and double
mutants compared to RutR-His₆ wildtype and K52Q. The double mutants RutR-His₆ K7Q/K11Q and K7R/K11R show a lower expression
compared to the single RutR-His₆ mutants K7Q, K11Q, K7R, K11R. All of these results suggest that the N-terminus of RutR-His₆ affects the
translation efficiency. For TetR, a related transcription regulator, it was shown that the sequence encoding for the N-terminal residues, affects
the mRNA stability (Berens *et al.* (1997) *JBC* 267: 1945-1952). A similar effect might be present in RutR but this would need further
investigation. If this is of any physiological significance is also not clear. As we were not able to incorporate acetyl lysine at position K7 and/or
K11 in RutR, we used the K to Q and K to R mutants. In the revised manuscript we only performed statistical comparisons between proteins
of similar expression levels, i.e. group 1 (wildtype and K52Q), group 2 (single mutants), group 3 (double mutants). These data showed that
K52Q results in statistically significant reduction in β -galactosidase activity compared to RutR wildtype. Moreover, K7Q/K11Q (acetylation
mimic) is statistically reduced compared to K7R/K11R (non-acetylated state, charge-conserving). We performed additional ITC experiments
to show that the RutR-His₆ K7Q/K11Q (acetylation mimic) and K7R/K11R (non-acetylated state, charge-conserving) double mutants behave
similar to the acetylated and non-acetylated RutR-His₆ concerning binding to *box_{rutA}* and *box_{carA}*, respectively (Supp. Fig. S15, Supp. Table
S3). Therefore, we draw the conclusion that acetylation in the RutR N-terminus affects its transcriptional regulator activity. See also our
answer to point 2.

We rewrote the sentence and also the paragraph as suggested:

**From line 553:**
"The N-terminally mutated RutR-proteins followed the same trend observed also for the N-terminally acetylated RutR-protein. While we
obtained protein for the N-terminal K to Q and K to R single mutants (group 2, Fig. 7A, left panel, red bars) the protein level was reproducibly
lower compared to RutR WT and RutR K52Q (group 1, Fig. 7, left panel, black bars). For the double-mutants RutR K7Q/K11Q and K7R/K11R
(group 3, Fig. 7, left panel, blue bars) the protein level was even more reduced compared to the single mutated RutR. Notably, for the N-
terminal deletion mutant RutR Δ 1-12 we could not detect any protein upon expression in *E. coli* U65 Δ *rutR P_{carA}-lacZ* (Fig. 7)."

**Point 8:**
505: I would change "before, we confirmed..." to "previously, we showed that...." My initial thought was "before what?" And then I realized
what the authors meant.

Answer:
Rephrased accordingly.

**Point 9:**
520: Here is an example of a comma in the wrong place. This is a common problem throughout. A little bit of editing would increase clarity.
There are other issues; these were just a few examples. Citations: There are several oversights and errors in referencing.

Answer:
We corrected commas and also checked references/citations as recommended by Prof. Wolfe.

**Point 10:**
126: With reference to acetylation of CRP, the authors should also cite Davis et al 2018 PMID: 29105190.

Answer:
Reference was included.

**Point 11:**
135: References 60 and 61, while relevant to acP, are inappropriate to the actual statement. Also, reference 42 is only a commentary on
reference 62. Instead, the authors should cite Kuhn et al., 2014 PMID: 24756028 and Schilling et al., 2019 PMID: 30782634.

Answer:
References were corrected and the references included.

**Point 12:**
180. Are these citations correct? I ask because ref 65 and 67 discuss IDH and tyr-tRNA synthetase.

Answer:
These references are correct. Both describe application of the genetic code expansion concept to produce site-specifically lysine acetylated
proteins, either for *E. coli* isocitrate dehydrogenase or *E. coli* tyrosyl-tRNA-synthetase.

**Point 13:**
368: The authors should cite Kuhn 2014. This paper specifically talks about the molecular features that permit this type of reaction

Answer:
Reference was included.

**Point 14:**
387: Again, the authors should cite Kuhn.

Answer:
Reference was included.

**Point 15:**
601: Also cite Klein et al. 2007 PMID: 17545286. These authors actually measured acetyl-P. Also, they might consider citing Christensen et
al. 2017 PMID: 28062462. These authors linked acetyl-P-dependent behaviors to carbon excess (due to magnesium limitation) as cells enter
stationary phase.

Answer:
Thank you for pointing this out. References were included.

**Point 16:**
605: Cite Wolfe 2005 PMID: 15755952.

Answer:
Reference was included.

**Point 17:**
646: Abouelfouth 2015 PMID: 25417765 also made this point.

Answer:
Reference was added.

**Point 18:**
67 and elsewhere. Prokaryote is actually an anachronism. It literally means not eukaryote. Many authors use prokaryote when they actually
mean bacteria. This manuscript is about bacteria and not archaea. However, citation 9 has "prokaryotes" in the title so it is possible that the
authors intend to include archaea but, if not, this should be corrected to bacteria.

Answer:
Prof. Wolfe is right, we corrected it and replaced prokaryotes with bacteria.

**Point 19:**
129. "Next to" implies actual physical proximity within the genome. Consider omitting this phrase, as the 5 Gcn5-like acetyltransferases are
not necessarily close within the genome.

Answer:
Phrase is omitted and sentence rephrased in the revised manuscript:

"Recently, it was reported that in *E. coli*, besides the known KAT PatZ/YfiQ, four additional Gcn5-like acetyltransferases encoded in the
genome function as epsilon-lysine acetyl transferases: RimI, YjaB, YiaC and PhnO⁸⁻¹³."

**Point 20:**
138. To my knowledge, it has since been shown that YcgC is not a deacetylase at all, but rather a hydrolase.

Answer:
Prof. Wolfe is right. In fact, it was our lab that showed that YcgC has no deacetylase activity as proposed by Tu *et al.* (Kremer *et al.* (2018)
eLife doi: 10.7554/eLife.37798). However, to my knowledge it was neither shown to be a deacetylase nor was it shown to have hydrolase
activity. Tu *et al.* showed that mutation of a serine residue abolished the potential deacetylase activity. Tu *et al.* measured deacetylase
activity, i.e. they performed Michaelis-Menten kinetics, indirectly by assessing autoproteolytic cleavage of RutR as a readout. But our lab
showed that rather a proteolytic contamination of the samples explains the activity found by Tu *et al.*. So, YcgC is neither a deacetylase nor
a hydrolase, i.e. specifically not a protease.

**Point 21:**
183. This is true but the limitation of using evolved amber-codon recognizing tRNAs is that the protein may be expressed poorly. How did
the authors account for/circumvent this problem? See major comment above regarding the necessity of a WT strain as a control for wild-
type RutR levels.

Answer:
See our answers to point 2 and 7. We performed the experiment as suggested by Prof. Wolfe.

**Point 22:**
190. What is the nature of this BL21 background? How did the authors account for in vivo acetylation of non-mutated sites? It is possible
that, while your acetyl-mimic RutR carries the genetic code expansion acetyllysine, other non-mutated lysines on the protein may have been
acetylated in vivo and are acetylated on your purified protein. For all the authors know, the proteins may all be acetylated the same way and
to the same degree. What if the authors had used BL21 deleted for *pta ackA* and *yfiQ* (if not multiple KATs that target RutR)? See above
major comment regarding the use of WT BL21.

Answer:
See our answer to point 1. All purified proteins show a single mass peak in ESI-MS confirming the correct molecular weight of the proteins,
showing that if there were molecules post-translationally modified at different sites these would occur at very low stoichiometry, as these are
not even detectable by ESI-MS. If present in such low stoichiometries, these are clearly not affecting protein function. Additionally, all proteins
expressed in this way would carry these very low stoichiometric modifications and are therefore regarded as part of the native protein. *E.*
*coli* BL21 is a strain that is commonly used for expression of proteins.

**Point 23:**
195. Is this antibody specific for that exact residue? On the surface, this appears to be an unacceptable method of confirming the acetylation
if the antibody binds any other acetylated lysine residues.

Answer:
Prof. Wolfe is correct. The anti-acetyl-lysine antibody used here in principle is able to detect all acetylated lysine side chains independently
of the exact position (see answer to point 24). To show the site-specificity of acetyl-lysine incorporation we performed additional LC-MS/MS
experiments on the purified proteins following tryptic digest (see Supp. Fig. S12). We confirmed site specific incorporation of the
recombinantly expressed and site-specifically lysine acetylated proteins. Notably, for the RutR protein acetylated at K11 we were not able to
detect the peptide by LC-MS/MS due to the small peptide generated upon tryptic digest (8-TTGKR-12). However, for all proteins the correct
incorporation of acetyl-lysine is ensured by the placement of an amber stop codon in the *rutR*-coding sequence. All DNA constructs were of
course sequenced before use. Moreover, by ESI-MS we confirmed the correct molecular weight of all acetylated proteins. A missing
incorporation of acetyllysine would result in a translationally truncated RutR protein as the amber stop codon would result in stop of
translation. By immunoblotting we furthermore confirmed that all proteins were indeed lysine acetylated.

We added:

**From line 198:**
"Therefore, additional molecular mass determination by electrospray-mass spectrometry (ESI-MS) was performed and showed the correct
molecular weights of all RutR proteins and LC-MS/MS experiments confirm the site-specific acetyl-lysine incorporation (Supp. Fig. S2 and
S12)."

**Point 24:**
197. The antibody can bind to other sites, albeit with different affinities, but then it begs the question why do this experiment this way? The
authors can argue that 100% of the RutR proteins are acetylated at the contrived site and then acetylated at normal stoichiometry at other
sites, but if one of those other sites is of consequence and a phenotype is observed at normal stoichiometry then the phenotype is not

necessarily linked to the residue they are testing. Had a strain deleted for *pta ackA yfiQ yiaC* been used to generate the RutR, it could be
compared to RutR purified from an otherwise wild-type strain and an important comparison could have been made.

Answer:

As stated above, we show that the purified acetylated RutR proteins are not acetylated at other sites at stoichiometries that can be detected
by ESI-MS. Prof. Wolfe is right in saying that the antibody can detect different acetylation sites with different efficiency. We observed before,
that there is some sequence bias of the antibody recognizing different acetylation sites in proteins with different detection efficiencies (de
Boor *et al.* (2015) *PNAS* 112: E3679-E3688). However, we use this antibody (abcam 21623) as this is highly specific for lysine acetylation
in contrast to other available antibodies. Importantly, we cannot detect wildtype RutR with the antibody furthermore supporting that there is
no detectable acetylation at other sites.

**Point 25:**

242. At this point, inclusion of Fig. 3A is disorienting, but I do really like the inclusion of this experiment to demonstrate how the authors
ensure a comparable induction of the Ack52 RutR, given the amber suppression will always be incomplete. But it might be best to draw
attention to this figure when the authors are actually discussing what it demonstrates.

Answer:

Fig. 3A shows the expression of RutR protein, which was analyzed by RT-qPCR. That is the reason we placed it here. To make this clearer
we rearranged the paragraph, reordered the figures and describe the analysis and adjustment of the expression levels prior to the β -
galactosidase and RT-qPCR experiment. We also describe the experiment with the K52Q and K52R mutants before describing the impact
of lysine acetylation at K52. We think that this is much clearer now.

**Point 26:**

249. How much RutR is expressed in this strain with this construct as compared to a WT strain expressing RutR? Is there less? Are the
authors concerned with the relevance of their observations given that the cell may express less or more RutR than a normal WT cell?

Answer:

See answers to points 2 and 7.

**Point 27:**

294. So earlier the K-to-Q wasn't good in vivo but here instead the K-to-R isn't good in vivo and the K-to-Q result is emphasized. I understand
why the authors have chosen to say what they have said and do what they did but I think it is seemingly contradictory and might be confusing
for a reader who is not familiar with the use of K-to-R and K-to-Q mutations. This is an issue throughout the paper. Yes, it is important to
point out that the mimics are imperfect but consider communicating it differently so as not to a) undercut your own results and b) avoid
confusing the reader about the legitimacy of this approach.

Answer:

Importantly, we want to stress that from our point of view it is important to indicate that the data obtained with mimetic mutants have to be
carefully analyzed to draw appropriate conclusions. It is fine to use K-to-Q to mimic lysine acetylation and K-to-R to conserve the non-
acetylated, charge-conserving state. However, this depends strongly on the context, the assay performed and the individual site whether
these mimics are good tools to study lysine acetylation at the molecular level. As you see from our results, K-to-Q of K52 in RutR results in
a reduction of *PcarAB*-dependent *lacZ* expression to levels of the empty vector control, i.e. cells that either lack RutR or containing RutR
that is acetylated at K52. However, K-to-R does not behave as non-acetylated K52, i.e. in this case it is not a mimic for RutR WT, as it
results in a statistically significant reduction of β -galactosidase expression compared to RutR WT. However, it is K52R is also statistically
relevant increased compared to K52Q suggesting that in principle it can be used as mimic but it does not perfectly mimic the non-acetylated
state. This suggests an additional sterical component in K-to-R affecting RutR-DNA-binding that is not present in RutR WT. In contrast, for
K7 and K11, we observed that K-to-Q and K-to-R are both similar to acetylated and non-acetylated RutR, respectively. We think that it does
not undercut our results obtained by mimetic mutants for each individual case but in contrast it emphasizes the power of our experimental
approach using site-specifically lysine acetylated protein. Our data clearly show that the possibility to use mimetic mutants has to be assessed
for each individual site, i.e. for some sites it is reliable to use mimetic mutants. Moreover, our data reveal the general trend obtained for the
results with mimetic mutants showing that in principle it is possible to mimic the acetylated or non-acetylated state. However, if performing a
thorough mechanistic analysis on molecular level some of these mutants behave differently from the acetylated/non-acetylated RutR. As an
example, K52Q strongly reduces the affinity towards DNA compared to RutR WT but K52-acetylated completely abolishes DNA-binding.
This reduction in affinity in this assay is sufficient for RutR K52Q to behave as K52-acetylated RutR as the intracellular concentration of RutR
determines if this affinity reduction is sufficient to see an effect. It strongly depends on the context of the experiment performed if the mutants
can be used as reliable tools to study acetylation. We rewrote the sections on the K52Q and K52R mutants and the conclusions drawn in
terms of being reliable tools to study lysine acetylation to not undercut our own results.

We rephrased the sentence to not undercut our results:

**From line 236:**

"EMSA and ITC experiments agree in showing that RutR K52R binds with a similar nanomolar affinity to both dsDNA-fragments compared
to non-acetylated RutR WT. However, RutR K52Q, in contrast to RutR Ack52, retains some DNA-binding affinity towards both, *box_{RutA}* and
*box_{carA}* dsDNA (Supp. Fig. S3; Supp. Table S3; Source Data Table). These data suggest that with respect to DNA-binding RutR K52R can
be used in vivo to conserve the non-acetylated, positively-charged state (Supp. Fig. S3B,C). However, RutR K52Q does not completely
abolish binding to both DNA-fragments and is therefore not a perfect mimic for K52-acetylation. Further factors such as the cellular
concentration of the RutR K52Q protein determine whether it can be used to study all aspects of RutR lysine acetylation *in vivo*."

and:

**From line 264:**

"For RutR K52Q, we observed a β -galactosidase activity similar to the empty vector control suggesting that in this assay under these
experimental conditions RutR K52Q is a reliable mimic for RutR K52-acetylation and that charge neutralization constitutes an important
mechanism to switch-off RutR transcriptional regulator activity (Fig. 3A)."

**Point 28:**

300. Again, the constant background conversation of whether these mimetics are adequate undercuts the results and as a reader leaves
me confused as to whether I should trust results with these mimetic mutants.

Answer:

See answer to point 27. We rephrased the paragraph.

**Point 29:**

344. Change to "to" not "do"

Answer:

Corrected.

**Point 30:**

361. So then doesn't this contradict your earlier experiment where you used the antibody to confirm incorporation of the acetyllysine at the
amber codon? (See comments for line 195)

Answer:

This is just one possible explanation why not observing a reduction in the signal of the immunoblot for K52Q. We want to say that in principle
K52 could be enzymatically lysine acetylated in case the signal by the antibody for RutR acetylated at K52 is much weaker compared to
RutR with other acetylation sites. At this stage we did not want to exclude that K52 is enzymatically acetylated. However, the analyses of
the N-terminally deleted RutR and our LC-MS/MS data confirm that K52 is indeed not enzymatically acetylated but only non-enzymatically
by acetyl phosphate. We rephrased it in the revised manuscript:

**From line 408:**

"This observation signifies that either K52 is not acetylated by PatZ or YiaC it is acetylated at a low stoichiometry or that other sites are
stronger epitopes for the antibody and therefore are better detected (Supp. Fig. S5B). We observed before that the antibody shows strong
bias in the recognition of different acetylation sites¹⁴. However, we later clearly show that indeed K52 is not enzymatically acetylated but only
acetylated non-enzymatically by acetyl-phosphate."

**Point 31:**

403. AcP acetylates K52, a residue of consequence based on your earlier results. Is this residue acetylated by YfiQ/PatZ and YiaC?

Answer:

See answer to point 30. Our data shows that K52 is not enzymatically acetylated by neither YfiQ/PatZ, YiaC, RimI, PhnO nor YjaB but is
acetylated only non-enzymatically by acetyl phosphate.

**Point 32:**

500. I don't think I saw any K-to-A mutants in this paper, which is an oversight. It is possible that the lysine is critical, and its acetylation is of
no consequence. The fact that both K-to-R and K-to-Q deviate from WT is a textbook example of when a K-to-A mutation is necessary to
tease out the difference between the structural import of the lysine and its acetylation status.

Answer:

We did not characterize K-to-A mutants as we were not interested to analyze a pure impact of the lysine residues on RutR function or RutR
stability/structure but rather the post-translational modification of lysine side chains by acetylation. If the effect mediated by lysine acetylation
is mainly charge-neutralization, this is assessed by the K-to-Q mutants. All other effects by K-to-A mutants, such as structural effects, were
of no physiological significance as there are no corresponding mutations in RutR described. We see, that all acetylated RutR proteins behave
as non-acetylated RutR on analytical size-exclusion chromatography. i.e. they elute as dimers, suggesting that none of the lysine acetylations
interfere with the RutR structural integrity. As we do not see an additional gain of knowledge over understanding RutR acetylation by
generating and analyzing RutR K-to-A mutants we decided to not perform these experiments. We would be happy to perform these
experiments if we misunderstood the point raised by Prof. Wolfe.

**Point 33:**

503. Why is it not shown given all the other supplemental data?

Answer:

We already presented an SDS-PAGE gel in the main text that shows that RutR Δ 1-12 is not expressed in *E. coli* U65 under the experimental
conditions (see Fig. 7A).

**Point 34:**

525. And those reports were in error, as YcgC was shown later to be a hydrolase with some activity on lysines if I'm not mistaken. This
should be mentioned. As currently stated, this line is misleading and implies that YcgC is a deacetylase.

Answer:

See answer to point 20.

We stated from line 587:

"There are reports stating that the protein YcgC is a second deacetylase encoded in *E. coli*. However, we showed earlier that YcgC does not show RutR-lysine deacetylase activity^{15,16}. Moreover, it is also not a hydrolyse with protease activity¹⁵."

From our point of view it makes clear that YcgC has no deacetylase activity. We are not aware of additional reports that show such an activity for YcgC.

Point 35:

544: Reverted has a very specific genetic meaning. I would use the term "reversed."

Answer:

We replaced "reverted" by "reversed" throughout if used in sense of enzymatically removed acetylation.

Point 36:

568: AcP is also a central metabolite that is an indicator of metabolism. it should be stated here too - with the appropriate references.

Answer:

We included it in the revised manuscript and cited the appropriate references.

Point 37:

597. This fast and slow dichotomy is very intriguing and makes a lot of sense when one considers the relative abundance of a KAT versus the small molecule AcP. Could this be framed as an ancient balancing act that has been evolutionarily conserved?

Answer:

This is an interesting thought. Our thoughts about this are that acetyl phosphate is a molecule that emerged very early during evolution enabling non-enzymatic phosphorylation and acetylation of molecules in a primordial metabolism before emergence of catalysts for phosphorylation or acetylation. The gradual accumulation of acetyl phosphate results in an alteration in protein functionalities due to non-enzymatic acetylation that also gradually increases in both, number of sites and stoichiometry, of acetylation. This ensures that these protein functionalities are adjusted appropriately to the cellular metabolic state. Besides, this non-enzymatic acetyl phosphate driven acetylation might also lead to an inappropriate decline/alteration of protein-functionalities in terms of toxification that might in part be reversed by CobB. Along this line, protein sequences might have been evolved to be prone for non-enzymatic acetylation at specific sites and as a consequential adjustment of protein function by, on the one hand, gradual, slow increase of acetyl phosphate. On the other hand, enzymatic acetylation is very fast and dependent on the expression levels of KATs and the availability of cellular acetyl-CoA/CoA. It was recently reported for mammalian cells that the enzymatically catalyzed sites undergo fast acetylation upon alteration of cellular conditions such as starvation and refeeding and that these enzymatically catalyzed sites accumulate to higher stoichiometry (Baeza *et al.* (2020) *J of Proteome Res.* 19: 2404-2418).

We added to the summary of the discussion:

From line 757:

".....i.e. fast and dynamic enzymatically catalyzed (by KATs) acetylation as well as slow and gradually increasing non-enzymatic (by acetyl-phosphate) acetylation. Acetyl-phosphate emerged very early during evolution enabling non-enzymatic acetylation as well as phosphorylation of molecules¹⁷. In a primordial metabolism this enabled adjustment of cellular processes to the metabolic state before the emergence of catalysts for acetylation or phosphorylation. Notably, this non-enzymatic acetyl-phosphate driven acetylation has been conserved until today and exists side-by-side with the KAT-catalyzed acetylation allowing to precisely adjust cellular processes according to their respective kinetic requirements."

Point 38:

Figure 2A. These EMSAs are really faint, particularly the *carA* ones for Ack52 and Ack150, but Ack62 and Ack95 as well. I don't necessarily disbelieve them, but they are a little difficult to assess at times. This is also true of some of the EMSAs in Figure 6A. If this is an issue that is common for the *carA* reporter, that should be noted in the text.

Answer:

We observed for all EMSAs performed with the *carA*-promoter DNA a fainter signal compared to the *rutA*-promoter, particularly of the signal that represents the RutR-DNA complex. We assume this is due to the intercalation of ethidium bromide into the *carA*-DNA, which is weaker compared to the *rutA*-promoter DNA. It is reported that ethidium bromide intercalation is to some extent sequence dependent favoring AT-rich sequences (Galindo-Murillo *et al.* (2021) *Nucleic Acids Research* 49: 73735–3747). Comparing *carA*- and *rutA*-DNA, there are 36 AT-base pairs in *rutA*-DNA and 31 AT-base pairs in *carA*-DNA. This might explain why we observe weaker ethidium bromide signals for *carA*-DNA compared to *rutA*-DNA. When evaluating the data, it is important to judge the complex formation between RutR and DNA by inspecting the signal representing the free DNA at the bottom of the gels. With increasing amount of complex that is formed with addition of increasing concentrations of RutR/acetylated RutR the amount of this signal representing free DNA is decreasing. We included a statement in the Material and Methods section of the revised manuscript that we observed a fainter signal in EMSAs with *carA* DNA compared to *rutA* DNA that might be due to lower AT-content in *carA*-DNA compared to *rutA*-DNA:

**From line 931:**
"For the box_{carA} dsDNA we observed a fainter signal compared to box_{rutA} dsDNA. This might be due to lower AT-content in box_{carA} dsDNA
compared to box_{rutA} dsDNA¹⁸."

**Point 39:**
669: This happens only when carbon:nitrogen or carbon:magnesium ratios are high. Carbon accumulates, acetate is excreted, and acetyl-P
accumulates.

Answer:
We included these points in the revised manuscript.

We added:

**From line 746:**
"These low to moderate acetylation stoichiometries might be obtained by non-enzymatic acetylation in the late exponential growth phase or
stationary phase when acetate is excreted under conditions of high ratios of carbon:nitrogen or carbon:magnesium and as a consequence
cellular acetyl-phosphate increases¹⁹⁻²¹."

**Reviewer #3 (Remarks to the Author):**

We thank reviewer 3 for these constructive comments to our manuscript and are really pleased that he/she is enthusiastic about our work.
We carefully worked on the suggestions and are confident that we concisely answered all points of concerns.

**Point 1:**
As is mentioned in the introduction, this study postulates that lysine acetylation regulates RutR function not only in response to nitrogen
supply but also other metabolic challenges, however, this study didn't test those conditions such as fuel switching and their relationship with
acetyl-CoA and acetyl phosphate, these important metabolites related to acetylation.

Answer:
The reviewer is right. The aim of this study was to investigate how RutR function is regulated by lysine acetylation. The metabolic cellular
state is sensed by the enzymes catalyzing acetylation and deacetylation of acetylated lysine side chains. The lysine acetyltransferases
(KATs) such as YfiQ/PatZ and YiaC use the central molecule of metabolism acetyl-CoA as an acetyl-group donor for the acetylation and
sirtuin deacetylases such as CobB use NAD⁺ as stoichiometric co-substrate for their deacetylase activity. Thereby, KATs and sirtuins are
sensors of the cellular metabolic state by being dependent on these important molecules, i.e. acetyl-CoA and NAD⁺, for acetylation and
deacetylation of their substrates such as RutR, thereby affecting protein functionalities and adapting it to the cellular metabolic state. We
observed that acetylation of RutR is catalyzed enzymatically by YfiQ/PatZ and YiaC, non-enzymatically by acetyl phosphate and
deacetylation is catalyzed by the NAD⁺-dependent sirtuin deacetylase CobB. Our data clearly show that KATs act efficiently with a fast
kinetics acetylating RutR at the N-terminal lysines K7 and K11, while K52 and K62 in the helix-turn-helix motif and ligand-binding domain,
respectively, are acetylated non-enzymatically by acetyl phosphate. It is known that acetyl phosphate gradually increases intracellularly with
a slower kinetics compared to KAT-catalyzed acetylation and is high under conditions of carbon overflow metabolism under which conditions
acetate is excreted and acetyl phosphate accumulates. Under these conditions millimolar concentrations of acetyl phosphate can be reached.
Our experiments were conducted under physiologically relevant conditions, i.e. pH 7.5 and millimolar concentrations of acetyl phosphate. It
is reported that for sites acetylated by acetyl phosphate there is a huge overlap in sites identified *in vivo* and *in vitro* (Weinert *et al.* (2013)
*Mol Cell.* 51:265-272). Therefore, we are confident that the results obtained are physiologically relevant. The strength of our study is that we
apply a system that allows us to analyze the outcome of RutR-acetylation on a resolution of the individual acetylation site. If performing
assays under conditions such as fuel switching it is impossible to specifically ascribe an outcome to an individual acetylation event on RutR
as this results in various alterations, i.e. gene expression programs, acetylations and further PTMs at other proteins, etc.

**Point 2:**
Since the acetylation and deacetylation tests in this study are performed *in vitro* only, I am wondering if you could knock out or overexpress
the acetyltransferases (PatZ and YiaC) and deacetylase (CobB) *in vivo* and detect the acetylation changes of RutR. Here, I just have
concerns that *in vitro* tests cannot tell the whole story of enzymic control of RutR acetylation in real cellular conditions.

Answer:
The regulation of RutR function by lysine acetylation depends on the cellular concentrations of acetyl-CoA and CoA, i.e. it is the ratio that
determines KAT activity, the cellular concentrations of NAD⁺ as it is a stoichiometric co-substrate for CobB and of acetyl phosphate, which
drives non-enzymatic acetylation of RutR. These concentrations vary dependent on the metabolic state. Moreover, the RutR acetylation
state also depends on the expression level of the enzymes, i.e. KATs, CobB and the enzymes involved in generation of acetyl-CoA, i.e.
acetyl-CoA synthetases. However, if overexpressing/knocking out KATs or CobB other processes will be affected as well which makes it
really hard to ascribe an outcome to the presence/absence of RutR acetylation. To this end, we think our experimental setup is excellent to
define exactly what the role of lysine acetylation is even up to a resolution of the individual RutR lysine acetylation site.

**Point 3:**
In Figure 3B and Figure 5B, some error bar is too large, showing a high standard deviation in the original data set.

Answer:
In Fig. 3B there are no error bars shown but the 95% confidence interval. The error bars shown in Figure 5B are correct. These are the
variations of the experiments.

**Point 4:**

Although this study claims that RutR K52-acetylation cannot be mimicked by RutR mutations, figure 3D does show that K52Q has similar
effects as Ack52. So, it is not very accurate to draw a conclusion that RutR mutations could not functionally mimic K52 acetylation.

Answer:

We agree that the K52Q mutant shows the same trend. However, while the mutation of K52Q results in a drastic reduction in affinity towards
both *carA*- and *rutA*-DNA, K52-acetylation completely abolishes binding to DNA as shown by the ITC data. In that sense K52Q is not a
perfect mimic. Reviewer 3 is correct that in this specific assay the mutation K52Q can be regarded as good mimic for K52-acetylation as the
reduction in affinity observed for K52Q is sufficient to reduce the β -galactosidase activity similar to the empty-vector control, i.e. it abolishes
RutR transcriptional regulator activity. However, it depends on the cellular concentrations of RutR if this reduction in RutR-DNA-binding
affinity has a consequence. To this end, although K52Q was a perfect mimic in another experimental contexts it cannot be generalized for
every experiment but this is a criteria by definition that a perfect mimic has to meet. RutR K52R shows an app. 2-fold reduced binding
compared to non-acetylated RutR WT towards *carA*-DNA. The β -galactosidase-assay (Fig. 3C) shows that K52R results in a strong reduction
in β -galactosidase activity compared to RutR WT, although it should be the same as RutR WT if the arginine mutant was perfectly mimicking
the non-acetylated state of K52. Notably, K52R and K52Q are statistically different showing that in principle they can be used to mimic the
non-acetylated or acetylated state in this assay. As stated above this might be different in other experimental contexts and assays. We
rewrote this section to be a bit more circumspect.

We rephrased the sentence and the paragraph and reordered it to not undercut our results and make it clearer (see also answers t reviewer
2):

**From line 264:**

"For RutR K52Q, we observed a β -galactosidase activity similar to the empty vector control suggesting that in this assay under these
experimental conditions RutR K52Q is a reliable mimic for RutR K52-acetylation and that charge neutralization constitutes an important
mechanism to switch-off RutR transcriptional regulator activity (Fig. 3A)."

**Point 5:**

I understand that genetic code expansion is not an ideal choice when it comes to K7 and K11 acetylation *in vivo*, however, I notice that in
Figure 7A, the protein level of RutR K7Q/K11Q and RutR K7R/K11R are much lesser than wild type RutR. Maybe the loading control is
different, but I would prefer a quantitative analysis of this figure.

Answer:

Reviewer 3 is right in pointing out that the protein level of the RutR double mutants K7Q/K11Q and K7R/K11R is lower compared to RutR
WT. For the determination of the β -galactosidase activity in Miller units, the optical density of the culture is included. In these experiments
there is variation in the mean growth of cultures represented by the determined OD₆₀₀ values. Importantly, in our experiments there is only
a slight variation in the range typical for biological experiments. These differences in the OD₆₀₀ are taken into account upon calculation of
the Miller units reflecting normalized β -galactosidase activity. Another source of variation is the protein amount loaded onto the SDS-PAGE
gels for immunoblotting showing the expressed RutR WT and the mutants thereof. If we normalize the protein amount to the loading control,
i.e. total protein staining with 2,2,2-trichloroethanol, there is still a difference in protein expressed of RutR WT and particularly of the double
mutants RutR K7Q/K11Q and K7R/K11R. This shows that differences in the loading do not account for the observed differences in protein
levels. This indicates a third layer of variation in this assay, which is a difference in processes resulting in diminished protein levels, such as
translation efficiency/mRNA stability. These secondary effects are not assessed in this β -galactosidase assay. To this end, this assay allows
only a comparison of samples that show similar protein levels after normalization. This enables a direct comparison only of RutR WT to RutR
K52Q (group 1, Fig. 7, black), of the N-terminal RutR single-mutants among each other (group 2, Fig. 7, red) and of the RutR double-mutants
(Fig. 7, blue) among each other. We altered these analyses in the revised manuscript. Our data clearly shows that RutR K7Q/K11Q results
in a reduced β -galactosidase activity compared to RutR K7R/K11R. We do not have a mechanistical explanation for these different protein
levels at the molecular level. However, there are reports for TetR suggesting that the region at the 5'-end of the gene encoding RutR affects
the translation efficiency/mRNA stability resulting in lower protein amounts (Berens *et al.* (1997) *JBC* 267: 1945-1952). We neither observed
an instability of the RutR double- or single-mutants nor do they behave differently in analytical gel filtration experiments compared to RutR
WT. Importantly, as a support showing that acetylation at K7 and K11 affects RutR-transcriptional regulator activity *in vivo*, we analyzed the
double mutants RutR K7Q/K11Q and RutR K7R/K11R by ITC and additionally include the RutR K7R/K11R data in the revised manuscript.
These data show that RutR K7Q/K11Q can be used as a mimic for acetylation on RutR K7/K11 as it results in reduction in *carA*- and *rutA*-
DNA-binding affinity. In contrast, RutR K7R/K11R binds with almost RutR WT affinity to both DNA-promoter sequences, which shows that
these mutants can be used to study the impact of lysine acetylation on RutR DNA-binding *in vivo*. Overall, these data shows that RutR
K7Q/K11Q as a mimic for acetylation affects RutR transcriptional regulator activity *in vivo*. In Fig. 7, we removed all statistical comparisons
of effects elicited by proteins between the three groups (1. RutR WT, RutR K52Q; 2. RutR single-mutants; 3. RutR double mutants) and only
show comparisons of effects observed for proteins within each of the groups. Comparing performance of both RutR double mutants, which
are expressed at a similar level (Fig. 7A), supports that lysine acetylation at the RutR N-terminus affects the transcriptional regulator activity
*in vivo* as RutR K7R/K11R shows the same trend to RutR WT in β -galactosidase activity while the RutR K7Q/K11Q-acetylation mimicking
mutant shows a strong, statistically significant reduction of β -galactosidase activity if compared to RutR K7R/K11R.

**Point 6:**

Proofreading, Line 94 "RutR is" should be "RutR was"

Answer:

We think that "RutR is" is correct here as the statement is valid until today, i.e. the studies were performed and showed that RutR is important
for XY.

**Point 7:**
Proofreading, Line 145 “functons” should be “functions”

Answer:
This is corrected in the revised manuscript.

**Point 8:**
Proofreading, Line 407 “migh” should be “might”

Answer:
This is corrected in the revised manuscript.

**Reviewer #4 (Remarks to the Author):**

We thank reviewer 4 for the thorough revision and for the constructive comments to our manuscript and are really pleased that he/she is
enthusiastic about our work. We carefully worked on the suggestions and are confident that we concisely answered all points of concerns.
We think the manuscript has improved strongly.

**Point 1:**
The authors reported that lysine acetylation of RutR adapts gene expression to the cellular metabolic state. Acetylation of RutR at K52 leads
to repression of *carAB* mRNA level as shown in Figure 3B. Whether the cellular metabolic state is really changed due to RutR acetylation?
For example, the synthesis of pyrimidine, purine or arginine may be weakened when RutR is acetylated on above sites.

Answer:
Our main message is that the activity of RutR to act as a transcriptional regulator is regulated by post-translational lysine acetylation. This
post-translational modification is catalyzed by lysine acetyltransferases (KATs) and lysine deacetylases (KDACs). *E. coli* encodes for five
reported N(ε)-lysine acetyltransferases and by only one KDAC, i.e. the sirtuin deacetylase CobB. We show that RutR is acetylated by
YfiQ/PatZ (and non-enzymatically by acetyl phosphate) and deacetylated by CobB. Sensing of the metabolic state is done by KATs, that
depend on the central metabolic intermediate acetyl-CoA, which interconnects all pathways of the energy metabolism. Sirtuins are NAD⁺-
dependent enzymes, i.e. they use NAD⁺ as stoichiometric co-substrate for the deacetylation reaction. To this end, by sensing acetyl-CoA/CoA
ratio and the NAD⁺-level, through the action of KATs and sirtuins, the activity of RutR is directly translated into altered gene expression
programs. This is also achieved non-enzymatically by cellular acetyl phosphate, the major molecule for non-enzymatic acetylation in bacteria.
However, while KAT driven acetylation occurs fast and highly dynamic, the non-enzymatic acetylation by gradient accumulation of acetyl
phosphate occurs with a slower kinetics, which in turn is also manifested in a slow kinetics of RutR acetylation. We show that RutR-acetylation
impairs binding to the *rutAG*-promoter or to the *carAB*-promoter. Moreover, we show by qRT-PCR that K52-acetylation of RutR re-represses
the transcription of the *carAB*-operon. This result is confirmed by the observed decreased mRNA level of the *carAB-lacZ* reporter by qRT-
PCR and by measurement of β-galactosidase activity. All these data strongly suggest that it will affect the levels of pyrimidines, purines and
arginine. Notably, we were not able to show that the repression of transcription of the *rutAG* operon by RutR WT is abolished by K52-
acetylated RutR. This suggests that the expression of the *rutAG*-operon is repressed by further, unknown mechanisms. It is reported in
several publications that RutR is a transcriptional repressor for the *rutAG* operon resulting in expression of genes encoding for
proteins/transporters needed for pyrimidine breakdown and it is an indirect activator for expression of the *carAB*-operon encoding proteins
involved in pyrimidine/purine/arginine biosynthesis (Shimada *et al.* (2007) *Mol. Microbiol.* 66:744–757; Bouvier *et al.* (1984) *Proc Natl Acad*
*Sci USA* 81: 4139–4143; Loh *et al.* (2006) *Proc Natl Acad Sci USA* 103: 5114–5119). The main focus of this manuscript is on the regulation
of RutR function by post-translational lysine acetylation. We show here that acetylation of RutR affects DNA-binding to both, *carAB*- and
*rutAG*-promoter DNA, *in vitro* and *in vivo*. To this end, it is likely that this will be manifested also in the cellular pyrimidine/purine/arginine
levels under physiological conditions.

**Point 2:**
RutR binds to the *carAB* promoter to activate its expression in the absence of uracil to promote de novo synthesis of pyrimidines. It means
*carAB*-operon is repressed by the negative transcriptional regulator PepA in presence of uracil to favor the metabolic process for degradation
of pyrimidines. When the RutR Ack52 protein was crystallised in complex with uracil, the authors showed that the acetylation at K52 does
not interfere with uracil binding. From Figure 4A, the N-terminal residues of RutR also seems to far away from uracil binding sites. What's
the physiological role of RutR acetylation to regulate *carAB* expression? Based on the authors' results, it seems that the acetylation of RutR
can regulate *carAB* expression regardless of concentration of uracil in bacteria.

Answer:
The regulation of RutR function by lysine acetylation is independent from the regulation of DNA-binding by uracil. The metabolic state is
sensed by KATs using acetyl-CoA as donor molecule for the acetylation and the sirtuin deacetylase CobB, which is an NAD⁺-dependent
enzyme using stoichiometric concentrations of NAD⁺ to catalyze substrate deacetylation. Thereby, the activity of KATs/CobB depends on
the prevalence of acetyl-CoA/CoA and NAD⁺ which is translated into the acetylation state of RutR and in turn affecting RutR's capability to
bind DNA and activate/repress *carAB* or *rutAG* expression. The regulation of RutR transcriptional regulator activity by uracil is another layer
of regulation. On the one hand, uracil-binding to the ligand-binding domain of RutR impairs interaction with DNA by stabilizing RutR in a
conformation that is incompatible with DNA-binding. On the other hand, RutR-DNA-binding also results in dissociation of uracil from the
ligand-binding domain. The effect of uracil on the RutR DNA-binding capacity depends on the cellular concentration of uracil. At higher
concentration of uracil the equilibrium is shifted towards a conformation of RutR that is incapable to bind to DNA. Under these conditions the
*rutAG*-operon is expressed, i.e. RutR as a transcriptional repressor does not bind to the *rutAG* promoter, resulting in pyrimidine degradation.
Besides, the expression of the *carAB*-operon encoding enzymes important for pyrimidine biosynthesis is indirectly repressed as RutR does
not bind to the *carAB*-promoter and the repressor PepA can bind. The reviewer is right in saying that acetylation does not interfere with uracil

binding. Precisely stated, in absence of DNA RutR acetylation does not interfere with uracil binding. To show the impact of uracil on binding
of acetylated RutR to DNA, we performed additional EMSAs analyzing the impact of uracil on DNA-binding of RutR WT and acetylated RutR
(see Supp. Fig. S14). These data suggest that acetylation of RutR and uracil act via distinct mechanisms to impair RutR-DNA-binding as the
effect observed is additive. These EMSA-assays are shown in the Supplemental data section of the revised manuscript (Supp. Fig. S14).

**Point 3:**

The authors mentioned that *rutAG* operon is expressed under nitrogen limitation in an NtrC-dependent manner (line 83), so whether nitrogen
limitation was used in all *rutAG* expression-related experiments? Moreover, what is the condition(s) for *carAB* operon expression?

Answer:

In our *in vivo* studies, U65 $\Delta rutR$ with a genomic *PcarA-lacZ* fusion was transformed with pRSF empty, pRSF *rutR*-His₆, or pRSF *rutR*-His
K52amb. 150 mL LB with 10 mM Ack and 20 mM NA was inoculated to an OD₆₀₀ of 0.05 in the presence of indicated concentrations of
isopropyl- β -D-thiogalactoside (IPTG). For RNA isolation or measurement of β -galactosidase activity as a direct indicator for RutR
transcriptional regulator activity samples were taken of cells in the exponential growth phase to stationary phase. Under these conditions we
can conclude that the medium is not completely deprived of nitrogen. We observed that expression of the *carAB* operon was upregulated
upon expression of RutR WT. This was repressed by expression of K52-acetylated RutR.

The regulation of RutR function by lysine acetylation depends on the cellular concentrations of acetyl-CoA and CoA, i.e. it is the ratio that
determines KAT activity, the cellular concentrations of NAD⁺ as it is a stoichiometric co-substrate for CobB and of acetyl phosphate, which
drives non-enzymatic acetylation of RutR. These concentrations vary dependent on the metabolic state. Moreover, the RutR acetylation
state also depends on the expression level of the enzymes, i.e. KATs, CobB and the enzymes involved in generation of acetyl-CoA, i.e.
acetyl-CoA synthetases. In our system, the acetylation state of RutR does not depend on the nitrogen supply. We are capable to analyze
RutR lysine acetylation at a resolution of an individual site. To this end, we think our experimental setup is excellent to define exactly what
the role of lysine acetylation for RutR function. See point 1: It is reported in several publications that RutR is a transcriptional repressor for
the *rutAG* operon resulting in expression of genes encoding for proteins/transporters needed for pyrimidine breakdown and it is an indirect
activator for expression of the *carAB*-operon encoding proteins involved in pyrimidine/purine/arginine biosynthesis (Shimada *et al.* (2007)
*Mol. Microbiol.* 66:744–757; Bouvier *et al.* (1984) *Proc Natl Acad Sci USA* 81: 4139–4143; Loh *et al.* (2006) *Proc Natl Acad Sci*
*USA* 103: 5114–5119). It is reported that *rutA* expression is high under conditions of high nitrogen:carbon ratio, i.e. under these conditions,
the repressor RutR should not bind to the *rutAG* promoter. This might be regulated by RutR acetylation (Schilling *et al.*, *J Bacteriol.* 201(9):
e00768-18). The main focus of this manuscript is on the regulation of RutR function by post-translational lysine acetylation. We show here
that acetylation of RutR affects DNA-binding to both, *carAB*- and *rutAG*-promoter DNA, *in vitro* and *in vivo*. To this end, it is likely that this
will be manifested also in the cellular pyrimidine/purine/arginine levels under physiological conditions.

**Point 4:**

The authors presented Figure 1A-1D in Introduction section, this writing style is uncommon, so please reorganize Introduction section and
Figure 1.

Answer:

We think that the reviewer is right and it is not the majority of articles in which figures are cited in the introduction section. However, if looking
at Nature Communication papers we observed in several articles that there are figures cited in the introduction section. As we think that
these figures support the text and allow a better understanding of the topic we would like to keep it as it is unless the editor wants us to
reorganize it. Moreover, if not showing these figures in the introduction we would need to repeat several points in the results section, which
is unnecessary if organized it this way.

**Point 5:**

Line 125-126, the authors stated that “Such mechanisms were shown for the transcriptional regulators CRP and RcsB”. Some key references
should be cited here, including PMID: 26943369, PMID: 28329249, PMID: 28118511, PMID: 29899473, PMID: 36700638 and PMID:
30866760.

Answer:

We included these references.

**Point 6:**

Line 253-“To obtain a similar level of acetylated and non-acetylated RutR, we adjusted the concentration of IPTG used for induction of
expression (non-acetylated RutR: 10 μ M IPTG; K52-acetylated RutR: 1 mM IPTG).” While in Figure 3A, the protein level of 1000 μ M Ack52
is higher than 10 or 15 μ M non-acetylated RutR. Please explain this issue.

Answer:

We had to adjust the concentration of IPTG to achieve a similar expression level of non-acetylated RutR and acetylated RutR. The reviewer
is right in suggesting that the anti-His₆-blot signal and therefore the protein level is slightly higher for the K52-acetylated protein. However,
the mRNA level determined by RT-qPCR (Fig. 3B; now Fig. 3D) shows similar mRNA levels for non-acetylated RutR WT and K52-acetylated
RutR. As our ITC data show that RutR K52-acetylation does completely abolish binding of RutR to DNA even a slightly higher protein level
of K52-acetylated RutR compared to RutR WT would not affect these results. In fact, that would even be true if there were substantial higher
concentrations of K52-acetylated RutR compared to RutR WT. K52-acetylation results in completely switching off RutR DNA-binding. We
think that we were indeed able to titer the protein levels quite precisely, i.e. the level of K52-acetylated RutR generated by the GCEC and
the level of non-acetylated RutR WT.

**Point 7:**

4. Line 419-“Both single-acetylated RutR-His6 AcK7 and AcK11 bind to both dsDNA fragments similarly to non-acetylated RutR-His6.
However, EMSAs suggest reduced binding of double-acetylated RutR-His6 AcK7/11 to both dsDNA fragments.” While in Figure 6, it seems

single-acetylated RutR-His6 Ack7 and Ack11 impaires the ability of RutR binding to *carA* promoter, and double-acetylated RutR-His6
Ack7/11 restores the binding ability to *carA* promoter.

Answer:

We observed for all EMSAs performed with the *carA*-promoter DNA that the signal is fainter compared to the *rutA*-promoter, particularly of
the higher molecular weight signal that represents the RutR-DNA complex. We assume that this is due to the intercalation of ethidium
bromide into the *carA*-DNA, which is weaker compared to the *rutA*-promoter DNA. It was reported that ethidium bromide intercalation is to
some extent sequence dependent favoring AT-rich sequences (Galindo-Murillo *et al.* (2021) *Nucleic Acids Research* 49: 73735–3747).
Comparing *carA*- and *rutA*-DNA, there are 36 AT-base pairs in *rutA*-DNA and 31 AT-base pairs in *carA*-DNA. This might explain the observed
weaker ethidium bromide signals for *carA*-DNA compared to *rutA*-DNA. By evaluating the data, it is important to judge the complex formation
between RutR and DNA by inspecting the signal representing the level of free DNA at the bottom of the gels. With increasing amount of
complex formed upon addition of RutR/acetylated RutR the amount of this signal representing free DNA is decreasing. By judging complex
formation of this signal you will see that the complexes are formed also for Ack7 and Ack11. EMSAs are semi-quantitative assays. To
quantify the interaction and to thermodynamically characterize the RutR-DNA interactions, we performed ITC and SPR measurements.
These data consistently show a decreased binding affinity of the double-acetylated RutR Ack7 and Ack11 to both *rutA*- and *carA*-DNA. We
included a statement in the revised manuscript to explain that we observe a fainter signal in EMSAs with *carA*-DNA compared to *rutA*-DNA
that might be due to lower AT-content in *carA*-DNA compared to *rutA*-DNA.

We added in the Material and Methods section:

**From line 931:**

"For the box_{carA} dsDNA we observed a fainter signal compared to box_{rutA} dsDNA. This might be due to lower AT-content in box_{carA} dsDNA
compared to box_{rutA} dsDNA¹⁸."

**Point 8:**

5. Line 510-"Although the mutation RutR K7Q alone does not result in a significant reduction of RutR", whether it means it is a poor mimic
for the acetylated state.

Answer:

The observation that RutR K7Q binds to both, *rutA*- and *carA*-DNA almost as RutR WT, while the K7-acetylated RutR resulting in an app. 2
to 2.5-fold reduction in affinity compared to non-acetylated RutR WT, suggests that by mutation of lysine to glutamine not all aspects of K7-
acetylation can be mimicked. It is likely that K7-acetylation exerts its effect via electrostatic and steric components of which both are not
mimicked by K7Q. So if you judge it mechanistically on the molecular level, it is an imperfect molecular mimic. For us it is important to say
that it has to be evaluated on a site-specific basis if the K to Q and K to R mutants are reliable as acetylation mimetic or charge-conserving
mimic for the non-acetylated state.

**Point 9:**

6. Line 543- "18nalysezyzed PatZ/YiaC acetylated" should be a typo.

Answer:

This is corrected in the revised manuscript.

**Point 10:**

7. What is the conservation of lysine residues (K7, 11, 52, 62, 95 and 150) of RutR among TetR family?

Answer:

We included an alignment of TetR-family members (Supp. Fig. S11A). This allows you to analyze the conservation of the lysine side chains.
Supplementary table S10 shows furthermore positively charged residues in the N-termini of selected TetR-related transcriptional regulators.
Moreover, we also present an alignment of RutR proteins (Supp. Fig. S10). This shows that K7 and K11 are almost totally conserved. K21
is not totally conserved. Interestingly, several RutR proteins have a Q at this position suggesting that the acetylated state would be the basic
state. K52 and K62 are totally conserved comparing all RutR-proteins. At K95 of *E. coli* RutR, most proteins show either a K or an R,
suggesting that the positive charge is important at that position. *E. coli* RutR K150 is the least conserved residue.

**Point 11:**

8. Is acetylation of K21, K95 or K150 involved in regulating RutR DNA-binding ability?

Answer:

As judged from the EMSAs that were shown in the manuscript, the DNA-binding is not affected. When inspecting the position of these lysine
side chains it becomes obvious that these lysine side chains are not within the DNA-binding helix-turn-helix motif. We performed additional
ITC-measurements of RutR Ack21, Ack95 and Ack150 to show that these acetylations do not affect DNA-binding to neither *carA*- nor *rutA*-
DNA (Supp. Fig. S15, Supp. Table S3).

**Point 12:**

9. Lysine residues are targeted by a particularly high number of PTMs including acetylation, methylation, succinylation and lactylation. So
whether other PTMs are involved in regulation of RutR besides acetylation?

Answer:

Reviewer 4 raised an interesting question. So far it is not known whether RutR is targeted by other PTMs such as methylation, succinylation
or lactylation. This would be an interesting future research direction to analyze whether RutR is regulated by these PTMs and how/when

these PTMs exert functionally different effects. Our ESI-MS data suggest that the purified RutR protein is not modified by any other PTM as
we do only see one molecular species with the correct molecular mass representing either non-acetylated RutR or acetylated RutR. This
does of course not exclude the possibility that under certain physiological conditions, RutR is modified by other PTMs.

**References:**

- 1. Johnson, D.B. et al. RF1 knockout allows ribosomal incorporation of unnatural amino acids at multiple sites. *Nat Chem Biol* **7**, 779-86
(2011).
- 2. Isaacs, F.J. et al. Precise manipulation of chromosomes in vivo enables genome-wide codon replacement. *Science* **333**, 348-53 (2011).
- 3. Mukai, T. et al. Codon reassignment in the Escherichia coli genetic code. *Nucleic Acids Res* **38**, 8188-95 (2010).
- 4. Fan, C., Xiong, H., Reynolds, N.M. & Soll, D. Rationally evolving tRNAPyl for efficient incorporation of noncanonical amino acids. *Nucleic*
*Acids Res* **43**, e156 (2015).
- 5. Berens, C., Altschmied, L. & Hillen, W. The role of the N terminus in Tet repressor for tet operator binding determined by a mutational
analysis. *J Biol Chem* **267**, 1945-52 (1992).
- 6. Nguyen Ple, M., Bervoets, I., Maes, D. & Charlier, D. The protein-DNA contacts in RutR*carAB operator complexes. *Nucleic Acids Res*
**38**, 6286-300 (2010).
- 7. Allfrey, V.G., Faulkner, R. & Mirsky, A.E. Acetylation and Methylation of Histones and Their Possible Role in the Regulation of Rna
Synthesis. *Proc Natl Acad Sci U S A* **51**, 786-94 (1964).
- 8. de Diego Puente, T. et al. The Protein Acetyltransferase PatZ from Escherichia coli Is Regulated by Autoacetylation-induced
Oligomerization. *J Biol Chem* **290**, 23077-93 (2015).
- 9. Christensen, D.G. et al. Identification of Novel Protein Lysine Acetyltransferases in Escherichia coli. *MBio* **9**(2018).
- 10. Vetting, M.W., Bareich, D.C., Yu, M. & Blanchard, J.S. Crystal structure of RimI from Salmonella typhimurium LT2, the GNAT responsible
for N(alpha)-acetylation of ribosomal protein S18. *Protein Sci* **17**, 1781-90 (2008).
- 11. Lu, J., Wang, X., Xia, B. & Jin, C. Solution structure of Apo-YjaB from Escherichia coli. *Proteins* **76**, 261-5 (2009).
- 12. Favrot, L., Blanchard, J.S. & Vergnolle, O. Bacterial GCN5-Related N-Acetyltransferases: From Resistance to Regulation. *Biochemistry*
**55**, 989-1002 (2016).
- 13. Xie, L., Zeng, J., Luo, H., Pan, W. & Xie, J. The roles of bacterial GCN5-related N-acetyltransferases. *Crit Rev Eukaryot Gene Expr* **24**,
77-87 (2014).
- 14. de Boor, S. et al. Small GTP-binding protein Ran is regulated by posttranslational lysine acetylation. *Proc Natl Acad Sci U S A* **112**,
E3679-88 (2015).
- 15. Kremer, M., Kuhlmann, N., Lechner, M., Baldus, L. & Lammers, M. Comment on 'YcgC represents a new protein deacetylase family in
prokaryotes'. *Elife* **7**(2018).
- 16. Tu, S. et al. YcgC represents a new protein deacetylase family in prokaryotes. *Elife* **4**(2015).
- 17. Whicher, A., Camprubi, E., Pinna, S., Herschy, B. & Lane, N. Acetyl Phosphate as a Primordial Energy Currency at the Origin of Life.
*Orig Life Evol Biosph* **48**, 159-179 (2018).
- 18. Galindo-Murillo, R. & Cheatham, T.E. Ethidium bromide interactions with DNA: an exploration of a classic DNA-ligand complex with
unbiased molecular dynamics simulations. *Nucleic Acids Res* **49**, 3735-3747 (2021).
- 19. Schilling, B. et al. Global Lysine Acetylation in Escherichia coli Results from Growth Conditions That Favor Acetate Fermentation. *J*
*Bacteriol* **201**(2019).
- 20. Christensen, D.G., Orr, J.S., Rao, C.V. & Wolfe, A.J. Increasing Growth Yield and Decreasing Acetylation in Escherichia coli by
Optimizing the Carbon-to-Magnesium Ratio in Peptide-Based Media. *Appl Environ Microbiol* **83**(2017).
- 21. Baeza, J. et al. Stoichiometry of site-specific lysine acetylation in an entire proteome. *J Biol Chem* **289**, 21326-38 (2014).

Reviewer #1 (Remarks to the Author):

The manuscript has been greatly improved after revision. However, some of my concerns are still there:

- 1) Multiple lysine residues of the protein can be acetylated through different mechanisms. Thus, RutR seems not to be an ideal target to be explored by the new technology at least at the present stage. I am not convinced that the new technology would be able to do multiple sites.
- 2) Usually, when a new technology is developed or introduced, comparisons between it and the "old ones" are necessary. So I do not understand why the authors refused to delete known genes responsible for reversible lysine acetylation of RutR.

Reviewer #2 (Remarks to the Author):

The authors have taken the previous review to heart and the result is an improved manuscript. This is a well-written report that describes a well-designed, well-executed, rigorous, complete, and impactful study. It could be used as a template for other studies of posttranslational modifications in bacteria. I have only a few grammatical and referencing suggestions.

Line 102: "shown" instead of "identified" would be the better verb.

Line 130: May I recommend a different sentence structure? "In *E. coli*, the best studied KAT is PatZ/YfiQ. Recently, four additional Gcn5-like KATs can act as...." Put the PatZ/YfiQ specific references after the first sentence, but I think your citations should also include at least one from Escalante-Semerena's lab. Jorge's group discovered it in *Salmonella* and did most of the characterization – I suggest the van Drisse review because it reviews the initial work in *Salmonella* but also mentions the *E. coli* homolog. Put the Christensen review at the end of the second sentence.

Line 138: Again, I recommend that you cite Escalante-Semerena. Although his work was in *Salmonella*, he discovered the protein and did most of the characterization. Perhaps a review that also mentions the *E. coli* homolog - maybe the van Drisse review would do.

Line 326: The comma belongs after dsDNA.

Line 409: I suggest "...YiaC or it is...."

Line 410: I suggest "Previously, we observed that...."

Reviewer #3 (Remarks to the Author):

I appreciate that authors answer my points one by one clearly and detailedly. I agree with most of the answers, but I still have a few recommendations to mention.

For point 1, I agree with the authors' point that it is somehow impossible and complicated to learn the site-specific effects of acetylation on RutR's function under conditions like fuel switching, however, it is also an opportunity for authors to think about some future studies to build up a proper system to address the relationship between RutR acetylation and complicated conditions like fuel switching.

For point 2, I understand the authors' points. However, it is not appropriate to directly conclude the enzyme-substrate relationship only based on an *in vitro* study. As the authors state acetylation is a complicated process *in vivo*, and depending on many factors, a straightforward *in vitro* acetylation assay will not be able to tell the whole story *in vivo*. I would like authors to at least do immunoprecipitation to show the interaction between the acetylation-related enzyme and RutR *in vivo*.

For point 5, just like point 1, it can be another study to learn how N terminal acetylation affects the translation of RutR and how these effects further impact on RutR's function in transcription in response to cellular metabolic state.

For points 3,4,6,7 and 8, I thank you for the answer.

Thanks again for your answers.

Reviewer #4 (Remarks to the Author):

The authors have addressed all my concerns, I do not have further questions.

Point-by-point response to the reviewer comments

Reviewer 1:

We thank reviewer 1 for carefully reading our manuscript and giving these valuable suggestions. As reviewer 1 still has some open questions we address these here.

Point 1:

Multiple lysine residues of the protein can be acetylated through different mechanisms. Thus, RutR seems not to be an ideal target to be explored by the new technology at least at the present stage. I am not convinced that the new technology would be able to do multiple sites.

Answer:

This technology is valuable to study the impact of lysine acetylation at individual sites. However, it is possible to simultaneously also incorporate acetyl-lysine site-specifically at several positions. A drawback is that the yield of protein acetylated at multiple sites is decreasing. This can partly be compensated by using *E. coli* strains that carry genomic replacement of all endogenous amber-stop codons (UAG) to ochre (UAA) codons. Moreover, *E. coli* strains have been developed that have a genomic deletion of the gene encoding for the release factor 1 (RF1) that competes with PylT for binding to the amber stop codon at the ribosome. This is added in the discussion section. Reviewer 1 is right in stating that acetylation at multiple sites is not assessed by site-specific incorporation of acetyl-lysine at one specific site. However, that is the strength of this technology to assess the real impact of an individual acetylation site to regulate protein function. At this stage it was not our intention to study RutR function by acetylation present simultaneously at several sites, although this could occur *in vivo*. In our *in vivo* experiments we always compared the impact of lysine acetylation at the individual site to the non-acetylated RutR protein. If RutR is acetylated at other sites than the incorporated acetyl-lysine during the experiment in the *E. coli* cells, this accounts for both proteins, i.e. the RutR WT protein and the site-specifically acetylated RutR protein. To this end, it is possible to attribute the effects we observe to the site-specifically incorporated acetyl-lysine. Moreover, all lysine acetylation events result in a reduction of DNA-binding affinity. Even if RutR were acetylated at other sites apart from the site-specifically incorporated acetyl-lysine the impact would be additive resulting in an even reduced binding of RutR to DNA, i.e. all known acetylation sites either affect RutR transcriptional regulator activity or have no effect at all (e.g. AcK 95, AcK 150). The situation were different if some sites impaired RutR DNA-binding while others improved RutR DNA-binding. For AcK7, AcK11, AcK52 and AcK62 we observe a reduction in RutR DNA-binding, all other acetylation sites have no effect, i.e. even if there were acetylation at other lysine side chains this would not interfere with RutR DNA-binding. We think that the technology based on genetic code expansion is of great value even because we are able to study individual sites and allows use the site-specifically acetylated protein in their natively-folded state as substrates to study enzymatically catalyzed deacetylation. This ensures that the effect can be attributed unambiguously to a specific lysine-acetylation event.

Point 2:

Usually, when a new technology is developed or introduced, comparisons between it and the “old ones” are necessary. So I do not understand why the authors refused to delete known genes responsible for reversible lysine acetylation of RutR.

Answer:

From our perspective it is not really possible to compare the strategies of performing deletions of genes encoding for acetyltransferases, deacetylases, acetyl-CoA-/acetyl-phosphate-/NAD⁺-generating enzymes to study the effect on protein function with the strategy based on genetic code expansion, which generates site-specifically lysine-acetylated proteins, presented here. Both strategies are valuable and compensate each other but they yield different information. While gene deletions can have diverse effects in cells, it is often not easy to attribute this to a specific post-translational modification event. Studying RutR acetylation with deletions in the pathways responsible for acetylation, deacetylation and generation of acetyl-CoA and NAD⁺ would result in valuable information but gave no information on the regulation of RutR function by lysine acetylation at a resolution of the individual site.

Reviewer 2:

We again thank Prof. Wolfe for his detailed and excellent review that resulted in a strongly improved manuscript. We agree on all the remaining open questions and included all of them in the revised manuscript.

Point 1:

Line 102: "shown" instead of "identified" would be the better verb.

Answer:

We replaced "identified" by "shown"

Point 2:

Line 130: May I recommend a different sentence structure? "In E. coli, the best studied KAT is PatZ/YfiQ. Recently, four additional Gcn5-like KATs can act as...." Put the PatZ/YfiQ specific references after the first sentence, but I think your citations should also include at least one from Escalante-Semerena's lab. Jorge's group discovered it in Salmonella and did most of the characterization – I suggest the van Drisse review because it reviews the initial work in Salmonella but also mentions the E. coli homolog. Put the Christensen review at the end of the second sentence.

Answer:

We adjusted the sentence structure as suggested and included the references. See below a list of the references we included in the revised manuscript.

Point 3:

Line 138: Again, I recommend that you cite Escalante-Semerena. Although his work was in Salmonella, he discovered the protein and did most of the characterization. Perhaps a review that also mentions the E. coli homolog - maybe the van Drisse review would do.

Answer:

We cited the papers according to the suggestion.

Point 4:

Line 326: The comma belongs after dsDNA.

Answer:

We corrected this.

Point 5:

Line 409: I suggest "...YiaC or it is...."

Answer:

We rewrote the sentence as suggested.

Point 6:

Line 410: I suggest "Previously, we observed that...."

Answer:

We rewrote the sentence as suggested.

Reviewer #3:

We thank reviewer 3 again for carefully reviewing our manuscript, which strongly improved it. We hope that we can solve all open questions.

Point 1:

For point 1, I agree with the authors' point that it is somehow impossible and complicated to learn the site-specific effects of acetylation on RutR's function under conditions like fuel switching, however, it is also an opportunity for authors to think about some future studies to build up a proper system to address the relationship between RutR acetylation and complicated conditions like fuel switching.

Answer:

Thank you for the suggestion. That is a really interesting future research direction.

Point 2:

For point 2, I understand the authors' points. However, it is not appropriate to directly conclude the enzyme-substrate relationship only based on an *in vitro* study. As the authors state acetylation is a complicated process *in vivo*, and depending on many factors, a straightforward *in vitro* acetylation assay will not be able to tell the whole story *in vivo*. I would like authors to at least do immunoprecipitation to show the interaction between the acetylation-related enzyme and RutR *in vivo*.

Answer:

Reviewer 3 is right in saying that it is sometimes not possible to directly transfer results obtained from *in vitro* studies to the physiological situation. However, we think that in this case the results can be transferred to the situation *in vivo* for several reasons.

Firstly, our system uses natively-folded protein for the acetyltransferase-catalyzed acetylation of RutR and even site-specifically lysine-acetylated and natively-folded RutR was used to assess CobB-catalyzed RutR-deacetylation. This represents the physiological situation much more than using peptides as substrates for deacetylases or acetyltransferases. Peptides do not form a three-dimensional structure similar to that of the substrate protein, which is a major drawback, but are still often used for acetylation/deacetylation experiments to identify enzymes by other labs.

Secondly, we show that all enzymes behave as reported by other labs. Acetyltransferases are active as shown by their auto-acetyltransferase activity (for PhnO, PatZ/YfiQ, RimI, YjaB) or the activity to acetylate RutR (for YiaC and PatZ/YfiQ). The deacetylase CobB shows an NAD⁺-dependent deacetylase activity towards the site-specifically acetylated RutR and even towards the RutR protein that was acetylated with the acetyltransferases PatZ/YfiQ and YiaC.

Thirdly, the enzymes show a high degree of site specificity. Neither do all *E. coli* acetyltransferases acetylate RutR, which would suggest a high level of promiscuity nor do they acetylate several lysines in RutR but instead are specific for only selected sites. Along this line, also for CobB we observe that it deacetylates RutR only at selected sites although it has been regarded to be a quite promiscuous enzyme as it is the only deacetylase encoded by *E. coli*. Moreover, we show that the CobB is NAD⁺-dependent as would be expected for a sirtuin-catalyzed deacetylation *in vivo*.

Reviewer 3 wants us to show an interaction of the RutR with the enzymes, i.e. the acetyltransferases PatZ/YfiQ and YiaC and the deacetylase CobB, to conclude the existence of an enzyme-substrate relationship occurring *in vivo*. For this purpose, he suggested to perform an immunoprecipitation to proof the interaction of RutR and the enzymes. We have to regret that this will be an almost impossible task. An enzyme-substrate complex is only very transiently formed due to high dissociation rate constant and a low association rate constant resulting in a very low affinity between an enzyme and its substrate, i.e. as soon as the substrate is converted the complex dissociates. We have shown that the acetylation sites RutR AcK7 and AcK11 are very efficiently enzymatically regulated suggesting the formation of very transient enzyme-substrate complexes. As a consequence, it will be impossible to immunoprecipitate detectable levels of the complexes with the enzymes unless the RutR protein were to form a complex independently from the enzyme-substrate relationship, in which case it were no proof that RutR is a substrate of the enzymes. As an example, calculation of the concentration of a complex using the experimentally shown intracellular concentration for RutR of 500 nM gave a 500 nM complex if the affinity were 1 nM. If the affinity was tenfold lower, but with 10 nM still very high, it resulted in only 50 nM complex and again tenfold lower, i.e. with 100 nM still high affinity not observable for a transient enzyme-substrate complex, resulted in only 5 nM complex formed between RutR and the enzymes. This cannot be detected by immunoprecipitation at endogenous levels of

the proteins. Many labs identify substrates of acetyltransferases and deacetylases by quantitative mass-spectrometry by either overexpressing enzymes or genomic knockout of the genes encoding for the enzymes and compare acetylation states of proteins compared to wildtype. However, these experiments do not always result in the identification of direct substrates, as they might result from indirect mechanisms and must be validated in follow up studies similar to experiments we performed here in our study. Others use microarrays with acetylated peptides to identify deacetylase substrates. However, these do not account for the proteins' three-dimensional structure which is an important determinant for enzyme-substrate specificity. From our point of view our experimental approach has the advantage to reveal direct substrates in their properly folded three-dimensional state. From our point of view the only requirement for an enzyme-substrate relationship existing *in vivo* is that the proteins are both present within the cell expressed at the same time. Several publications showed that the deacetylase CobB is constitutively expressed in *E. coli* with its activity being dependent on the availability of the stoichiometric co-substrate NAD⁺.^{6,7} The expression of PatZ was shown to be upregulated in stationary phase when cells grow in presence of glucose as carbon source and in all growth phases when cells are cultivated with acetate as carbon source.^{6,7} We and others show that RutR protein is present under all growth phases supporting the physiological importance of the regulation by lysine acetylation.⁸ We do not see any possibility to experimentally tackle the question that is raised by reviewer three to demonstrate the presence of enzyme-substrate complexes *in vivo* under physiological conditions. This is a general issue if investigating post-translational modifications. To show a direct interaction between a substrate and enzyme by immunoprecipitation is difficult *per se* but to detect a transient enzyme-substrate complex depending on the presence/absence of a post-translational modification at an individual site is even more challenging if not impossible.

Point 3:

For point 5, just like point 1, it can be another study to learn how N terminal acetylation affects the translation of RutR and how these effects further impact on RutR's function in transcription in response to cellular metabolic state.

Answer:

We thank the reviewer for pointing this out. As it happens sometimes in research you obtain unexpected results and this is one of them. We totally agree that it is an interesting research question to understand how the N-terminal residues affect translation of RutR. Maybe this guides us into another interesting research project, which might be interesting to understand bacterial translation regulation in a broader context. This shows that also these unstructured regions can be highly important for regulation at the post-transcriptional and/or post-translational level.

Reviewer 4:

Answer:

We thank reviewer 4 for reading and reviewing our manuscript.

References included in the revised manuscript:

1. Thao, S. & Escalante-Semerena, J.C. Biochemical and thermodynamic analyses of Salmonella enterica Pat, a multidomain, multimeric N(epsilon)-lysine acetyltransferase involved in carbon and energy metabolism. *mBio* **2**(2011).
2. Starai, V.J. & Escalante-Semerena, J.C. Identification of the protein acetyltransferase (Pat) enzyme that acetylates acetyl-CoA synthetase in Salmonella enterica. *J Mol Biol* **340**, 1005-12 (2004).
3. VanDrisse, C.M. & Escalante-Semerena, J.C. Protein Acetylation in Bacteria. *Annu Rev Microbiol* **73**, 111-132 (2019).
4. Tucker, A.C. & Escalante-Semerena, J.C. Biologically active isoforms of CobB sirtuin deacetylase in Salmonella enterica and Erwinia amylovora. *J Bacteriol* **192**, 6200-8 (2010).
5. Tsang, A.W. & Escalante-Semerena, J.C. CobB, a new member of the SIR2 family of eucaryotic regulatory proteins, is required to compensate for the lack of nicotinate mononucleotide:5,6-dimethylbenzimidazole phosphoribosyltransferase activity in cobT mutants during cobalamin biosynthesis in Salmonella typhimurium LT2. *J Biol Chem* **273**, 31788-94 (1998).
6. Castano-Cerezo, S., Bernal, V., Blanco-Catala, J., Iborra, J.L. & Canovas, M. cAMP-CRP co-ordinates the expression of the protein acetylation pathway with central metabolism in Escherichia coli. *Mol Microbiol* **82**, 1110-28 (2011).
7. Castano-Cerezo, S. et al. Protein acetylation affects acetate metabolism, motility and acid stress response in Escherichia coli. *Mol Syst Biol* **10**, 762 (2014).
8. Shimada, T., Hirao, K., Kori, A., Yamamoto, K. & Ishihama, A. RutR is the uracil/thymine-sensing master regulator of a set of genes for synthesis and degradation of pyrimidines. *Mol Microbiol* **66**, 744-57 (2007).

Reviewer #1 (Remarks to the Author):

No further comments.

Reviewer #2 (Remarks to the Author):

The authors have responded positively to my earlier concerns and comments. I am satisfied. This is very good work, well designed, well executed, and well written study that will make an important impact in the field.

Reviewer #3 (Remarks to the Author):

Thanks for the authors' responses!

For point 2, I take the point that it is challenging to do IP to show the transient interaction between acetyltransferase or deacetylase and the target protein.

So here, I have no other comments.